# EVODiff: Entropy-aware Variance Optimized Diffusion Inference

**Shigui Li**
School of Mathematics
South China University of Technology
Guangzhou, China
sgl.shiguili@gmail.com

**Wei Chen**
School of Mathematics
South China University of Technology
Guangzhou, China
maweichen@mail.scut.edu.cn

**Delu Zeng** *
School of Electronic and Information Engineering
South China University of Technology, Guangzhou, China;
Department of Electrical and Computer Engineering
University of Waterloo, Waterloo, Canada
dlzeng@scut.edu.cn

## Abstract

Diffusion models (DMs) excel in image generation but suffer from slow inference and training-inference discrepancies. Although gradient-based solvers for DMs accelerate denoising inference, they often lack theoretical foundations in information transmission efficiency. In this work, we introduce an information-theoretic perspective on the inference processes of DMs, revealing that successful denoising fundamentally reduces conditional entropy in reverse transitions. This principle leads to our key insights into the inference processes: (1) data prediction parameterization outperforms its noise counterpart, and (2) optimizing conditional variance offers *a reference-free way* to minimize both transition and reconstruction errors. Based on these insights, we propose an entropy-aware variance optimized method for the generative process of DMs, called *EVODiff*, which systematically reduces uncertainty by optimizing conditional entropy during denoising. Extensive experiments on DMs validate our insights and demonstrate that our method significantly and consistently outperforms state-of-the-art (SOTA) gradient-based solvers. For example, compared to the DPM-Solver++, EVODiff reduces the reconstruction error by up to *45.5%* (FID improves from 5.10 to 2.78) at 10 function evaluations (NFE) on CIFAR-10, cuts the NFE cost by *25%* (from 20 to 15 NFE) for high-quality samples on ImageNet-256, and improves text-to-image generation while reducing artifacts. Code is available at https://github.com/ShiguiLi/EVODiff.

## 1 Introduction

Diffusion models (DMs) [1–3] have emerged as powerful generative models, achieving success in tasks such as image synthesis and editing [4, 5], text-to-image generation [6], voice synthesis [7], and video generation [8]. DMs generate high-fidelity samples by simulating a denoising process that iteratively refines noisy inputs through diffusion inference guided by a trained model. The model is trained via a forward process that corrupts data with Gaussian noise across multiple scales.

Despite their impressive performance, DMs face the challenge of a slow refinement process and a discrepancy between training and inference [2, 9, 10]. To address this, training-free inference methods

---

*Corresponding author.

39th Conference on Neural Information Processing Systems (NeurIPS 2025).

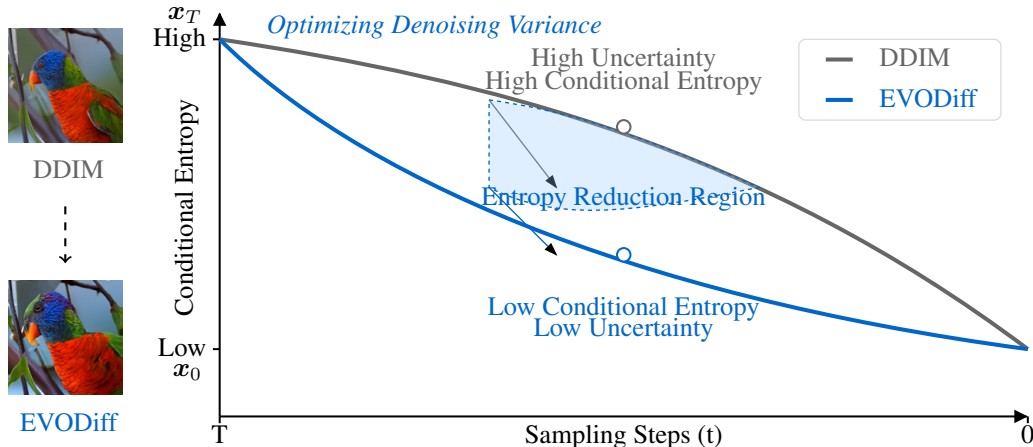

Figure 1: Illustration of conditional entropy reduction during diffusion model inference. Our EVODiff (blue) achieves lower conditional entropy in reverse transitions compared to DDIM (gray).

reformulate the denoising process as the solution to an ODE using numerical techniques. Such examples include PNDM [11], EDM [12], DPM-Solver [13], DEIS [14], SciRE-Solver [15], UniPC [16], and DPM-Solver-v3 [17]. Despite their empirical success, these ODE-focused methods lack an information-theoretic foundation. A central limitation is their neglect of information transmission efficiency. This theoretical gap suggests that the principles governing inference remain underexplored.

Our work addresses this gap by developing an information-theoretic framework for diffusion inference centered on conditional entropy dynamics. In this view, the forward diffusion process systematically increases conditional entropy as noise is added, while the reverse process seeks to recover lost information through denoising. Unlike gradient-based ODE solvers that primarily focus on numerical approximation, our framework reveals that effective denoising fundamentally operates by reducing conditional entropy during reverse transitions, a principle largely overlooked by existing methods. This insight not only guides algorithm design, but also offers a unified theoretical explanation of the varying inference efficiency across successful strategies, rooted in their entropy reduction efficiency.

Building on this framework, we propose *EVODiff*, an entropy-aware method that reduces conditional entropy by optimizing the conditional variance of each denoising iteration. Our approach provides three key technical advantages: (1) it enhances information transmission between successive denoising steps through entropy-reduction optimization; (2) it accelerates convergence by steering samples towards high-probability regions of the data distribution, drawing on principles from statistical physics [18, 19]; (3) it minimizes both transition errors and reconstruction errors through reference-free variance optimization (detailed in Section 3). These advantages lead to our main contributions:

- We introduce an information-theoretic framework for diffusion inference, demonstrating that gradient-based methods enhance inference by reducing conditional entropy. Our analysis provides the *first theoretical evidence* for why data prediction parameterization outperforms its noise counterpart, theoretically grounding previous empirical findings [13, 20].

- Guided by our insights, we propose *EVODiff*, an entropy-aware variance optimized diffusion inference method. Fundamentally, it differs from existing ODE-based approaches by directly targeting information recovery, not just approximating an ODE trajectory. This approach reduces both transition and reconstruction errors via principled variance optimization.

- Extensive experiments validate our insights and demonstrate significant improvements in inference. EVODiff reduces FID by *45.5%* on CIFAR-10 (from 5.10 to 2.78 at 10 NFE) and *43.4%* on LSUN-Bedrooms (from 13.97 to 7.91 at 5 NFE) over strong solvers like DPM-Solver++ and UniPC, while also mitigating visual artifacts in text-to-image generation.

## 2 Background

Let $d$ denote the dimension of the data. The forward process of DMs defines a Markov sequence $\{\boldsymbol{x}_t, t \in [0, T]\}$, where $\boldsymbol{x}_0 \in \mathbb{R}^d$ is the starting state drawn from the data distribution $q(\boldsymbol{x}_0)$ [1, 2]. This sequence is pushed forward with the transition kernel: $q\left(\boldsymbol{x}_t \mid \boldsymbol{x}_0\right) = \mathcal{N}(\boldsymbol{x}_t; \alpha_t \boldsymbol{x}_0, \sigma_t^2 \boldsymbol{I})$, where $\alpha_t$ and $\sigma_t$ are the noise schedules and $\alpha_t^2/\sigma_t^2$ is the signal-to-noise ratio (SNR). This transition kernel

| | DDIM | DPM-Solver | UniPC | DPM-Solver-v3 | **EVODiff** (Ours) |
|---|---|---|---|---|---|
| Gradient-based | ✗ | ✓ | ✓ | ✓ | ✓ |
| Bias term (need $\tilde{x}_0$) | ✗ | ✗ | ✗ | ✓ | ✗ |
| Variance term | ✓ | ✓ | ✓ | ✓ | ✓ |
| Entropy-aware | ✗ | ✗ | ✗ | ✗ | ✓ |

Table 1: Strategies employed for optimizing reconstruction error in different methods.

is reformulated as the stochastic differential equation (SDE) [3]: $\mathrm{d}\boldsymbol{x}_t = f(t)\boldsymbol{x}_t\,\mathrm{d}t + g(t)\mathrm{d}\boldsymbol{\omega}_t$, $\boldsymbol{x}_0 \sim q(\boldsymbol{x}_0)$, where $\boldsymbol{\omega}_t$ denotes a Wiener process, $f(t) := \frac{\mathrm{d}\log\alpha_t}{\mathrm{d}t}$, $g^2(t) := \frac{\mathrm{d}\sigma_t^2}{\mathrm{d}t} - 2\frac{\mathrm{d}\log\alpha_t}{\mathrm{d}t}\sigma_t^2$ [21]. In the denoising inference process, the reverse-time SDE of the forward diffusion process takes the form:

$$\mathrm{d}\boldsymbol{x}_t = \left[f(t)\boldsymbol{x}_t - g^2(t)\nabla_{\boldsymbol{x}}\log q(\boldsymbol{x}_t)\right]\mathrm{d}t + g(t)\mathrm{d}\overline{\boldsymbol{\omega}}_t, \tag{1}$$

where $\overline{\boldsymbol{\omega}}_t$ represents a Wiener process. The inference generative process based on diffusion (or probability flow) ordinary differential equation (ODE) [3] is governed by $\mathrm{d}\boldsymbol{x}_t = \left(f(t)\boldsymbol{x}_t - \frac{1}{2}g^2(t)\nabla_{\boldsymbol{x}}\log q(\boldsymbol{x}_t)\right)\mathrm{d}t$, where the marginal distribution $q(\boldsymbol{x}_t)$ of $\boldsymbol{x}_t$ is equivalent to that of $\boldsymbol{x}_t$ in the SDE of Eq. (1). The model is generally trained by minimizing the mean squared error (MSE) [2]:

$$\mathbb{E}_{\boldsymbol{x}_0,\,\boldsymbol{\epsilon},\,t}[w(t)\,\|\boldsymbol{\epsilon}_\theta(\alpha_t\boldsymbol{x}_0 + \sigma_t\boldsymbol{\epsilon}, t) - \boldsymbol{\epsilon}\|_2^2], \tag{2}$$

where $\alpha_t^2 + \sigma_t^2 = 1$, $\boldsymbol{x}_0 \sim q(\boldsymbol{x}_0)$, $\boldsymbol{\epsilon} \sim \mathcal{N}(\boldsymbol{0}, \boldsymbol{I})$, $t \sim \mathcal{U}(0, T)$, and $w(t)$ is a weight function w.r.t. $t$.

**Diffusion ODE.** Based on the relationship of $\boldsymbol{\epsilon}_\theta(\boldsymbol{x}_t, t) = -\sigma_t\nabla_{\boldsymbol{x}}\log q(\boldsymbol{x}_t)$ [3], samples can be generated by the diffusion inference process from $T$ to 0 defined diffusion ODE:

$$\frac{\mathrm{d}\boldsymbol{x}_t}{\mathrm{d}t} = f(t)\boldsymbol{x}_t + \frac{g^2(t)}{2\sigma_t}\boldsymbol{\epsilon}_\theta(\boldsymbol{x}_t, t), \quad \boldsymbol{x}_T \sim \mathcal{N}(\boldsymbol{0}, \hat{\sigma}^2\boldsymbol{I}). \tag{3}$$

By $\boldsymbol{x}_\theta(\boldsymbol{x}_t, t) := (\boldsymbol{x}_t - \sigma_t\boldsymbol{\epsilon}_\theta(\boldsymbol{x}_t, t))/\alpha_t$ [21], the data prediction ODE can be expressed as follows

$$\frac{\mathrm{d}\boldsymbol{x}_t}{\mathrm{d}t} = \left(f(t) + \frac{g^2(t)}{2\sigma_t^2}\right)\boldsymbol{x}_t - \alpha_t\frac{g^2(t)}{2\sigma_t^2}\boldsymbol{x}_\theta(\boldsymbol{x}_t, t). \tag{4}$$

*Remark* 2.1. A unified solution formula for both ODE formulations in (3) and (4) can be expressed as

$$\boldsymbol{f}(\boldsymbol{x}_t) - \boldsymbol{f}(\boldsymbol{x}_s) = \int_{\kappa(s)}^{\kappa(t)} \boldsymbol{d}_\theta(\boldsymbol{x}_{\psi(\tau)}, \psi(\tau))\,\mathrm{d}\tau. \tag{5}$$

where $\psi(\kappa(t)) = t$. *When using noise prediction*, we have $\boldsymbol{d}_\theta = \boldsymbol{\epsilon}_\theta$, $\boldsymbol{f}(\boldsymbol{x}_t) := \frac{\boldsymbol{x}_t}{\alpha_t}$ and $\kappa(t) := \frac{\sigma_t}{\alpha_t}$; *when using data prediction*, we have $\boldsymbol{d}_\theta = \boldsymbol{x}_\theta$, $\boldsymbol{f}(\boldsymbol{x}_t) := \frac{\boldsymbol{x}_t}{\sigma_t}$ and $\kappa(t) := \frac{\alpha_t}{\sigma_t}$ [22, 15].

**Gradient-based Inference.** Denote $h_{t_i} := \kappa(t_{i-1}) - \kappa(t_i)$ and $\boldsymbol{\iota}(\boldsymbol{x}_{t_{i-1}}) := \boldsymbol{f}(\boldsymbol{x}_{t_{i-1}}) - \boldsymbol{f}(\boldsymbol{x}_{t_i})$. By substituting the Taylor expansion of $\boldsymbol{d}_\theta(\boldsymbol{x}_{t_{i-1}}, t_{i-1})$ at $\tau_{t_i}$ into Eq. (5), we can obtain $\boldsymbol{\iota}(\boldsymbol{x}_{t_{i-1}}) = \sum_{k=0}^{n} \frac{h_{t_i}^{k+1}}{(k+1)!}\boldsymbol{d}_\theta^{(k)}(\boldsymbol{x}_{t_i}, t_i) + \mathcal{O}(h_{t_i}^{n+2})$, where $\boldsymbol{d}_\theta^{(k)}(\boldsymbol{x}_{\psi(\tau)}, \psi(\tau)) := \frac{\mathrm{d}^k\boldsymbol{d}_\theta(\boldsymbol{x}_{\psi(\tau)}, \psi(\tau))}{\mathrm{d}\tau^k}$ as $k$-th order total derivative of $\boldsymbol{d}_\theta(\boldsymbol{x}_{\psi(\tau)}, \psi(\tau))$ w.r.t. $\tau$. When $n = 1$, this iteration is equivalent to *DDIM* [9]:

$$\boldsymbol{f}(\tilde{\boldsymbol{x}}_{t_{i-1}}) = \boldsymbol{f}(\tilde{\boldsymbol{x}}_{t_i}) + h_{t_i}\boldsymbol{d}_\theta(\tilde{\boldsymbol{x}}_{t_i}, t_i). \tag{6}$$

where $\tilde{\boldsymbol{x}}_t$ depends on the type of $\boldsymbol{d}_\theta$. For gradient-based inference, a widely used technique is the finite difference (FD) method [23] as follows: $\boldsymbol{d}_\theta^{(k+1)}(\boldsymbol{x}_t, t) = \frac{1}{\hat{h}_t}\left(\boldsymbol{d}_\theta^{(k)}(\boldsymbol{x}_l, l) - \boldsymbol{d}_\theta^{(k)}(\boldsymbol{x}_t, t)\right) + \mathcal{O}(\hat{h}_t)$. Then, a gradient-based diffusion inference can be derived by using the FD method as follows:

$$\boldsymbol{f}(\tilde{\boldsymbol{x}}_{t_{i-1}}) = \boldsymbol{f}(\tilde{\boldsymbol{x}}_{t_i}) + h_{t_i}\boldsymbol{d}_\theta(\tilde{\boldsymbol{x}}_{t_i}, t_i) + \frac{1}{2}h_{t_i}^2 F_\theta(s_i, t_i), \tag{7}$$

where $F_\theta(s_i, t_i) := (\boldsymbol{d}_\theta(\tilde{\boldsymbol{x}}_{s_i}, s_i) - \boldsymbol{d}_\theta(\tilde{\boldsymbol{x}}_{t_i}, t_i))/\hat{h}_{t_i}$ denotes the forward FD, $\hat{h}_{t_i} := \kappa(s_i) - \kappa(t_i)$.

## 3 Conditional Entropy Reduction in Diffusion Inference

**Conditional Entropy in Information Transfer.** During DM inference, each iteration reduces uncertainty in intermediate representations via structured denoising. From an information-theoretic view, the information gain between successive states is quantified by the *mutual information* [24]:

$$I_p(\boldsymbol{x}_{t_i}; \boldsymbol{x}_{t_{i+1}}) = H_p(\boldsymbol{x}_{t_i}) - H_p(\boldsymbol{x}_{t_i}|\boldsymbol{x}_{t_{i+1}}), \tag{8}$$

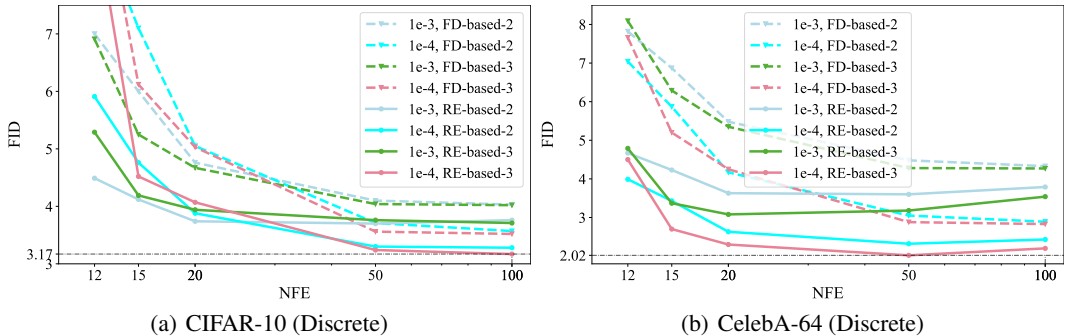

(a) CIFAR-10 (Discrete)  (b) CelebA-64 (Discrete)

Figure 2: Quantitative results of FID $\downarrow$ show that efficient entropy reduction (RE) method consistently improves image quality compared to FD-based method in Eq.(7) across various ablation scenarios.

where $H_p(\boldsymbol{x}_{t_i})$ is the entropy of state $\boldsymbol{x}_{t_i}$ and $H_p(\boldsymbol{x}_{t_i}|\boldsymbol{x}_{t_{i+1}})$ is the *conditional entropy* of $\boldsymbol{x}_{t_i}$ given $\boldsymbol{x}_{t_{i+1}}$. A lower $H_p(\boldsymbol{x}_{t_i}|\boldsymbol{x}_{t_{i+1}})$ results in a higher $I_p(\boldsymbol{x}_{t_i};\boldsymbol{x}_{t_{i+1}})$, which suggests that a well-designed method effectively utilizes the information from $\boldsymbol{x}_{t_{i+1}}$ to refine the estimate of $\boldsymbol{x}_{t_i}$.

**Conditional Variance and Conditional Entropy.** We denote $p(\boldsymbol{x}_{t_i}|\boldsymbol{x}_{t_{i+1}},\boldsymbol{x}_0)$ as $p(\boldsymbol{x}_{t_i}|\boldsymbol{x}_{t_{i+1}})$, and $\mathrm{Var}(\boldsymbol{x}_{t_i}|\boldsymbol{x}_{t_{i+1}})$ as the conditional variance $\mathrm{Var}(\boldsymbol{x}_{t_i}|\boldsymbol{x}_{t_{i+1}},\boldsymbol{x}_0)$ for brevity. In DMs [2, 3], the reverse transition $p(\boldsymbol{x}_{t_i}|\boldsymbol{x}_{t_{i+1}},\boldsymbol{x}_0)$ is commonly approximated as a Gaussian distribution under the *Markov assumption*, i.e., $p(\boldsymbol{x}_{t_i}|\boldsymbol{x}_{t_{i+1}}) \approx \mathcal{N}(\boldsymbol{\mu}_{t_i},\boldsymbol{\Sigma}_{t_i})$, simplifying both model training and theoretical analysis [21, 25, 26, 12]. Accordingly, the conditional entropy $H_p(\boldsymbol{x}_{t_i}|\boldsymbol{x}_{t_{i+1}})$ simplifies to:

$$H_p(\boldsymbol{x}_{t_i}|\boldsymbol{x}_{t_{i+1}}) \approx C + 1/2 \cdot \log \det(\mathrm{Var}(\boldsymbol{x}_{t_i}|\boldsymbol{x}_{t_{i+1}})), \tag{9}$$

where $C = 1/2 \cdot d(\log 2\pi + 1)$. Thus, $H_p(\boldsymbol{x}_{t_i}|\boldsymbol{x}_{t_{i+1}})$ is intrinsically tied to its conditional variance:

$$H_p(\boldsymbol{x}_{t_i}|\boldsymbol{x}_{t_{i+1}}) \propto \log \det(\mathrm{Var}(\boldsymbol{x}_{t_i}|\boldsymbol{x}_{t_{i+1}})). \tag{10}$$

This indicates that minimizing conditional variance directly reduces conditional entropy.

**Reconstruction Error and Conditional Variance.** In the forward process, $q(\boldsymbol{x}_t)$ approaches a standard Gaussian as $t$ increases, but $q(\boldsymbol{x}_t|\boldsymbol{x}_0)$ remains structured around scaled versions of $\boldsymbol{x}_0$. Since $H_q(\boldsymbol{x}_t|\boldsymbol{x}_0) \leq H_q(\boldsymbol{x}_t)$, information about $\boldsymbol{x}_0$ persists in the modeled $\boldsymbol{x}_t$, and the inference process seeks to recover it. Let $\boldsymbol{\mu}_{t_i|t_{i+1}} = \mathbb{E}_q[\boldsymbol{x}_{t_i}|\boldsymbol{x}_{t_{i+1}}]$. The MSE between the inference states and its posterior mean is $\mathbb{E}_q[\|\boldsymbol{x}_{t_i} - \boldsymbol{\mu}_{t_i|t_{i+1}}\|^2] = \mathrm{Tr}(\mathrm{Var}_q(\boldsymbol{x}_{t_i}|\boldsymbol{x}_{t_{i+1}}))$. Leveraging the connection with conditional variance, under the isotropic assumption commonly used in DMs, we obtain:

$$\min H_q(\boldsymbol{x}_{t_i}|\boldsymbol{x}_{t_{i+1}}) \Leftrightarrow \min \mathbb{E}_q[\|\boldsymbol{x}_{t_i} - \boldsymbol{\mu}_{t_i|t_{i+1}}\|^2]. \tag{11}$$

We now decompose the reconstruction error between $\boldsymbol{x}_{t_i}$ and $\boldsymbol{x}_0$ using the law of total expectation.

**Proposition 3.1.** *Note that $\boldsymbol{x}_{t_i} - \boldsymbol{x}_0 = (\boldsymbol{x}_{t_i} - \boldsymbol{\mu}_{t_i|t_{i+1}}) + (\boldsymbol{\mu}_{t_i|t_{i+1}} - \boldsymbol{x}_0)$, we have*

$$\mathbb{E}_q\left[\|\boldsymbol{x}_{t_i} - \boldsymbol{x}_0\|^2\right] = \underbrace{\mathbb{E}_q\left[\|\boldsymbol{x}_{t_i} - \boldsymbol{\mu}_{t_i|t_{i+1}}\|^2\right]}_{\textit{Variance term}} + \underbrace{\mathbb{E}_q\left[\|\boldsymbol{\mu}_{t_i|t_{i+1}} - \boldsymbol{x}_0\|^2\right]}_{\textit{Bias term}}. \tag{12}$$

*where details of this reconstruction error decomposition are provided in Appendix C.1.*

This decomposition reveals two distinct inference approaches for reducing the reconstruction error: *(1) minimizing the conditional variance term directly; and (2) optimizing the bias error term with the prior $\boldsymbol{x}_0$, which is often approximated by deterministic DM samplers.* Although the total reconstruction error includes both the **variance** and **bias** terms, *optimizing conditional variance becomes the main actionable mechanism, since we do not often have access to the desired $\boldsymbol{x}_0$ during inference.* Thus, optimizing conditional variance is central to inference; various methods are summarized in Table 1.

**Variance-Driven Conditional Entropy Analysis.** We demonstrate how entropy reduction effectively steers samples toward the target distribution, supported by both theory and empirical evidence. Theoretically, DM denoising functions as an entropy reduction mechanism grounded in Langevin dynamics [19] and non-equilibrium thermodynamics [18]. This principle dictates that more efficient entropy reduction will accelerate convergence by steering samples toward high-probability regions of the target distribution. Figure 1 visually illustrates this, showing the trajectories of DDIM versus our gradient-based inference. Further empirical evidence is presented in Figure 2, which details an

**Algorithm 1** EVODiff: Optimizing Denoising Variance of Diffusion Model Inference.

**Require:** initial $\boldsymbol{x}_T$, time schedule $\{t_i\}_{i=0}^N$, model $\boldsymbol{x}_\theta$.
1: $\boldsymbol{x}_{t_N} \leftarrow \boldsymbol{x}_T$, $h_{t_i} := \kappa(t_{i-1}) - \kappa(t_i)$, $r_{\text{logSNR}}(i) := \frac{\log \kappa(t_i) - \log \kappa(t_{i+1})}{\log \kappa(t_{i-1}) - \log \kappa(t_i)}$.
2: Denote $\boldsymbol{g}(\boldsymbol{x}_{t_i}) := \frac{\sigma_{t_{i-1}}}{\sigma_{t_i}} \boldsymbol{x}_{t_i} + \sigma_{t_{i-1}} h_{t_i} \boldsymbol{x}_\theta(\boldsymbol{x}_{t_i}, t_i)$. # Euler's or DDIM's iteration.
3: **for** $i \leftarrow N$ to $1$ **do**
4: $\quad \boldsymbol{x}_{t_i} \leftarrow \boldsymbol{g}(\boldsymbol{x}_{t_{i+1}})$.
5: $\quad \boldsymbol{x}_{t_{i-1}} \leftarrow \boldsymbol{g}(\boldsymbol{x}_{t_i}) + \sigma_{t_{i-1}} \frac{h_{t_i}^2}{2} B_\theta(t_i, l_i)$.
6: $\quad B_\theta(t_i) \leftarrow \left(1 - \frac{\eta_i}{2}\right) B_\theta(s_i, t_i) + \frac{\eta_i}{2} B_\theta(t_i, l_i)$, where $\eta_i$ is refined by Eq. (25).
7: $\quad \boldsymbol{x}_{t_{i-1}} \leftarrow \boldsymbol{g}(\boldsymbol{x}_{t_i}) + \sigma_{t_{i-1}} \frac{h_{t_i}^2}{2\zeta_i} B_\theta(t_i)$, where $\zeta_i$ is refined by Eq. (25).
8: **end for**
**Ensure:** $\boldsymbol{x}_0$.

ablation study on CIFAR-10 and CelebA-64 comparing our entropy reduction-focused (RE) method with traditional FD-based gradients.

Our analysis reveals that gradient-based methods excel at driving entropy reduction by optimizing conditional variance, which efficiently guides noisy states toward the desired distribution. For theoretical tractability, we assume independence between estimated noise at different timesteps, in line with DDPM's training objective of independent MSE minimization. While neural network parameter-sharing during training could introduce dependencies, prior works like [2, 9] justify this surrogate by showing that these dependencies have a negligible performance impact.

We identify two sources of conditional entropy in the reverse transition of DMs: the inference path uncertainty (e.g., from ODE/SDE solvers) and the model's own intrinsic uncertainty. *While simple ODE-solver paths address the former, they may not effectively reduce the entropy contributed by the model. We therefore propose an approach that moves beyond these simple paths to target the total conditional entropy.* From this perspective, we first derive Proposition 3.2 (Proof in Appendix C.2).

**Proposition 3.2.** *The gradient-based inference in Eq. (7) can reduce conditional entropy more efficiently than the first-order inference in Eq. (6) when* $\frac{h_{t_i}}{\tilde{h}_{t_i}} \in \left[1, \frac{4 \cdot \text{Var}(\boldsymbol{\epsilon}_\theta(\tilde{\boldsymbol{x}}_{t_i}, t_i))}{\text{Var}(\boldsymbol{\epsilon}_\theta(\tilde{\boldsymbol{x}}_{s_i}, s_i)) + \text{Var}(\boldsymbol{\epsilon}_\theta(\tilde{\boldsymbol{x}}_{t_i}, t_i))}\right]$.

This shows that gradient-based inference can achieve larger reductions in uncertainty when the step size ratio is appropriately chosen. A practical interval for Proposition 3.2 is discussed in Remark C.1.

Furthermore, we identify that solvers like DPM-Solver [13] and the Heun iterations in EDM [12] can be understood through conditional entropy reduction, with details in the Appendix C.3.

**Proposition 3.3.** *The acceleration mechanisms of DPM-Solver and the Heun iterations in EDM can be unified and explained as specific implementations of the conditional entropy reduction framework.*

Finally, we theoretically establish why denoising iterations using data prediction perform better than those using noise prediction. The proof is provided in Appendix C.5.

**Theorem 3.4.** *Data prediction parameterization reduces reconstruction errors more effectively than its noise counterpart. Under independence assumptions, it also reduces conditional entropy.*

In summary, we provide a variance-driven conditional entropy analysis for diffusion inference, which theoretically explains the superior performance of gradient-based inference and data prediction parameterization. In particular, Theorem 3.4 demonstrates that data parameterization, by directly targeting the data distribution, avoids the error-prone chain of $\boldsymbol{\epsilon}_t \mapsto \boldsymbol{x}_t \mapsto \boldsymbol{x}_0$.

# 4 Optimizing Diffusion Model Inference via Entropy-aware Variance Control

## 4.1 Denoising Variance Analysis for Gradient-based Inference

Our focus is on multi-step iterations using data parameterization, which has shown superiority through the theoretical result in Theorem 3.4 and prior empirical evidence from [13, 20].

| NFE=5 | NFE=8 | NFE=10 | NFE=12 |
|---|---|---|---|

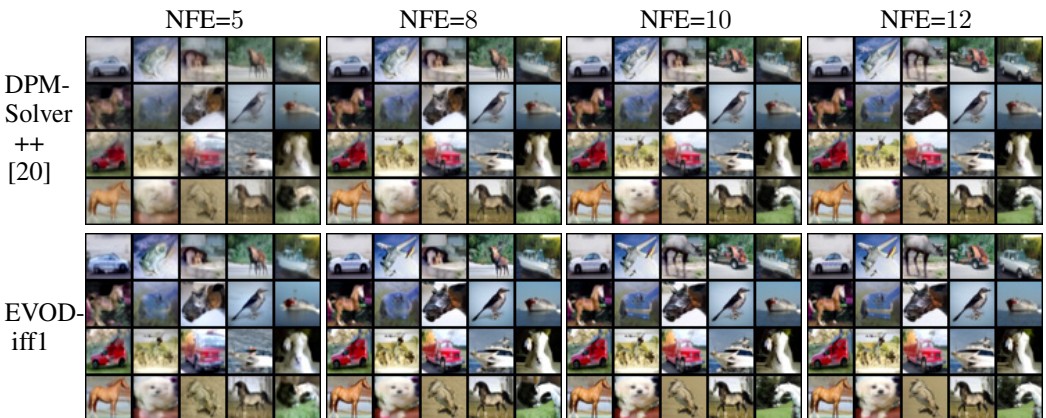

DPM-Solver++ [20]

EVOD-iff1

Figure 3: Sample comparison of our method vs. baseline using the pre-trained EDM on CIFAR-10.

Note that $\boldsymbol{f}(\boldsymbol{x}_t) = \frac{\boldsymbol{x}_t}{\sigma_t}$ for the ODE for data prediction defined in Eq. (5), $\iota(\boldsymbol{x}_{t_{i-1}}) = \frac{\boldsymbol{x}_{t_{i-1}}}{\sigma_{t_{i-1}}} - \frac{\boldsymbol{x}_{t_i}}{\sigma_{t_i}}$.
Formally, the multi-step iteration can be written as:

$$\frac{\boldsymbol{x}_{t_{i-1}}}{\sigma_{t_{i-1}}} - \frac{\boldsymbol{x}_{t_i}}{\sigma_{t_i}} = h_{t_i}\boldsymbol{x}_\theta\left(\boldsymbol{x}_{t_i}, t_i\right) + \frac{1}{2}h_{t_i}^2 B_\theta(t_i, t_{i+1}), \tag{13}$$

where $B_\theta(t_i, t_{i+1}) := (\boldsymbol{x}_\theta(\boldsymbol{x}_{t_i}, t_i) - \boldsymbol{x}_\theta(\boldsymbol{x}_{t_{i+1}}, t_{i+1}))/h_{t_{i+1}}$ denotes the backward FD. In this iteration, the smaller step size $|h_{t_i}|$ compared to $|h_{t_i} - h_{t_{i+1}}|$ in the single-step case (Appendix C.6) reduces the conditional entropy, offering greater potential to improve the denoising process.

Denote $\bar{\zeta}_i = (1 - \zeta_i)$. A straightforward improvement for Eq. (13) can be formulated as follows:

$$\frac{\boldsymbol{x}_{t_{i-1}}}{\sigma_{t_{i-1}}} - \frac{\boldsymbol{x}_{t_i}}{\sigma_{t_i}} = h_{t_i}\boldsymbol{x}_\theta\left(\boldsymbol{x}_{t_i}, t_i\right) + \frac{1}{2}h_{t_i}^2 B_\theta(t_i, l_i), \tag{14}$$

where $\boldsymbol{x}_\theta(\boldsymbol{x}_{l_i}, l_i) := \zeta_i\boldsymbol{x}_\theta(\boldsymbol{x}_{t_i}, t_i) + \bar{\zeta}_i\boldsymbol{x}_\theta(\boldsymbol{x}_{t_{i+1}}, t_{i+1})$ represents a linear interpolation. Similarly, the implicit approach is: $\frac{\boldsymbol{x}_{t_{i-1}}}{\sigma_{t_{i-1}}} - \frac{\boldsymbol{x}_{t_i}}{\sigma_{t_i}} = h_{t_i}\boldsymbol{x}_\theta(\boldsymbol{x}_{t_i}, t_i) + \frac{1}{2}h_{t_i}^2 B_\theta(s_i, t_i)$, where $\boldsymbol{x}_\theta(\boldsymbol{x}_{s_i}, s_i) := \zeta_i\boldsymbol{x}_\theta(\boldsymbol{x}_{t_{i-1}}, t_{i-1}) + \bar{\zeta}_i\boldsymbol{x}_\theta(\boldsymbol{x}_{t_i}, t_i)$. Note that these two improvement approaches can be unified as

$$\frac{\boldsymbol{x}_{t_{i-1}}}{\sigma_{t_{i-1}}} - \frac{\boldsymbol{x}_{t_i}}{\sigma_{t_i}} = h_{t_i}\boldsymbol{x}_\theta\left(\boldsymbol{x}_{t_i}, t_i\right) + \frac{1}{2}h_{t_i}^2 \zeta_i \bar{B}_\theta(t_i; u_i), \tag{15}$$

where $\bar{B}_\theta(t_i; u_i) = B_\theta(s_i, t_i)$ when $u_i = s_i$, and $\bar{B}_\theta(t_i; u_i) = B_\theta(t_i, l_i)$ when $u_i = l_i$.

*Remark* 4.1. The RE-based multi-step iterations in Eq. (15) reduce the conditional entropy of iteration in Eq. (13) by leveraging model parameters from low-variance regions.

We provide the convergence guarantees for the RE-based multi-step iteration described in Eq. (15).

**Theorem 4.2.** *If $\boldsymbol{x}_\theta\left(\boldsymbol{x}_t, t\right)$ satisfies Assumption D.1, the RE-based multi-step iteration constitutes a globally convergent second-order iterative algorithm. The proof is provided in Appendix D.2.*

However, a key question arises: how should $\zeta_i$ and $\hat{h}_{t_i}$ be determined? As $\boldsymbol{x}_\theta\left(\boldsymbol{x}_{t_i}, t_i\right)$ is predict the clean data $\boldsymbol{x}_0$ from the noisy data $\boldsymbol{x}_{t_i}$, we present a remark for ideal cases.
*Remark* 4.3. If $\mathrm{Var}\left(\boldsymbol{x}_\theta\left(\boldsymbol{x}_{t_i}, t_i\right)\right) \propto \sigma_{t_i}^2$, the variance minimization can be achieved by setting $\zeta_i = \sigma_{t_{i-1}}^2/(\sigma_{t_i}^2 + \sigma_{t_{i-1}}^2)$ for $\boldsymbol{x}_\theta\left(\boldsymbol{x}_{s_i}, s_i\right)$ and $\zeta_i = \sigma_{t_i}^2/(\sigma_{t_i}^2 + \sigma_{t_{i+1}}^2)$ for $\boldsymbol{x}_\theta\left(\boldsymbol{x}_{l_i}, l_i\right)$.

Furthermore, we can improve the iteration of Eq. (13) by incorporating $B_\theta(t_i, s_i)$ and $B_\theta(s_i, t_i)$:

$$\frac{\boldsymbol{x}_{t_{i-1}}}{\sigma_{t_{i-1}}} - \frac{\boldsymbol{x}_{t_i}}{\sigma_{t_i}} = h_{t_i}\boldsymbol{x}_\theta\left(\boldsymbol{x}_{t_i}, t_i\right) + \frac{1}{2}h_{t_i}^2\left((1 - \eta_i)B_\theta(s_i, t_i) + \eta_i B_\theta(t_i, l_i)\right), \tag{16}$$

where $\eta_i$ determines the gradient term variance. Therefore, from the lens of conditional entropy reduction, we can establish an optimization objective to directly minimize this variance.

## 4.2 Optimizing Denoising Variance with Evolution State Differences

We observe that the conditional variance in gradient-based iterations can be composed of two critical components: the variance of the gradient estimation term itself and the variance between the gradient

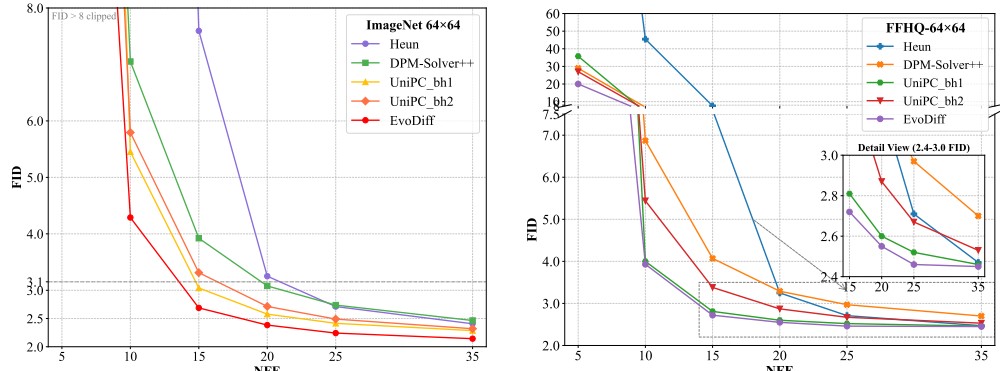

Figure 4: FID ↓ scores for gradient-based inference methods on ImageNet-64 and FFHQ-64.

term and the first-order term. Specifically, the unified iteration in Eq. (15) can be rewritten as:

$$\frac{\boldsymbol{x}_{t_{i-1}}}{\sigma_{t_{i-1}}} - \frac{\boldsymbol{x}_{t_i}}{\sigma_{t_i}} = h_{t_i} \left( \left( 1 \mp \frac{\zeta_i}{2} \frac{h_{t_i}}{h_{\mu_i}} \right) \boldsymbol{x}_\theta \left( \boldsymbol{x}_{t_i}, t_i \right) \pm \frac{\zeta_i}{2} \frac{h_{t_i}}{h_{\mu_i}} \boldsymbol{x}_\theta \left( \boldsymbol{x}_{\mu_i}, \mu_i \right) \right). \tag{17}$$

As $\boldsymbol{x}_\theta \left( \boldsymbol{x}_{\mu_i}, \mu_i \right)$ and $\boldsymbol{x}_\theta \left( \boldsymbol{x}_{t_i}, t_i \right)$ are known in multi-step iterative mechanisms, the $\mathrm{Var}(\boldsymbol{x}_{t_{i-1}} | \boldsymbol{x}_{t_i})$ is controlled by the value of $\zeta_i$. Inspired by Eq. (17), we observe that $\zeta_i$ balances the variance between the gradient term and the first-order term. By harmonizing their statistical characteristics, we can efficiently reduce the conditional variance. To achieve this, we define $G(\zeta_i) = (1 - \zeta_i) \boldsymbol{x}_\theta \left( \boldsymbol{x}_{t_i}, t_i \right) + \zeta_i \boldsymbol{x}_\theta \left( \boldsymbol{x}_{\mu_i}, \mu_i \right)$ to balance the variance between two terms. Moreover, the variance of the gradient term itself should be balanced by $\eta_i$, as we should optimize $\eta_i$ in Eq. (16).

**Refining $\zeta_i$ with Evolution State Differences.** Our goal is to refine $\zeta_i$ using the *available information at current step*. As discussed above, we can optimize $\zeta_i$ to control the conditional variance by formulating an objective involving $G(\zeta_i)$. On the one hand, we can rewrite the iteration in Eq. (17) as

$$\hat{\boldsymbol{x}}_{1,t_i} = \frac{\sigma_{t_i}}{\sigma_{t_{i-1}}} \boldsymbol{x}_{t_{i-1}} - \sigma_{t_i} h_{t_i} G(\zeta_i). \tag{18}$$

Notice that $\tilde{\boldsymbol{x}}_{t_i}$ in Eq. (18) is determined by $\zeta_i$. On the other hand, we can consider $\boldsymbol{x}_{t_{i-1}}$ as a starting point and perform an inverse iterative from $t_{i-1}$ to $t_i$ to approximate $\boldsymbol{x}_{t_i}$ as follows:

$$\frac{\boldsymbol{x}_{t_i}}{\sigma_{t_i}} - \frac{\boldsymbol{x}_{t_{i-1}}}{\sigma_{t_{i-1}}} = \int_{\kappa(t_{i-1})}^{\kappa(t_i)} \boldsymbol{x}_\theta \left( \boldsymbol{x}_{\psi(\tau)}, \psi(\tau) \right) \mathrm{d}\tau. \tag{19}$$

Similar to the Eq. (13), this inverse estimation of Eq. (19) is as follow:

$$\hat{\boldsymbol{x}}_{2,t_i} = \frac{\sigma_{t_i}}{\sigma_{t_{i-1}}} \boldsymbol{x}_{t_{i-1}} - \sigma_{t_i} h_{t_i} \boldsymbol{x}_\theta \left( \boldsymbol{x}_{t_{i-1}}, t_{i-1} \right) + \sigma_{t_i} \frac{1}{2} h_{t_i}^2 B_\theta(s_i, t_i). \tag{20}$$

Drawing from equations (18) and (20), we can determine $\zeta_i$ by minimizing the differences between two estimations. Then, the optimization objective for $\zeta_i$ is defined as follows:

$$\min_{\zeta_i > 0} \mathcal{L}_1(\zeta_i) := \left\| (\hat{\boldsymbol{x}}_{1,t_i} - \boldsymbol{x}_{t_i}) + (\hat{\boldsymbol{x}}_{2,t_i} - \boldsymbol{x}_{t_i}) \right\|, \tag{21}$$

Directly solving this objective is challenging, as the optimal $\boldsymbol{x}_{t_i}$ is unknown. Fortunately, as $\mathcal{L}_1(\zeta_i) \leq \left\| \hat{\boldsymbol{x}}_{1,t_i} + \hat{\boldsymbol{x}}_{2,t_i} \right\| + \left\| 2\boldsymbol{x}_{t_i} \right\|$, we observe that $\left\| 2\boldsymbol{x}_{t_i} \right\|$ is independent of the $\zeta_i$. Then, we can use $\tilde{\boldsymbol{x}}_{t_i}$ to replace $\boldsymbol{x}_{t_i}$ as when optimizing $\mathcal{L}_1(\zeta_i)$. Denote $P(\boldsymbol{x}_{t_{i-1}}) := \hat{\boldsymbol{x}}_{2,t_i} + \frac{\sigma_{t_i}}{\sigma_{t_{i-1}}} \boldsymbol{x}_{t_{i-1}} - 2\tilde{\boldsymbol{x}}_{t_i}$ for brevity. Then, $\mathcal{L}_1(\zeta_i)$ can be rewritten as: $\mathcal{L}_1(\zeta_i) = \left\| P(\boldsymbol{x}_{t_{i-1}}) - \sigma_{t_i} h_{t_i} G(\zeta_i) \right\|$.

**Lemma 4.4.** *When the constraint on $\zeta_i$ is relaxed, $\min_{\zeta_i} \mathcal{L}_1^2(\zeta_i)$ possesses the closed-form solution:*

$$\zeta_i^* = -(vec^T(D_i) vec(\tilde{P}_i)) / (\sigma_{t_i} h_{t_i} vec^T(D_i) vec(D_i)), \tag{22}$$

*where $\tilde{P}_i := P(\boldsymbol{x}_{t_{i-1}}) - \sigma_{t_i} h_{t_i} \boldsymbol{x}_\theta \left( \tilde{\boldsymbol{x}}_{t_i}, t_i \right)$, $D_i := \boldsymbol{x}_\theta \left( \tilde{\boldsymbol{x}}_{t_{i-1}}, t_{i-1} \right) - \boldsymbol{x}_\theta \left( \tilde{\boldsymbol{x}}_{t_i}, t_i \right)$, and $vec(\cdot)$ denotes the vectorization operation. The proof is provided in Appendix D.3.*

Table 2: Quantitative results of FID ↓ and IS ↑ scores for gradient-based methods on ImageNet-256, FFHQ-64, and CIFAR-10. The results are evaluated on 10k and 50k samples for various NFEs. *The DPM-Solver++ is our baseline*. Error optimization strategies across methods are shown in Table 1.

| Model | Method/NFE | Entropy-aware? | 5 | | 8 | | 10 | | 12 | |
|---|---|---|---|---|---|---|---|---|---|---|
| | | | FID↓ | IS↑ | FID↓ | IS↑ | FID↓ | IS↑ | FID↓ | IS↑ |
| CIFAR-10 EDM, 50k logSNR-time | Heun | × | 270.75 | 1.87 | 52.84 | 6.93 | 22.82 | 8.72 | 10.74 | 9.37 |
| | DPM-Solver++ | × | 27.96 | 7.47 | 8.40 | 8.80 | 5.10 | 9.14 | 3.70 | 9.36 |
| | UniPC | × | 27.03 | 7.69 | 7.67 | 9.07 | 3.98 | 9.40 | 2.76 | 9.62 |
| | EVODiff (our) | ✓ | **17.84** | **7.89** | **3.98** | **9.37** | **2.78** | **9.64** | **2.30** | **9.80** |

| Model | Method/NFE | Entropy-aware? | 5 | | 10 | | 15 | | 20 | |
|---|---|---|---|---|---|---|---|---|---|---|
| | | | FID↓ | IS↑ | FID↓ | IS↑ | FID↓ | IS↑ | FID↓ | IS↑ |
| FFHQ-64 EDM, 50k edm-time | Heun | × | 347.09 | 2.29 | 29.92 | 3.03 | 9.95 | 3.19 | 4.58 | 3.34 |
| | DPM-Solver++ | × | 25.08 | 2.99 | 6.81 | 3.27 | 3.80 | 3.29 | 3.00 | 3.35 |
| | UniPC | × | 28.87 | **3.20** | 6.65 | 3.25 | 3.40 | 3.28 | 2.69 | 3.37 |
| | EVODiff (our) | ✓ | **19.65** | 3.18 | **5.31** | **3.32** | **3.04** | **3.35** | **2.66** | **3.38** |

| Model | Method/NFE | Reference-based? | 5 | | 10 | | 15 | | 20 | |
|---|---|---|---|---|---|---|---|---|---|---|
| | | | FID↓ | IS↑ | FID↓ | IS↑ | FID↓ | IS↑ | FID↓ | IS↑ |
| ImageNet-256 ADM, 10k uniform-time | DPM-Solver++ | × | 16.62 | 98.07 | 8.68 | 143.59 | 7.80 | 152.01 | 7.51 | 153.89 |
| | UniPC | × | 15.37 | 104.41 | 8.40 | 146.95 | 7.71 | 152.16 | 7.47 | 154.30 |
| | DPM-Solver-v3 | ✓ | 14.92 | 105.85 | 8.14 | 146.82 | 7.70 | 153.79 | 7.42 | 154.35 |
| | EVODiff (our) | × | **13.98** | **110.79** | **8.14** | **147.53** | **7.48** | **154.78** | **7.25** | **157.79** |

**Refining $\eta_i$ by Balancing Gradient Errors.** Our goal is to refine $\eta_i$ using the *available information **at current step***. Denote $\tilde{\Delta}_{t_i}^g = (1-\eta_i)B_\theta(s_i, t_i) + \eta_i B_\theta(t_i, l_i)$, $E(t_{i-1}, t_i) := B_\theta(s_i, t_i) - x_\theta^{(1)}(x_{t_i}, t_i)$ as gradient error. For balancing this errors, we formulate the following optimization objective:

$$\min_{\eta_i \in (0,1]} \mathcal{L}_2(\eta_i) := \|(1 - \eta_i)E(t_{i-1}, t_i) + \eta_i E(t_i, t_{i+1})\|. \tag{23}$$

We can rewrite $\mathcal{L}_2(\eta_i)$ as $\mathcal{L}_2(\eta_i) = \|\tilde{\Delta}_{t_i}^g - x_\theta^{(1)}(x_{t_i}, t_i)\|$. Denote $\mathcal{L}_{2s}(\eta_i) := \|\tilde{\Delta}_{t_i}^g\|$. Then, $\mathcal{L}_2(\eta_i) \leq \mathcal{L}_{2s}(\eta_i) + \|x_\theta^{(1)}(x_{t_i}, t_i)\|$, where $x_\theta^{(1)}(x_{t_i}, t_i)$ can be regarded as a specific constant term independent of the target $\eta_i$. Thus, the $\eta_i$ can be obtained by optimizing the tractable $\mathcal{L}_{2s}(\eta_i)$.

**Lemma 4.5.** *When the constraint on $\eta_i$ is relaxed, $\min_{\eta_i} \mathcal{L}_{2s}^2(\eta_i)$ possesses the closed-form solution:*

$$\eta_i^* = -(vec^T(\tilde{B}_i)vec(B_\theta(t_i, l_i)))/(vec^T(\tilde{B}_i)vec(\tilde{B}_i)), \tag{24}$$

*where $\tilde{B}_i := B_\theta(s_i, t_i) - B_\theta(t_i, l_i)$. The proof is similar to that of Lemma 4.4.*

From Lemmas 4.4 and 4.5, $\mathcal{L}_{1s}(\zeta_i)$ and $\mathcal{L}_{2s}(\eta_i)$ have closed-form solutions when the constraints are relaxed. Then, the parameters in Algorithm 1 are derived by mapping these solutions as follows:

$$\eta_i = \text{Sigmoid}(|\eta_i^*|), \ \zeta_i = \text{Sigmoid}(-(|\zeta_i^*| - \mu)), \tag{25}$$

where $\zeta_i^*$ and $\eta_i^*$ are defined in Eqs. (22) and (24), respectively, and $\mu$ is the shift parameter. This optimization-driven approach captures the state differences during iteration while avoiding the computational cost of constrained optimization problems [27]. Additional details are provided in Appendix E.7. Ablation studies on $\mu$ are provided in Table 9 of Appendix E.7.1.

Finally, we prove the global convergence of the proposed inference method for data parameterization.

**Theorem 4.6.** *The diffusion inference method in Algorithm 1 exhibits second-order global convergence with a local error of $\mathcal{O}(h_{t_i}^3)$. The proof is provided in Appendix D.4.*

## 5 Experiments

We experimentally validate our method on a diverse suite of DMs and datasets, including CIFAR-10, CelebA-64, FFHQ-64, ImageNet-64, ImageNet-256, and LSUN-Bedrooms. Our evaluation uses standard metrics such as Fréchet Inception Distance (FID) and Inception Score (IS) across a varying number of function evaluations (NFEs). We also compare our method on Stable Diffusion v1.4 and v1.5 using CLIP and Aesthetic scores. Our evaluation focuses on the data prediction parameterization on different diffusion-based generative models including pixel-space diffusion and latent-space diffusion. Additional experimental details and results are provided in Appendix E.

| NFE=25 | NFE=35 |
|---|---|

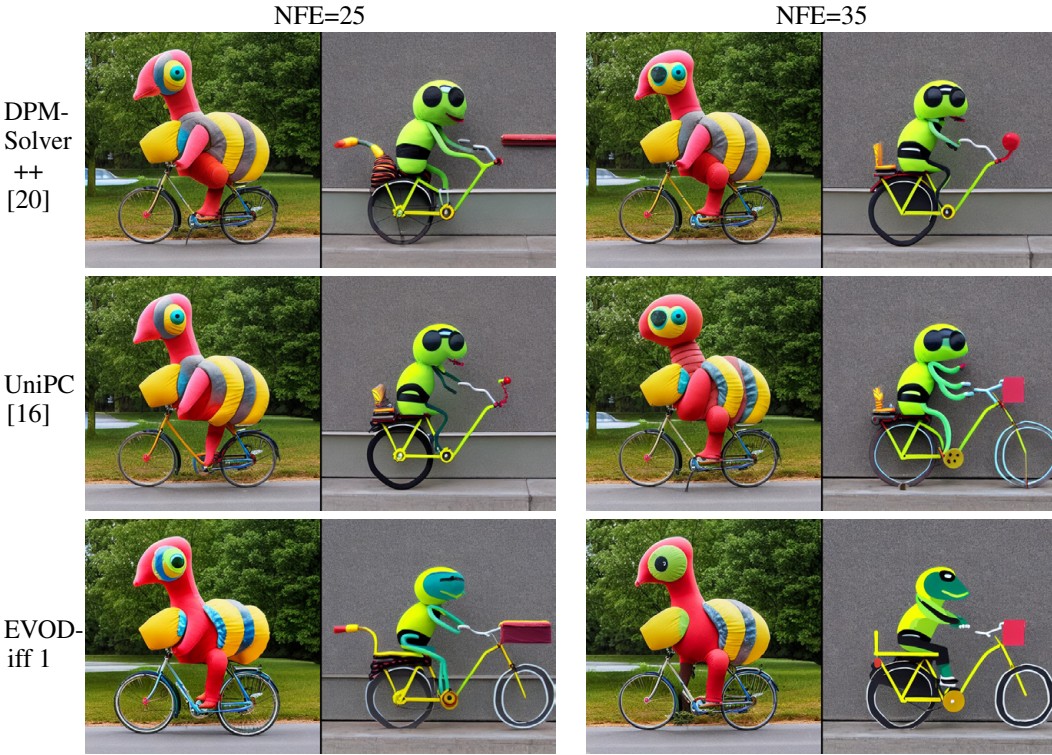

Figure 5: Random samples from the Stable-Diffusion-v1.5 model [28] with a guidance scale of 7.5, using varying NFEs and the prompt "*Giant caterpillar riding a bicycle*". Even at a low 25 NFE, EVODiff produces *high-fidelity*, *semantically correct images* while competing methods fail with severe artifacts, demonstrating the superiority of our entropy-aware variance optimized method.

Table 3: Ablation study of our variance-driven gradient-based approach applied to DMs with data parameterization on ImageNet-256 and CIFAR-10. The *DPM-Solver++ is our baseline method*.

| Method | Model | NFE | | | | | | |
|---|---|---|---|---|---|---|---|---|
| | | 5 | 6 | 8 | 10 | 12 | 15 | 20 |
| Baseline | | 27.96 | 16.87 | 8.40 | 5.10 | 3.70 | 2.83 | 2.33 |
| RE-based in Eq. (15) | CIFAR-10 | 21.39 | 12.14 | 4.81 | 2.98 | 2.44 | 2.15 | 2.08 |
| EVODiff | | **17.83** | **9.17** | **3.98** | **2.78** | **2.30** | **2.12** | **2.06** |
| Baseline | | 16.62 | 12.86 | 9.73 | 8.68 | 8.17 | 7.80 | 7.51 |
| Eq. (15) with balanced $\zeta_i$ | ImageNet-256 | 15.31 | 12.14 | 9.46 | 8.57 | 8.07 | 7.76 | 7.48 |
| Eq. (16) with refined $\zeta_i$ | | **13.80** | **10.91** | 8.91 | 8.23 | 7.89 | 7.58 | 7.36 |
| EVODiff ($r_{\text{logSNR}}$) | | 13.98 | 10.98 | **8.84** | 8.16 | 7.81 | 7.52 | 7.32 |
| EVODiff ($r_{\text{refined}}$) | | 14.33 | 11.16 | 8.95 | **8.14** | **7.79** | **7.48** | **7.25** |

To validate the contributions of each component of EVODiff, we performed a constructive ablation study (Table 3), which builds upon our initial finding that entropy-reduction (RE) based methods consistently outperform traditional FD-based approaches (Figure 2). Starting from a baseline second-order solver, we incrementally introduced our entropy-reduction (RE-based) formulation and the final evolution-state-driven parameter optimization. Each step demonstrates a clear improvement in FID, culminating in the full EVODiff algorithm which consistently achieves the best performance. Further detailed ablations on hyperparameters such as the step-size ratio $r_i$ and the shift parameter $\mu$ can be found in Appendix E.7.1 (Tables 11 and 9 in of Appendix E.7.1), confirming the robustness of our method. Additionally, we demonstrate the generalizability of our core principles by applying them to enhance other frameworks like DPM-Solver-v3, with results shown in Table 14 and Figure 8.

We evaluate our method against advanced gradient-based solvers, including DPM-Solver++ [20], DEIS [14], UniPC [16], and DPM-Solver-v3 [17] on CIFAR-10, FFHQ-64, ImageNet-64, and ImageNet-256 datasets. The results in Tables 2 and 11 and Figure 4 consistently demonstrate the superior performance of EVODiff. Furthermore, we evaluate our method with available logSNR and EDM noise schedules, further validating its consistently robust performance. The results are shown in Tables 12, 18, 19, 17, 16. We also evaluate our method on the text-to-image generation task, as shown

Table 4: FID score (↓) and generation time comparison between EVODiff, DPM-Solver++ (DPM++ for short), and UniPC. All methods were evaluated on a latent diffusion model [28] trained on the LSUN-Bedrooms dataset with 50k samples.

| | FID Score (↓) | | | | | Generation Time (s) | | |
|---|---|---|---|---|---|---|---|---|
| NFE | DPM++(2m) | DPM++(3m) | UniPC(3m) | **EVODiff** | Gain | DPM++ | **EVODiff** | Gain |
| 5 | 21.286 | 18.611 | 13.969 | **7.912** | 43.4% | 3577.6 | **3488.4** | -89.2 (2.5%) |
| 6 | 10.966 | 8.519 | 6.556 | **4.909** | 25.1% | 3800.6 | **3719.4** | -81.2 (2.1%) |
| 8 | 5.127 | 4.148 | 3.963 | **3.756** | 5.2% | 4273.3 | **4046.9** | -226.4 (5.3%) |
| 10 | 3.881 | 3.607 | 3.563 | **3.332** | 6.5% | 4746.7 | **4699.6** | -47.1 (1.0%) |
| 12 | 3.516 | 3.429 | 3.357 | **3.084** | 8.1% | 4703.8 | **4678.1** | -25.7 (0.5%) |
| 15 | 3.341 | 3.284 | 3.182 | **2.918** | 8.3% | 5973.1 | **5913.5** | -59.6 (1.0%) |
| 20 | 3.251 | 3.167 | 3.075 | **2.853** | 7.2% | 7238.4 | **7154.2** | -84.2 (1.2%) |

Table 5: Single-batch sample quality comparison for text-to-image generation, measured by CLIP score (↑) and Aesthetic score (↑) using Stable Diffusion v1.4 and v1.5 under identical settings.

| model | method | NFN=10 | | | | NFN=25 | | | |
|---|---|---|---|---|---|---|---|---|---|
| | | CLIP | | Aesthetic | | CLIP | | Aesthetic | |
| | | Average | maximum | Average | maximum | Average | maximum | Average | maximum |
| sd-v1.4 | DPM-Solver++ | 33.07 | 34.12 | 5.71 | 5.83 | 32.66 | **36.12** | 5.74 | 5.92 |
| | EVODiff | **33.79** | **35.00** | **5.77** | **5.87** | **32.83** | 34.50 | **5.79** | **5.97** |
| sd-v1.5 | DPM-Solver++ | 32.98 | 35.84 | 5.74 | **5.99** | **32.54** | 34.70 | 5.79 | 5.92 |
| | EVODiff | **33.07** | **36.62** | **5.75** | 5.98 | 32.53 | **34.72** | **5.81** | **5.99** |

in Table 5. Figures 3 and 5 provide a visual comparison of a generated sample. Finally, Table 15 compares the inference time at various NFEs on ImageNet-256 between our method and the baseline.

Finally, beyond pixel-space DMs, we evaluate EVODiff against advanced gradient-based solvers on popular latent-space DMs. On the LSUN-Bedrooms dataset (Table 4), EVODiff consistently achieves the best FID scores across all NFE settings, with particularly significant improvements at low NFE counts (*43.4%* reduction at 5 NFE, from *13.969* to *7.912* compared to UniPC). Notably, EVODiff also reduces generation time by up to *5.3%* while maintaining superior quality. For text-to-image generation using Stable Diffusion v1.4 and v1.5 (Table 5), EVODiff achieves competitive CLIP scores and the best Aesthetic scores compared to the strong DPM-Solver++ baseline, demonstrating its effectiveness in preserving both semantic alignment and visual quality.

## Conclusions

In this work, we propose *EVODiff*, a novel inference-time refinement method based on entropy-aware variance optimization. It significantly improves both efficiency and generative quality without relying on reference trajectories. Specifically, our work first establishes a principled, information-theoretic foundation that explains why data-prediction parameterization outperforms its noisy counterpart and demonstrates how optimizing conditional variance reduces transition and reconstruction errors *without relying on reference trajectories*. Building on these insights, EVODiff systematically reduces uncertainty in each denoising step, thereby accelerating convergence and significantly improving sample quality. Extensive experiments demonstrate EVODiff's effectiveness across diverse settings with SOTA performance: on CIFAR-10, it outperforms the DPM-Solver++ baseline by *45.5%* at 10 NFE (from *5.10* to *2.78* FID); on ImageNet-256, it reduces NFE cost by *25%* (from 20 to 15 NFE) while maintaining high-fidelity generation; on LSUN-Bedrooms, it achieves up to *43.4%* FID improvement over UniPC with *5.3%* faster generation; and for text-to-image generation with Stable-Diffusion models, it produces superior visual quality while preserving semantic alignment. Our method achieves SOTA results and establishes a reference-free, variance-controlled inference framework, effectively addressing the trade-off between sampling efficiency and generative quality.

**Limitations and Broader Impacts** A limitation of EVODiff is that it currently relies on the data-prediction parameterization for diffusion model inference. Additionally, leveraging information-theoretic principles to enhance inference efficiency and optimize information flow during the sampling process remains a promising direction for future research. While EVODiff improves both generation quality and efficiency, we acknowledge its dual-use nature, similar to that of other generative models.

## Acknowledgments

This work was supported in part by grants from National Natural Science Foundation of China (52539005), the fundamental research program of Guangdong, China (2023A1515011281), the China Scholarship Council (202306150167), Guangdong Basic and Applied Basic Research Foundation (24202107190000687), Foshan Science and Technology Research Project(2220001018608).

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

# Appendix

## Table of Contents

# A List of Notations

| Symbol | Description |
|---|---|
| $t, s, l, T$ | Time variables and endpoint in the diffusion process, $t \in [0, T]$. |
| $i, j, k$ | Indices for discrete time steps. |
| $\lambda(t)$ | Log-Signal-to-Noise Ratio (log-SNR) time, defined as $\log(\alpha_t/\sigma_t)$. |
| $\boldsymbol{x}_t, \boldsymbol{x}_{t_i}, \tilde{\boldsymbol{x}}_t$ | State vector at time $t$ (continuous, discrete, or approximate). |
| $\boldsymbol{x}_0, \boldsymbol{x}_T$ | Endpoints: $\boldsymbol{x}_0 \sim q(\boldsymbol{x}_0)$, $\boldsymbol{x}_T \sim \mathcal{N}(\boldsymbol{0}, \hat{\sigma}^2 \boldsymbol{I})$. |
| $\alpha_t, \sigma_t$ | Noise schedule parameters (signal preservation $\alpha_t$, noise level $\sigma_t$). |
| $\boldsymbol{\epsilon}$ | Random noise sampled from $\mathcal{N}(\boldsymbol{0}, \boldsymbol{I})$. |
| $\boldsymbol{\omega}_t, \overline{\boldsymbol{\omega}}_t$ | Forward and reverse Wiener processes. |
| SNR | Signal-to-noise ratio: $\alpha_t^2/\sigma_t^2$. |
| $\theta$ | Parameters of the neural network model. |
| $\boldsymbol{\epsilon}_\theta(\cdot), \boldsymbol{x}_\theta(\cdot), \boldsymbol{d}_\theta(\cdot)$ | Network predictions: noise, clean data, unified noise and data. |
| $\boldsymbol{d}_\theta^{(k)}(\cdot)$ | $k$-th order derivative of $\boldsymbol{d}_\theta$ w.r.t. $\tau$. |
| $q(\cdot)$ | Forward process distributions, e.g., $q(\boldsymbol{x}_0), q(\boldsymbol{x}_t), q(\boldsymbol{x}_t\|\boldsymbol{x}_0)$. |
| $p(\boldsymbol{x}_{t_i}\|\boldsymbol{x}_{t_{i+1}})$ | Reverse transition distribution. |
| $\mathcal{N}(\mu, \Sigma), \mathcal{U}(a, b)$ | Gaussian distribution with mean $\mu$, covariance $\Sigma$; uniform distribution on $[a, b]$. |
| $f(t), g(t)$ | Drift and diffusion coefficients, $f(t) = \frac{\mathrm{d}\log\alpha_t}{\mathrm{d}t}$, $g(t)$ satisfies $g^2(t) = \frac{\mathrm{d}\sigma_t^2}{\mathrm{d}t} - 2\frac{\mathrm{d}\log\alpha_t}{\mathrm{d}t}\sigma_t^2$. |
| $\boldsymbol{f}(\boldsymbol{x}_t)$ | Transformation: $\frac{\boldsymbol{x}_t}{\alpha_t}$ (noise-prediction) or $\frac{\boldsymbol{x}_t}{\sigma_t}$ (data-prediction). |
| $\kappa(t), \psi(\tau)$ | Time reparameterization $\kappa(t)$ and its inverse $\psi(\tau)$ with $\psi(\kappa(t)) = t$. |
| $\boldsymbol{\iota}(\boldsymbol{x}_{t_{i-1}})$ | Difference term: $\boldsymbol{f}(\boldsymbol{x}_{t_{i-1}}) - \boldsymbol{f}(\boldsymbol{x}_{t_i})$. |
| $h_{t_i}, \hat{h}_{t_i}$ | Step sizes: $\kappa(t_{i-1}) - \kappa(t_i)$, auxiliary $\kappa(s_i) - \kappa(t_i)$. |
| $h_{\lambda_i}$ | Step size in the log-SNR space: $\lambda(t_{i-1}) - \lambda(t_i)$. |
| $r_i, r_{\mathrm{logSNR}}(i)$ | Ratio of consecutive step sizes in log-SNR space, used for gradient estimation. |
| $F_\theta, B_\theta, \bar{B}_\theta$ | Finite difference terms for gradient approximation (Forward, Backward, and unified). |
| $\mathrm{Var}(\cdot\|\cdot), H_p(\cdot\|\cdot), \boldsymbol{\mu}_{t_i\|t_{i+1}}, \boldsymbol{\Sigma}_{t_i}$ | Conditional variance, entropy, mean, and covariance matrix. |
| $\zeta_i, \bar{\zeta}_i, \eta_i, \mu$ | Optimization parameters (interpolation, complement, balance, shift). |
| $\mathcal{L}(\theta), \mathcal{L}_1, \mathcal{L}_2, w(t)$ | General training objective (e.g., Eq. (3)), specific optimization objectives ($\mathcal{L}_1, \mathcal{L}_2$), and training weight $w(t)$. |
| $G(\zeta_i), P(\boldsymbol{x}_{t_{i-1}})$ | Auxiliary interpolation and optimization terms. |
| $D_i, E(t_{i-1}, t_i), \tilde{\Delta}_{t_i}^g$ | Difference, gradient error, and weighted gradient terms. |
| $\nabla_{\boldsymbol{x}}, \mathbb{E}[\cdot]$ | Gradient, expectation. |
| $\mathrm{Tr}(\cdot), \det(\cdot), \|\cdot\|$ | Trace, determinant, and norm operators. |
| $\mathrm{vec}(\cdot)$ | Vectorization operator. |
| $\mathbb{R}^d, \boldsymbol{I}, \boldsymbol{0}, C$ | $d$-dimensional real space, identity matrix, zero vector, Gaussian entropy constant $C = \frac{1}{2}d(\log 2\pi + 1)$. |

# B Related Work

## B.1 Diffusion Models

Diffusion Models (DMs) represent a powerful class of generative models rooted in stochastic thermodynamics and statistical physics [29, 30]. The foundational work by Sohl-Dickstein et al. [1]

adapted these principles to deep generative modeling through a Markov chain approach based on non-equilibrium thermodynamics. This pioneering research addressed the critical challenge of balancing tractability and flexibility in probabilistic models. The field of diffusion modeling was substantially extended by Song and Ermon [31, 32], who introduced score-based generative models with noise conditional score networks. Their methodology enabled efficient estimation of score functions $\nabla \mathbf{x} \log p_\sigma(\mathbf{x})$ across multiple noise scales, coupled with annealed Langevin dynamics for sample generation. Subsequently, Ho et al. [2] proposed Denoising Diffusion Probabilistic Models (DDPMs), which offered a significant methodological refinement by parameterizing the reverse process. Their contribution included a well-formulated training objective: $L = \mathbb{E}_{t,\boldsymbol{\epsilon}} \|\boldsymbol{\epsilon} - \boldsymbol{\epsilon}_\theta(\boldsymbol{x}_t, t)\|^2$, where $\boldsymbol{\epsilon}$ represents the original noise and $\boldsymbol{\epsilon}_\theta$ denotes the predicted noise by the model at time step $t$. This formulation facilitated high-quality sample generation with remarkable stability. In a landmark contribution, Song et al. [3] established a comprehensive theoretical unification by formulating score-based models and DMs within a continuous-time stochastic differential equation (SDE) framework. This theoretical advancement provided a unified mathematical foundation that elegantly bridged previously disparate approaches in diffusion modeling research.

Building upon this theoretical framework, DMs have demonstrated exceptional capabilities across a wide range of domains. In image synthesis, they have achieved SOTA performance [4] and established new benchmarks in photorealism [12]. Their success has extended to multimodal generation tasks, including text-to-image synthesis [6, 33], speech generation [7], video synthesis [8], 3D content generation [34]. Moreover, DMs have also advanced related areas, including deep Gaussian processes [35], diffusion bridges, and density ratio estimation [36, 37]. Furthermore, DMs have shown remarkable capabilities in controllable generation tasks [38], such as image editing, style transfer, and inpainting [5, 39]. In theory, Despite these significant advances, DMs continue to face a critical challenge: the inherently slow sequential generation process, which limits their real-time applicability in certain domains.

## B.2 Training-based Inference for DMs

Training-based models or methods accelerate DMs through novel training strategies and architectures. Knowledge distillation techniques, such as Progressive Distillation [40], enable efficient sampling by allowing student models to learn compressed sampling processes from teacher models. Recent advances have further explored innovative approaches to diffusion model inference. Meng et al. [41] investigated knowledge distillation in guided DMs, addressing model efficiency challenges. Complementing these efforts, Karras et al. [42, 43] conducted a comprehensive analysis of training dynamics for DMs, offering critical insights into the underlying mechanisms of model performance and generation quality. Consistency-based methods, exemplified by Consistency Models [44–46] and Latent Consistency Models [47], achieve parallel generation by learning score functions through consistency training, grounded in probability flow ODE frameworks. Reflow [48] further optimizes the generation paths by reformulating rectified flow ODEs with paired retraining strategies. Architectural innovations have also played an important role in improving efficiency. Latent DMs [28] reduce computational complexity by operating in lower-dimensional spaces using an auto-encoder framework [49]. EDM [12] introduces $\sigma$-parameterization and principled weighting schemes, allowing fewer sampling steps without compromising the generation quality. Shortcut models [50] and Mean Flows [51] achieve efficiency by step-aware network learning and mean field modeling. In addition, there are some other approaches that exploit architectural characteristics and learning-based solvers to improve the efficiency of DMs [52–59].

Despite these advancements, training-based methods often require specialized training procedures and a careful balance between quality and speed trade-offs [4, 60, 50, 51], making them computationally expensive. Additionally, their training data are typically obtained through iterative sampling from pre-trained DMs using deterministic, training-free samplers such as DDIM [9] or DPM-Solver [13, 20], which introduces further computational overhead and sampling dependencies. These factors limit their practicality in resource-constrained environments. Furthermore, while some methods achieve generation through a single neural network pass (referred to as one-step generation), similar to GANs [61], they often sacrifice the iterative refinement process that is a hallmark of traditional DMs. This process, which involves progressive noise reduction and iterative refinement, is a core strength of DMs, and its absence may adversely affect the quality and controllability of the generated outputs.

## B.3 Training-free Inference for DMs

In contrast, training-free inference methods focus on denoising strategies for DMs without requiring any additional training, making them more adaptable and practical for use with open-source DMs. Early sampling methods in DMs primarily relied on ancestral sampling [2]. Score-based models [3] used predictor-corrector methods to refine samples and introduced PF ODEs as a faster sampling alternative. DDIM [9] advanced sampling methods by introducing a non-Markovian deterministic process that enables deterministic sampling through a variance-minimizing path, significantly reducing the number of required steps. PNDM [11] demonstrated the adaptability of ODE solvers to diffusion sampling by effectively utilizing linear multistep methods. EDM [12] explored the design space of DMs with a $\sigma$-parameterization linked to the signal-to-noise ratio (SNR), analyzed noise dynamics to optimize time steps, and achieved high-quality samples using the Heun solver.

DPM-Solver [13] introduced an exponential integrator-based (EI) sampling framework that discretizes PF ODEs in the semi-log-SNR space with high-order solvers for accelerated sampling. DEIS [14] investigated the effectiveness of EI in addressing the stiffness of diffusion ODEs. DPM-Solver++ [20] extended DPM-Solver to guided sampling by using data-based parameterization. Based on DPM-Solver, UniPC [16] designed high-order predictor-corrector schemes within a unified framework and demonstrated strong empirical performance. These methods effectively focus on optimizing the variance term of reconstruction error, as formulated in our proposed decomposition in Eq. (3.1).

Beyond these solvers, another approach reformulates ODE solvers by treating the solutions at multiple steps as ground truth. Leveraging prior information about the target distribution, this strategy simultaneously optimizes both the variance and bias terms in the reconstruction error, as decomposed in Eq. (3.1). For instance, DPM-Solver-v3 [17] accelerates the sampling inference in DMs by optimizing the ODE solver using empirical model statistics (EMS), where the EMS coefficients are learned from the sampling results of DPM-Solver++ with 200 function evaluations (NFEs). In addition, other studies have explored discretization techniques and noise schedule tuning [62, 63, 12, 64–74].

Although various numerical discretization techniques and approaches have been proposed for training-free methods, the underlying mechanisms driving their acceleration are not adequately understood. Despite their empirical success, these ODE-based methods lack an information-theoretic foundation. A central limitation is their neglect of information transmission efficiency during the reverse process. This theoretical gap suggests that the principles governing diffusion inference remain underexplored. Our work addresses this limitation by introducing a framework that unifies efficient numerical iterations, such as DPM-Solver and EDM, through the lens of conditional entropy reduction. We demonstrate that by explicitly optimizing for conditional entropy dynamics rather than focusing solely on optimizing the numerical error of ODE solvers, we can achieve better sample quality across inference steps. EVODiff is designed to fill this gap by introducing an explicit and optimizable information-theoretic objective.

# C   Analysis and Proofs of Variance-Driven Conditional Entropy Reduction

## C.1   The Proof of Proposition 3.1

*Proof.* We prove the decomposition of the reconstruction error using the orthogonality property of conditional expectations in Proposition 3.1. First, we express the squared norm as a sum of components:

$$\|\boldsymbol{x}_{t_i} - \boldsymbol{x}_0\|^2 = \|(\boldsymbol{x}_{t_i} - \boldsymbol{\mu}_{t_i|t_{i+1}}) + (\boldsymbol{\mu}_{t_i|t_{i+1}} - \boldsymbol{x}_0)\|^2 \tag{26}$$

$$= \|\boldsymbol{x}_{t_i} - \boldsymbol{\mu}_{t_i|t_{i+1}}\|^2 + \|\boldsymbol{\mu}_{t_i|t_{i+1}} - \boldsymbol{x}_0\|^2$$

$$+ 2\langle \boldsymbol{x}_{t_i} - \boldsymbol{\mu}_{t_i|t_{i+1}}, \boldsymbol{\mu}_{t_i|t_{i+1}} - \boldsymbol{x}_0 \rangle \tag{27}$$

Taking the expectation of both sides, we obtain

$$\mathbb{E}_q[\|\boldsymbol{x}_{t_i} - \boldsymbol{x}_0\|^2] = \mathbb{E}_q[\|\boldsymbol{x}_{t_i} - \boldsymbol{\mu}_{t_i|t_{i+1}}\|^2] + \mathbb{E}_q[\|\boldsymbol{\mu}_{t_i|t_{i+1}} - \boldsymbol{x}_0\|^2] + 2C \tag{28}$$

where $C = \mathbb{E}_q[\langle \boldsymbol{x}_{t_i} - \boldsymbol{\mu}_{t_i|t_{i+1}}, \boldsymbol{\mu}_{t_i|t_{i+1}} - \boldsymbol{x}_0 \rangle]$. Next, we show that the cross-term $C$ vanishes:

$$C = \mathbb{E}_q \left[ \mathbb{E}_q \left[ \langle \boldsymbol{x}_{t_i} - \boldsymbol{\mu}_{t_i|t_{i+1}}, \boldsymbol{\mu}_{t_i|t_{i+1}} - \boldsymbol{x}_0 \rangle \mid \boldsymbol{x}_{t_{i+1}} \right] \right] \tag{29}$$

$$= \mathbb{E}_q \left[ \langle \mathbb{E}_q[\boldsymbol{x}_{t_i} - \boldsymbol{\mu}_{t_i|t_{i+1}} \mid \boldsymbol{x}_{t_{i+1}}], \boldsymbol{\mu}_{t_i|t_{i+1}} - \boldsymbol{x}_0 \rangle \right] \tag{30}$$

Since $\boldsymbol{\mu}_{t_i|t_{i+1}} = \mathbb{E}_q[\boldsymbol{x}_{t_i} \mid \boldsymbol{x}_{t_{i+1}}]$ by definition, we have $\mathbb{E}_q[\boldsymbol{x}_{t_i} - \boldsymbol{\mu}_{t_i|t_{i+1}} \mid \boldsymbol{x}_{t_{i+1}}] = \mathbf{0}$, and therefore $C = 0$. Therefore, we obtain the final decomposition:

$$\mathbb{E}_q[\|\boldsymbol{x}_{t_i} - \boldsymbol{x}_0\|^2] = \underbrace{\mathbb{E}_q[\|\boldsymbol{x}_{t_i} - \boldsymbol{\mu}_{t_i|t_{i+1}}\|^2]}_{\text{Variance term}} + \underbrace{\mathbb{E}_q[\|\boldsymbol{\mu}_{t_i|t_{i+1}} - \boldsymbol{x}_0\|^2]}_{\text{Bias term}}. \tag{31}$$

The proof is complete. $\qquad\square$

## C.2 Proof of Proposition 3.2

*Proof.* Denote the Gaussian transition distributions governed by the iterative equations (6) and (7) as $p_1\left(\boldsymbol{f}(\tilde{\boldsymbol{x}}_{t_{i-1}})|\boldsymbol{f}(\tilde{\boldsymbol{x}}_{t_i})\right)$ and $p_2\left(\boldsymbol{f}(\tilde{\boldsymbol{x}}_{t_{i-1}})|\boldsymbol{f}(\tilde{\boldsymbol{x}}_{t_i})\right)$, respectively. Without loss of generality, we use the common part $\boldsymbol{f}(\tilde{\boldsymbol{x}}_{t_i})$ of the two iterative equations as the mean of both distributions. The remaining components represent the perturbation terms associated with each transition distribution, respectively. Since the noise prediction model is specifically trained to predict the noise, we can interpret $\boldsymbol{\epsilon}_\theta(\tilde{\boldsymbol{x}}_t, t)$ as representing the noise perturbation term. Since the estimated noise by the model at different time steps can be considered mutually independent, the conditional variances of the remaining terms for the two different iterations are, respectively, expressed as follows:

$$\text{Var}_{p_1} = h_{t_i}^2 \cdot \text{Var}(\boldsymbol{\epsilon}_\theta(\tilde{\boldsymbol{x}}_{t_i}, t_i)), \ \text{Var}_{p_2} = h_{t_i}^2 \left(1 - \frac{h_{t_i}}{2\hat{h}_{t_i}}\right)^2 \cdot \text{Var}(\boldsymbol{\epsilon}_\theta(\tilde{\boldsymbol{x}}_{t_i}, t_i)) + \frac{h_{t_i}^4}{4\hat{h}_{t_i}^2} \cdot \text{Var}(\boldsymbol{\epsilon}_\theta(\tilde{\boldsymbol{x}}_{s_i}, s_i)). \tag{32}$$

Denote $\Delta H(p) = H_{p_2}(\tilde{\boldsymbol{x}}_{t_{i-1}}|\tilde{\boldsymbol{x}}_{t_i}) - H_{p_1}(\tilde{\boldsymbol{x}}_{t_{i-1}}|\tilde{\boldsymbol{x}}_{t_i})$. Then, by equations (32) and (9), we have:

$$\Delta H(p) = \frac{d}{2} \log \left| 1 - \frac{h_{t_i}}{\hat{h}_{t_i}} + \frac{h_{t_i}^2}{4\hat{h}_{t_i}^2} + \frac{h_{t_i}^2}{4\hat{h}_{t_i}^2} \cdot \frac{\text{Var}(\boldsymbol{\epsilon}_\theta(\tilde{\boldsymbol{x}}_{s_i}, s_i))}{\text{Var}(\boldsymbol{\epsilon}_\theta(\tilde{\boldsymbol{x}}_{t_i}, t_i))} \right|. \tag{33}$$

Therefore, $\Delta H(p) \leq 0$ if and only if $\frac{h_{t_i}^2}{4\hat{h}_{t_i}^2} + \frac{h_{t_i}^2}{4\hat{h}_{t_i}^2} \cdot \frac{\text{Var}(\boldsymbol{\epsilon}_\theta(\tilde{\boldsymbol{x}}_{s_i}, s_i))}{\text{Var}(\boldsymbol{\epsilon}_\theta(\tilde{\boldsymbol{x}}_{t_i}, t_i))} \leq \frac{h_{t_i}}{\hat{h}_{t_i}}$. By solving this inequality and note that $\hat{h}_{t_i} \leq h_{t_i}$, the proof is complete. $\qquad\square$

As the reverse process in DMs aims to estimate $p(\boldsymbol{x}_t|\boldsymbol{x}_{t+1}, \boldsymbol{x}_0)$ [2, 25], we examine $\text{Var}(\boldsymbol{\epsilon}_\theta(\tilde{\boldsymbol{x}}_t, t) \mid \boldsymbol{x}_0)$ to capture the model's uncertainty in noise prediction conditioned on the clean data. For brevity, we denote this variance as $\text{Var}(\boldsymbol{x}_\theta(\tilde{\boldsymbol{x}}_t, t))$. Based on this consideration, we can establish the practical interval for Proposition 3.2 using the prior-like conditional variance from the forward diffusion process.

*Remark* C.1. In the forward process of DMs, the clean data at each step can be expressed by $\boldsymbol{x}_0 = \boldsymbol{x}_t/\alpha_t - \sigma_t/\alpha_t\boldsymbol{\epsilon}$. If we assume that $\text{Var}(\boldsymbol{\epsilon}_\theta(\tilde{\boldsymbol{x}}_t, t)) \propto \sigma_t^2/\alpha_t^2$ to quantify the extent of deviation from the clean data. Under this prior-like assumption, we obtain $\frac{\text{Var}(\boldsymbol{\epsilon}_\theta(\tilde{\boldsymbol{x}}_{s_i}, s_i))}{\text{Var}(\boldsymbol{\epsilon}_\theta(\tilde{\boldsymbol{x}}_{t_i}, t_i))} = \frac{\text{SNR}(t_i)}{\text{SNR}(s_i)}$. Then, the conditional entropy reduction condition in Propostion 3.2 is $h_{t_i}/\hat{h}_{t_i} \in \left[1, \frac{4\,\text{SNR}(s_i)}{\text{SNR}(t_i)+\text{SNR}(s_i)}\right]$.

## C.3 The Perspective of Conditional Entropy Reduction for Some Accelerated Iterations

As an application of conditional entropy analysis, we deepen our understanding of the iterations in accelerated denoising diffusion solvers, such as DPM-Solver [13] and EDM [12], by elucidating the associated changes in conditional entropy. We then demonstrate that the iterations of both well-known solvers are denoising iterations grounded in conditional entropy reduction and represent two special cases of RE-based iterations.

Firstly, let us revisit the accelerated iteration introduced by EDM [12]. Formally, the iteration formula of EDM can be written as follows:

$$\boldsymbol{f}(\tilde{\boldsymbol{x}}_{t_{i-1}}) = \boldsymbol{f}(\tilde{\boldsymbol{x}}_{t_i}) + h_{t_i} \frac{\boldsymbol{\epsilon}_\theta\left(\tilde{\boldsymbol{x}}_{t_i}, t_i\right) + \boldsymbol{\epsilon}_\theta\left(\tilde{\boldsymbol{x}}_{t_{i-1}}, t_{i-1}\right)}{2}, \tag{34}$$

which can be equivalently rewritten as the following gradient estimation-based iteration:

$$\boldsymbol{f}(\tilde{\boldsymbol{x}}_{t_{i-1}}) = \boldsymbol{f}(\tilde{\boldsymbol{x}}_{t_i}) + h_{t_i}\boldsymbol{\epsilon}_\theta\left(\tilde{\boldsymbol{x}}_{t_i}, t_i\right) + \frac{h_{t_i}^2}{2}\frac{\boldsymbol{\epsilon}_\theta\left(\tilde{\boldsymbol{x}}_{t_{i-1}}, t_{i-1}\right) - \boldsymbol{\epsilon}_\theta\left(\tilde{\boldsymbol{x}}_{t_i}, t_i\right)}{h_{t_i}}. \tag{35}$$

As $\hat{h}_{t_i} = h_{t_i}$ in the iteration of EDM described by Eq. (35), based on Remark C.1, we obtain the following conclusion:

*Remark* C.2. The EDM iteration in Eq. (34) can reduce conditional entropy more effectively than the DDIM iteration in Eq. (6). Thus, the iteration of EDM can be interpreted as an iterative scheme for reducing conditional entropy.

Next, we revisit the accelerated iteration framework established by DPM-Solver [13] with exponential integrator. Specifically, the sampling algorithm of DPM-Solver decouples the semi-linear structure of the diffusion ODE, with its iterations formulated by solving the integral driven by half of the log-SNR. The exponentially weighted score integral in DPM-Solver can be written as follows:

$$\boldsymbol{f}(\boldsymbol{x}_t) - \boldsymbol{f}(\boldsymbol{x}_s) = -\int_{\lambda(s)}^{\lambda(t)} e^{-\tau}\boldsymbol{\epsilon}_\theta\left(\boldsymbol{x}_{\psi(\tau)}, \psi(\tau)\right)\mathrm{d}\tau. \tag{36}$$

where $\lambda(t) := \log\frac{\alpha_t}{\sigma_t}$. It follows that Eq. (36) and Eq. (5) can be mutually transformed through the function relation $\lambda(t) = -\log(\kappa(t))$. Denote $h_{\lambda_i} := \lambda(t_{i-1}) - \lambda(t_i)$ and $\hat{h}_{\lambda_i} := \lambda(s_i) - \lambda(t_i)$. Formally, the second-order iteration of DPM-Solver can be written as follows:

$$\boldsymbol{f}(\tilde{\boldsymbol{x}}_{t_{i-1}}) = \boldsymbol{f}(\tilde{\boldsymbol{x}}_{t_i}) - \frac{\sigma_{t_{i-1}}}{\alpha_{t_{i-1}}}\left(e^{h_{\lambda_i}} - 1\right)\boldsymbol{\epsilon}_\theta\left(\tilde{\boldsymbol{x}}_{t_i}, t_i\right) - \frac{\sigma_{t_{i-1}}}{\alpha_{t_{i-1}}}\left(e^{h_{\lambda_i}} - 1\right)\frac{\boldsymbol{\epsilon}_\theta\left(\tilde{\boldsymbol{x}}_{s_i}, s_i\right) - \boldsymbol{\epsilon}_\theta\left(\tilde{\boldsymbol{x}}_{t_i}, t_i\right)}{2r_1},$$
$$\tag{37}$$

where $s_i = \psi\left(\lambda(t_i) + r_1 h_{\lambda_i}\right)$. Note that $r_1 = \frac{\hat{h}_{\lambda_i}}{h_{\lambda_i}}$, $\kappa(t_{i-1}) = \frac{\sigma_{t_{i-1}}}{\alpha_{t_{i-1}}}$ and $h_{t_i} = \kappa(t_{i-1}) - \kappa(t_i)$. As $e^{h_{\lambda_i}} = \frac{\kappa(t_i)}{\kappa(t_{i-1})}$, then $\frac{\sigma_{t_{i-1}}}{\alpha_{t_{i-1}}}\left(e^{h_{\lambda_i}} - 1\right) = -h_{t_i}$. Thus, this second-order iteration can be equivalently rewritten as:

$$\boldsymbol{f}(\tilde{\boldsymbol{x}}_{t_{i-1}}) = \boldsymbol{f}(\tilde{\boldsymbol{x}}_{t_i}) + h_{t_i}\boldsymbol{\epsilon}_\theta\left(\tilde{\boldsymbol{x}}_{t_i}, t_i\right) + \frac{h_{t_i} h_{\lambda_i}}{2}\frac{\boldsymbol{\epsilon}_\theta\left(\tilde{\boldsymbol{x}}_{s_i}, s_i\right) - \boldsymbol{\epsilon}_\theta\left(\tilde{\boldsymbol{x}}_{t_i}, t_i\right)}{\hat{h}_{\lambda_i}}. \tag{38}$$

Note that the $s_i$ here in DPM-Solver differs from the one in Eq. (7), due to the variations arising from the function space. Based on conditional analysis, similarly, we have the following conclusion.

*Remark* C.3. Based on Remark C.1, when $\frac{h_{\lambda_i}}{\hat{h}_{\lambda_i}} \in \left[1, \frac{4\,\mathrm{SNR}(s_i)}{\mathrm{SNR}(t_i)+\mathrm{SNR}(s_i)}\right]$, the DPM-Solver's iteration in Eq. (37) can reduce conditional entropy more effectively than the DDIM iteration in Eq. (6). Note that $\frac{h_{\lambda_i}}{\hat{h}_{\lambda_i}} = 2$ in the practical implementation of DPM-Solver. Thus, as $\mathrm{SNR}(s_i) > \mathrm{SNR}(t_i)$, the DPM-Solver's iteration can be interpreted as an iterative scheme for reducing conditional entropy.

Finally, we summarize the relationship between these two iterations and RE-based iterations. In fact, the iteration described in Eq. (34) is an RE-based iteration within the EDM iteration framework. Clearly, the RE-based iteration within the DPM-Solver iteration framework can be formulated as:

$$\boldsymbol{f}(\tilde{\boldsymbol{x}}_{t_{i-1}}) = \boldsymbol{f}(\tilde{\boldsymbol{x}}_{t_i}) + h_{t_i}\left(\gamma\boldsymbol{\epsilon}_\theta\left(\tilde{\boldsymbol{x}}_{s_i}, s_i\right) + (1-\gamma)\boldsymbol{\epsilon}_\theta\left(\tilde{\boldsymbol{x}}_{t_i}, t_i\right)\right) + \frac{h_{t_i} h_{\lambda_i}}{2}\frac{\boldsymbol{\epsilon}_\theta\left(\tilde{\boldsymbol{x}}_{s_i}, s_i\right) - \boldsymbol{\epsilon}_\theta\left(\tilde{\boldsymbol{x}}_{t_i}, t_i\right)}{\hat{h}_{\lambda_i}}. \tag{39}$$

Therefore, the iterations in both EDM and DPM-Solver can be interpreted as specific instances of RE-based denoising iterations from the perspective of the conditional entropy.

This analysis reveals that methods like DPM-Solver and EDM have implicitly leveraged principles of conditional entropy reduction. Our work, EVODiff, makes this process explicit, optimizable, and adaptive for the first time, which is the key to its superior performance.

## C.4 Difference Analysis of Gradient-based Iterations in Multi-step Framework

On one hand, let us revisit the multi-step accelerated framework established by DPM-Solver++ [20]. Formally, the second-order iteration of DPM-Solver++ can be written as follows:

$$
\boldsymbol{f}(\tilde{\boldsymbol{x}}_{t_{i-1}}) = \boldsymbol{f}(\tilde{\boldsymbol{x}}_{t_i}) - \frac{\alpha_{t_{i-1}}}{\sigma_{t_{i-1}}} \left( e^{-h_{\lambda_i}} - 1 \right) \boldsymbol{x}_\theta \left( \tilde{\boldsymbol{x}}_{t_i}, t_i \right)
$$
$$
- \frac{\alpha_{t_{i-1}}}{\sigma_{t_{i-1}}} \left( e^{-h_{\lambda_i}} - 1 \right) \frac{\boldsymbol{x}_\theta \left( \tilde{\boldsymbol{x}}_{t_i}, t_i \right) - \boldsymbol{x}_\theta \left( \tilde{\boldsymbol{x}}_{t_{i+1}}, t_{i+1} \right)}{2 r_i},
\tag{40}
$$

where $\boldsymbol{x}_\theta \left( \tilde{\boldsymbol{x}}_{t_i}, t_i \right)$ denotes the data-prediction prediction model and $r_i = \frac{h_{\lambda_{i+1}}}{h_{\lambda_i}}$. Since $\frac{\alpha_{t_{i-1}}}{\sigma_{t_{i-1}}} \left( e^{-h_{\lambda_i}} - 1 \right) = \frac{\alpha_{t_i}}{\sigma_{t_i}} - \frac{\alpha_{t_{i-1}}}{\sigma_{t_{i-1}}} = -h_{t_i}$ in data-prediction prediction models, Eq. (40) can be rewritten as

$$
\boldsymbol{f}(\tilde{\boldsymbol{x}}_{t_{i-1}}) = \boldsymbol{f}(\tilde{\boldsymbol{x}}_{t_i}) + h_{t_i} \boldsymbol{x}_\theta \left( \tilde{\boldsymbol{x}}_{t_i}, t_i \right) + \frac{h_{t_i} h_{\lambda_i}}{2} \frac{\boldsymbol{x}_\theta \left( \tilde{\boldsymbol{x}}_{t_i}, t_i \right) - \boldsymbol{x}_\theta \left( \tilde{\boldsymbol{x}}_{t_{i+1}}, t_{i+1} \right)}{h_{\lambda_{i+1}}}.
\tag{41}
$$

On the other hand, we can rewrite the iteration presented in Eq. (13) as follows:

$$
\boldsymbol{f}(\tilde{\boldsymbol{x}}_{t_{i-1}}) = \boldsymbol{f}(\tilde{\boldsymbol{x}}_{t_i}) + h_{t_i} \boldsymbol{x}_\theta \left( \tilde{\boldsymbol{x}}_{t_i}, t_i \right) + \frac{h_{t_i}^2}{2} \frac{\boldsymbol{x}_\theta \left( \tilde{\boldsymbol{x}}_{t_i}, t_i \right) - \boldsymbol{x}_\theta \left( \tilde{\boldsymbol{x}}_{t_{i+1}}, t_{i+1} \right)}{h_{t_{i+1}}}.
\tag{42}
$$

It has been observed that the differences in the multi-step iterations presented in Eq. (41) and Eq. (42) are still caused by the variations in $r_i$. Therefore, in gradient estimation-based iterations, the core characteristic of the DPM-Solver++ iteration is the determination of $r_i$ in the half-logarithmic SNR space. For convenience, we will hereafter refer to half-logarithmic SNR simply as 'logSNR'.

Without loss of generality, the core differences between various gradient estimation-based iterations can be generalized as variations in the determination of $r_i$. Then, a natural question arises: how can $r_i$ be determined better or systematically? Therefore, a principle for determining $r_i$ is of great importance. This inquiry drives our investigation from the perspective of conditional entropy within the context of multi-step iterations.

## C.5 Proof of Theorem 3.4: Conditional Entropy Comparison Between Data Prediction and Noise Prediction parameterizations

Before presenting the formal proof, we provide the core intuition. This theorem aims to show that data-prediction parameterization is more efficient because it directly estimates the target $\boldsymbol{x}_0$, thereby minimizing reconstruction error more directly. In contrast, noise prediction follows a more indirect path $(\boldsymbol{x}_t \to \boldsymbol{\epsilon}_\theta \to \boldsymbol{x}_0)$, which can accumulate more variance. The following steps formalize this variance reduction and its connection to conditional entropy.

*Proof.* Without loss of generality, we only need to prove that the conditional entropy of the first-order iteration using data-prediction parameterization is lower than that of the first-order iteration using noise-prediction parameterization. Let us revisit both first-order denoising iterations. Clearly, based on Eqs. (6) and (5), the first-order iteration of data-prediction parameterization as follows:

$$
\tilde{\boldsymbol{x}}_{t_{i-1}} = \underbrace{\frac{\sigma_{t_{i-1}}}{\sigma_{t_i}} \tilde{\boldsymbol{x}}_{t_i}}_{L_{\text{data}}:\ \text{linear}} + \underbrace{\sigma_{t_{i-1}} \left( \frac{\alpha_{t_{i-1}}}{\sigma_{t_{i-1}}} - \frac{\alpha_{t_i}}{\sigma_{t_i}} \right) \boldsymbol{x}_\theta \left( \tilde{\boldsymbol{x}}_{t_i}, t_i \right)}_{N_{\text{data}}:\ \text{non-linear}},
\tag{43}
$$

where $\boldsymbol{x}_\theta \left( \tilde{\boldsymbol{x}}_{t_i}, t_i \right) = \frac{\tilde{\boldsymbol{x}}_{t_i} - \sigma_{t_i} \boldsymbol{\epsilon}_\theta \left( \tilde{\boldsymbol{x}}_{t_i}, t_i \right)}{\alpha_{t_i}}$. The first-order iteration of noise-prediction parameterization as follows:

$$
\tilde{\boldsymbol{x}}_{t_{i-1}} = \underbrace{\frac{\alpha_{t_{i-1}}}{\alpha_{t_i}} \tilde{\boldsymbol{x}}_{t_i}}_{L_{\text{noise}}:\ \text{linear}} + \underbrace{\alpha_{t_{i-1}} \left( \frac{\sigma_{t_{i-1}}}{\alpha_{t_{i-1}}} - \frac{\sigma_{t_i}}{\alpha_{t_i}} \right) \boldsymbol{\epsilon}_\theta \left( \tilde{\boldsymbol{x}}_{t_i}, t_i \right)}_{N_{\text{noise}}:\ \text{non-linear}}.
\tag{44}
$$

Denote the Gaussian transition kernels governed by the iterative equations (43) and (44) as $p_1 \left( \tilde{\boldsymbol{x}}_{t_{i-1}} | \boldsymbol{x}_0 \right)$ and $p_2 \left( \tilde{\boldsymbol{x}}_{t_{i-1}} | \boldsymbol{x}_0 \right)$, respectively. In both iterations of equations (43) and (44), the

randomness of the linear term is solely related to the noise introduced in the iterations preceding time step $t_i$, whereas the randomness of the nonlinear term depends entirely on the noise at the current time step $t_i$. Since the noise introduced at each time step in DMs is independent, under this assumption, the randomness of the linear term is independent of the randomness of the nonlinear term in both iterations. Therefore, we will consider the variances of the linear and nonlinear components separately. Formally, the variances of the linear terms for the two different iterations are, respectively, as follows:

$$\mathrm{Var}(L_{\mathrm{data}} \mid \boldsymbol{x}_0) = \frac{\sigma_{t_{i-1}}^2}{\sigma_{t_i}^2} \mathrm{Var}(\tilde{\boldsymbol{x}}_{t_i} \mid \boldsymbol{x}_0), \quad \mathrm{Var}(L_{\mathrm{noise}} \mid \boldsymbol{x}_0) = \frac{\alpha_{t_{i-1}}^2}{\alpha_{t_i}^2} \mathrm{Var}(\tilde{\boldsymbol{x}}_{t_i} \mid \boldsymbol{x}_0). \quad (45)$$

For simplicity, we denote $\mathrm{Var}\,(\tilde{\boldsymbol{x}}_{t_i} \mid \boldsymbol{x}_0)$ as $\mathrm{Var}\,(\tilde{\boldsymbol{x}}_{t_i})$ where appropriate. Based on monotonicity, $\frac{\sigma_{t_{i-1}}}{\sigma_{t_i}} < \frac{\alpha_{t_{i-1}}}{\alpha_{t_i}}$, as $\alpha_t$ is monotonically decreasing with respect to time $t$ and $\sigma_t$ is monotonically increasing with respect to time $t$. Therefore, $\mathrm{Var}\,(L_{\mathrm{data}}) < \mathrm{Var}\,(L_{\mathrm{noise}})$. Subsequently, we consider the variance of the non-linear terms for both iterations. For clarity, we denote $c(t_i, t_{i-1}) := \alpha_{t_i} \sigma_{t_{i-1}} - \alpha_{t_{i-1}} \sigma_{t_i}$. Then,

$$\sigma_{t_{i-1}} \left( \frac{\alpha_{t_{i-1}}}{\sigma_{t_{i-1}}} - \frac{\alpha_{t_i}}{\sigma_{t_i}} \right) = \frac{-1}{\sigma_{t_i}} c(t_i, t_{i-1}), \quad \alpha_{t_{i-1}} \left( \frac{\sigma_{t_{i-1}}}{\alpha_{t_{i-1}}} - \frac{\sigma_{t_i}}{\alpha_{t_i}} \right) = \frac{1}{\alpha_{t_i}} c(t_i, t_{i-1}). \quad (46)$$

Thus, the variances of the nonlinear terms for the two different iterations are, respectively, as follows:

$$\mathrm{Var}(N_{\mathrm{noise}}) = \frac{c^2(t_i, t_{i-1})}{\alpha_{t_i}^2} \cdot \mathrm{Var}\,(\boldsymbol{\epsilon}_\theta\,(\tilde{\boldsymbol{x}}_{t_i}, t_i)), \; \mathrm{Var}(N_{\mathrm{data}}) = \frac{(-c(t_i, t_{i-1}))^2}{\sigma_{t_i}^2} \cdot \mathrm{Var}\,(\boldsymbol{x}_\theta\,(\tilde{\boldsymbol{x}}_{t_i}, t_i)). \quad (47)$$

Note that

$$\mathrm{Var}\,(\boldsymbol{x}_\theta\,(\tilde{\boldsymbol{x}}_{t_i}, t_i)) = \mathrm{Var}\left( \frac{\tilde{\boldsymbol{x}}_{t_i} - \sigma_{t_i} \boldsymbol{\epsilon}_\theta\,(\tilde{\boldsymbol{x}}_{t_i}, t_i)}{\alpha_{t_i}} \right) = \frac{\sigma_{t_i}^2}{\alpha_{t_i}^2} \mathrm{Var}\,(\boldsymbol{\epsilon}_{t_i} - \boldsymbol{\epsilon}_\theta\,(\tilde{\boldsymbol{x}}_{t_i}, t_i)). \quad (48)$$

as $\tilde{\boldsymbol{x}}_{t_i} = \alpha_{t_i} \boldsymbol{x}_0 + \sigma_{t_i} \boldsymbol{\epsilon}_{t_i}$. Then,

$$\mathrm{Var}(N_{\mathrm{data}}) = \frac{(-c(t_i, t_{i-1}))^2}{\sigma_{t_i}^2} \cdot \frac{\sigma_{t_i}^2}{\alpha_{t_i}^2} \mathrm{Var}\,(\boldsymbol{\epsilon}_{t_i} - \boldsymbol{\epsilon}_\theta\,(\tilde{\boldsymbol{x}}_{t_i}, t_i)) = \frac{c^2(t_i, t_{i-1})}{\alpha_{t_i}^2} \mathrm{Var}\,(\boldsymbol{\epsilon}_\theta\,(\tilde{\boldsymbol{x}}_{t_i}, t_i) - \boldsymbol{\epsilon}_{t_i}). \quad (49)$$

Clearly, since $\boldsymbol{\epsilon}_\theta\,(\tilde{\boldsymbol{x}}_{t_i}, t_i)$ is designed to predict the injected noise into the clean data at time step $t_i$, and based on Eq. (2), the variance $\mathrm{Var}\,(\boldsymbol{\epsilon}_\theta\,(\tilde{\boldsymbol{x}}_{t_i}, t_i) - \boldsymbol{\epsilon}_{t_i})$ can theoretically approach arbitrarily small values as the accuracy of the model's estimation improves. Therefore, as $\mathrm{Var}\,(\boldsymbol{\epsilon}_\theta\,(\tilde{\boldsymbol{x}}_{t_i}, t_i) - \boldsymbol{\epsilon}_{t_i}) < \mathrm{Var}\,(\boldsymbol{\epsilon}_\theta\,(\tilde{\boldsymbol{x}}_{t_i}, t_i))$, we have $\mathrm{Var}(N_{\mathrm{data}}) < \mathrm{Var}(N_{\mathrm{niose}})$. Since the randomness of the linear term is independent of that of the nonlinear term in both iterations, and given that $\mathrm{Var}\,(L_{\mathrm{data}}) < \mathrm{Var}\,(L_{\mathrm{noise}})$ and $\mathrm{Var}(N_{\mathrm{data}}) < \mathrm{Var}(N_{\mathrm{niose}})$, we have

$$\begin{aligned} 0 \leq \mathrm{Var}(p_1(\tilde{\boldsymbol{x}}_{t_{i-1}} \mid \boldsymbol{x}_0)) &= \mathrm{Var}(L_{\mathrm{data}}) + \mathrm{Var}(N_{\mathrm{data}}) \\ &< \mathrm{Var}(L_{\mathrm{noise}}) + \mathrm{Var}(N_{\mathrm{noise}}) = \mathrm{Var}(p_2(\tilde{\boldsymbol{x}}_{t_{i-1}} \mid \boldsymbol{x}_0)). \end{aligned} \quad (50)$$

Consequently, based on Eq. (9), which provides the conditional entropy formula for a Gaussian distribution, we have $H_{p_1}(\tilde{\boldsymbol{x}}_{t_{i-1}} \mid \boldsymbol{x}_0) < H_{p_2}(\tilde{\boldsymbol{x}}_{t_{i-1}} \mid \boldsymbol{x}_0)$. The proof is complete. $\qquad\square$

### C.6    Single-step Analysis

For single-step iteration, one insight is that the model parameter $\boldsymbol{\epsilon}_\theta\,(\tilde{\boldsymbol{x}}_{s_i}, s_i)$ can be used further to improve the iteration governed by Eq. (7), without additional model parameters. Formally, we can formulate the iteration as

$$\boldsymbol{f}(\tilde{\boldsymbol{x}}_{t_{i-1}}) = \boldsymbol{f}(\tilde{\boldsymbol{x}}_{t_i}) + h_{t_i} G(\gamma_i) + \frac{h_{t_i}^2}{2} F_\theta(s_i, t_i), \quad (51)$$

where $G(\gamma_i) = \gamma_i \boldsymbol{\epsilon}_\theta\,(\tilde{\boldsymbol{x}}_{s_i}, s_i) + \bar{\gamma}_i \boldsymbol{\epsilon}_\theta\,(\tilde{\boldsymbol{x}}_{t_i}, t_i)$, $\bar{\gamma}_i = 1 - \gamma_i$, $\gamma_i \in (0, 1]$. This improved iteration shares the same limit state as the vanilla iteration in Eq. (7) when $s_i \to t_i$. For convenience, we refer to the vanilla iteration as the *FD-based* single iteration. For clarity, we identify the denoising iteration by **r**educing conditional **e**ntropy as the *RE-based* single iteration.

In the analysis of conditional entropy, we can compare the different components of Eq. (7) and Eq. (51). Then, the variance of the key distinct components in each conditional distribution is as follows:

$$\mathrm{Var}_{p_1} = h_{t_i}^2 \cdot \mathrm{Var}(\boldsymbol{\epsilon}_\theta(\tilde{\boldsymbol{x}}_{t_i}, t_i)),$$
$$\mathrm{Var}_{p_2}(\gamma_i) = \gamma_i^2 h_{t_i}^2 \mathrm{Var}(\boldsymbol{\epsilon}_\theta(\tilde{\boldsymbol{x}}_{s_i}, s_i)) + \bar{\gamma}_i^2 \mathrm{Var}_{p_1},$$

(52)

where $\bar{\gamma}_i = 1 - \gamma_i$. Then, the difference in conditional entropy between two gradient estimation-based iterations is

$$\Delta H(p) = \frac{1}{2} \log \frac{\mathrm{Var}_{p_2}(\gamma_i)}{\mathrm{Var}_{p_1}} = \frac{1}{2} \log \left(1 + v(\gamma_i)\right).$$

(53)

where $v(\gamma_i) = -2\gamma_i + \gamma_i^2 + \gamma_i^2 \frac{\mathrm{Var}(\boldsymbol{\epsilon}_\theta(\tilde{\boldsymbol{x}}_{s_i}, s_i))}{\mathrm{Var}(\boldsymbol{\epsilon}_\theta(\tilde{\boldsymbol{x}}_{t_i}, t_i))}$. Due to $\gamma_i \in (0, 1]$ and $\mathrm{SNR}(t_i) \leq \mathrm{SNR}(s_i)$, $\Delta H(p) \leq 0$ consistently holds under the assumption that $\mathrm{Var}(\boldsymbol{\epsilon}_\theta(\tilde{\boldsymbol{x}}_t, t)) \propto \sigma_t^2 / \alpha_t^2$. Therefore, this improved iteration can more efficiently reduce conditional entropy compared to the vanilla iteration by using subsequent model parameters in lower-variance regions as guidance. Consequently, based on $\Delta H(p) \leq 0$, we have the following Remark.

*Remark* C.4. The RE-based single-step iteration specified in Eq. (51) consistently achieves a more efficient reduction in conditional entropy than the FD-based iteration.

Accordingly, Remark C.4 show that the RE-based iteration can consistently surpass the FD-based iteration in reducing conditional entropy.

## D  Proofs for the EVODiff Optimization Framework in Section 4.2

### D.1  Assumption

**Assumption 1**: The total derivative $\boldsymbol{d}_\theta^{(k)}\left(\boldsymbol{x}_{\psi(\tau)}, \psi(\tau)\right) := \frac{\mathrm{d}^k \boldsymbol{d}_\theta\left(\boldsymbol{x}_{\psi(\tau)}, \psi(\tau)\right)}{\mathrm{d}\,\tau^k}$ exists and is continuous if necessary, where $k$ is determined by the specific context.

**Assumption 2**: The function $\boldsymbol{d}_\theta\left(\boldsymbol{x}_{\psi(\tau)}, \psi(\tau)\right)$ is Lipschitz w.r.t. to its first parameter $\boldsymbol{x}_{\psi(\tau)}$.

### D.2  Proof of Theorem 4.2

*Proof.* Denotes $\hat{\boldsymbol{x}}_t = \boldsymbol{f}(\tilde{\boldsymbol{x}}_t)$ for short. Without loss of generality, the RE-based multi-step iteration described in Eq. (15) can be decomposed into:

$$\hat{\boldsymbol{x}}_\mu = \hat{\boldsymbol{x}}_{t_i} + h_{t_i} \boldsymbol{d}_\theta\left(\tilde{\boldsymbol{x}}_{t_i}, t_i\right) + \frac{h_{t_i}^2}{2} B_\theta(s_i, t_i),$$

and

$$\hat{\boldsymbol{x}}_{t_{i-1}} = \hat{\boldsymbol{x}}_\mu + \gamma h_{t_i} \left(\boldsymbol{d}_\theta\left(\tilde{\boldsymbol{x}}_{s_i}, s_i\right) - \boldsymbol{d}_\theta\left(\tilde{\boldsymbol{x}}_{t_i}, t_i\right)\right).$$

Clearly, $\hat{\boldsymbol{x}}_\mu = \hat{\boldsymbol{x}}_{t_i} + \mathcal{O}(h_{t_i}^3)$ based on the Taylor expansion. Since the model $\boldsymbol{d}_\theta(\tilde{\boldsymbol{x}}_t, t)$ satisfies the Lipschitz assumption with respect to $\tilde{\boldsymbol{x}}_t$, then

$$\|\hat{\boldsymbol{x}}_{t_{i-1}} - \hat{\boldsymbol{x}}_\mu\| = \|\gamma h_{t_i} \left(\boldsymbol{d}_\theta\left(\tilde{\boldsymbol{x}}_{s_i}, s_i\right) - \boldsymbol{d}_\theta\left(\tilde{\boldsymbol{x}}_{t_i}, t_i\right)\right)\|$$
$$= L_1 \hat{h}_{t_i} \|\boldsymbol{d}_\theta\left(\tilde{\boldsymbol{x}}_{s_i}, s_i\right) - \boldsymbol{d}_\theta\left(\tilde{\boldsymbol{x}}_{t_i}, t_i\right)\|$$
$$\leq L_2 \hat{h}_{t_i} \|\tilde{\boldsymbol{x}}_{s_i} - \tilde{\boldsymbol{x}}_{t_i}\| = \mathcal{O}(|\hat{h}_{t_i}|^3).$$

(54)

Subsequently, by the triangle inequality, we have

$$\|\hat{\boldsymbol{x}}_{t_{i-1}} - \hat{\boldsymbol{x}}_{t_i}\| = \|\hat{\boldsymbol{x}}_{t_{i-1}} - \hat{\boldsymbol{x}}_\mu + \hat{\boldsymbol{x}}_\mu - \hat{\boldsymbol{x}}_{t_i}\| \leq \|\hat{\boldsymbol{x}}_{t_{i-1}} - \hat{\boldsymbol{x}}_\mu\| + \|\hat{\boldsymbol{x}}_\mu - \hat{\boldsymbol{x}}_{t_i}\| = \mathcal{O}(|h_{t_i}|^3), \quad (55)$$

where the last equality holds because $|h_{t_i}| \geq |\hat{h}_{t_i}|$.

Therefore, we prove that the local error of the RD-based iteration is of the same order as the corresponding Taylor expansion. Consequently, the RE-based iteration in Eq. 51 is a second-order convergence algorithm. The proof is complete. $\qquad\square$

## D.3 Proofs of Lemma 4.4 and Lemma 4.5

*Proof.* Without loss of generality,

$$\frac{\partial \left\| A - \sigma h \left( \lambda F_1 + (1-\lambda) F_2 \right) \right\|_F^2}{\partial \lambda}$$

$$= \frac{\partial \left( \mathrm{vec}^\top \left( A - \sigma h \left( \lambda F_1 + (1-\lambda) F_2 \right) \right) \mathrm{vec} \left( A - \sigma h \left( \lambda F_1 + (1-\lambda) F_2 \right) \right) \right)}{\partial \lambda}$$

$$= 2 \, \mathrm{vec}^\top \left( \frac{\partial \left( A - \sigma h \left( \lambda F_1 + (1-\lambda) F_2 \right) \right)}{\partial \lambda} \right) \mathrm{vec} \left( A - \sigma h \left( \lambda F_1 + (1-\lambda) F_2 \right) \right)$$

$$= 2 \, \mathrm{vec}^\top \left( -\sigma h \left( F_1 - F_2 \right) \right) \mathrm{vec} \left( A - \sigma h \left( \lambda F_1 + (1-\lambda) F_2 \right) \right)$$

Let $\frac{\partial \| A - \sigma h (\lambda F_1 + (1-\lambda) F_2) \|_F^2}{\partial \lambda} = 0$, we have

$$\mathrm{vec}^\top \left( F_1 - F_2 \right) \mathrm{vec} \left( \sigma h \lambda \left( F_1 - F_2 \right) - \left( A - \sigma h F_2 \right) \right) = 0. \tag{56}$$

Therefore,

$$\lambda = \frac{\mathrm{vec}^\top \left( F_1 - F_2 \right) \mathrm{vec} \left( A - \sigma h F_2 \right)}{\sigma h \, \mathrm{vec}^\top \left( F_1 - F_2 \right) \mathrm{Vec} \left( F_1 - F_2 \right)}. \tag{57}$$

The proof is complete. □

## D.4 Proof of Theorem 4.6

Let us review the EVODiff iteration, without loss of generality, in Algorithm 1 as follows:

$$\boldsymbol{f}(\tilde{\boldsymbol{x}}_{t_{i-1}}) = \boldsymbol{f}(\tilde{\boldsymbol{x}}_{t_i}) + h_{t_i} \boldsymbol{x}_\theta \left( \tilde{\boldsymbol{x}}_{t_i}, t_i \right) + \frac{h_{t_i}^2}{2} \zeta_i B_\theta(t_i)$$

$$= \boldsymbol{f}(\tilde{\boldsymbol{x}}_{t_i}) + h_{t_i} \boldsymbol{x}_\theta \left( \tilde{\boldsymbol{x}}_{t_i}, t_i \right) + \frac{h_{t_i}^2}{2} \zeta_i \left( \frac{\eta_i}{2} B_\theta(s_i, t_i) + \left( 1 - \frac{\eta_i}{2} \right) B_\theta(t_i, l_i) \right)$$

$$= \boldsymbol{f}(\tilde{\boldsymbol{x}}_{t_i}) + h_{t_i} \left( \frac{\eta_i}{2} \boldsymbol{x}_\theta \left( \tilde{\boldsymbol{x}}_{t_i}, t_i \right) + \left( 1 - \frac{\eta_i}{2} \right) \boldsymbol{x}_\theta \left( \tilde{\boldsymbol{x}}_{t_i}, t_i \right) \right)$$

$$+ \frac{h_{t_i}^2}{2} \zeta_i \left( \frac{\eta_i}{2} B_\theta(s_i, t_i) + \left( 1 - \frac{\eta_i}{2} \right) B_\theta(t_i, l_i) \right),$$

where $B_\theta(t_i, t_{i+1}) = \frac{\boldsymbol{x}_\theta \left( \tilde{\boldsymbol{x}}_{t_i}, t_i \right) - \boldsymbol{x}_\theta \left( \tilde{\boldsymbol{x}}_{t_{i+1}}, t_{i+1} \right)}{h_{t_{i+1}}}$.

In the following, we now proof the convergence properties of this EVODiff iteration scheme and establish its convergence order.

*Proof.* Denote $\hat{\boldsymbol{x}}_t = \boldsymbol{f}(\tilde{\boldsymbol{x}}_t)$ for short. The RE-based iteration in EVODiff 1 can be decomposed as:

$$\hat{\boldsymbol{x}}_{t_{i-1}} = \hat{\boldsymbol{x}}_{t_i} + \frac{\eta_i}{2} \hat{\boldsymbol{x}}_{\mu_1} + \left( 1 - \frac{\eta_i}{2} \right) \hat{\boldsymbol{x}}_{\mu_2}$$

$$= \frac{\eta_i}{2} \left( \hat{\boldsymbol{x}}_{t_i} + \hat{\boldsymbol{x}}_{\mu_1} \right) + \left( 1 - \frac{\eta_i}{2} \right) \left( \hat{\boldsymbol{x}}_{t_i} + \hat{\boldsymbol{x}}_{\mu_2} \right),$$

where

$$\hat{\boldsymbol{x}}_{\mu_1} = h_{t_i} \boldsymbol{x}_\theta(\tilde{\boldsymbol{x}}_{t_i}, t_i) + \frac{h_{t_i}^2}{2} \zeta_i B_\theta(s_i, t_i), \quad \hat{\boldsymbol{x}}_{\mu_2} = h_{t_i} \boldsymbol{x}_\theta(\tilde{\boldsymbol{x}}_{t_i}, t_i) + \frac{h_{t_i}^2}{2} \zeta_i B_\theta(t_i, l_i).$$

Let us now consider the case of $\hat{\boldsymbol{x}}_{t_i} + \hat{\boldsymbol{x}}_{\mu_1}$. Denote

$$\hat{\boldsymbol{x}}_{1, t_{i-1}} = \hat{\boldsymbol{x}}_{t_i} + \hat{\boldsymbol{x}}_{\mu_1}, \quad \hat{\boldsymbol{x}}_{\mu_3} = \hat{\boldsymbol{x}}_{t_i} + h_{t_i} \boldsymbol{x}_\theta(\tilde{\boldsymbol{x}}_{t_i}, t_i) + \frac{h_{t_i}^2}{2} B_\theta(s_i, t_i).$$

Then $\hat{\boldsymbol{x}}_{1, t_{i-1}} = \hat{\boldsymbol{x}}_{\mu_3} + (\zeta_i - 1) \frac{h_{t_i}^2}{2} B_\theta(s_i, t_i)$. Note that $\hat{\boldsymbol{x}}_{\mu_3} = \hat{\boldsymbol{x}}_{t_i} + \mathcal{O}(h_{t_i}^3)$ and $B_\theta(s_i, t_i) = \mathcal{O}(h_{t_i})$ based on the Taylor expansion. Therefore, we have

$$\| \hat{\boldsymbol{x}}_{1, t_{i-1}} - \hat{\boldsymbol{x}}_{t_i} \| = \left\| \hat{\boldsymbol{x}}_{\mu_3} - \hat{\boldsymbol{x}}_{t_i} + \frac{\zeta_i - 1}{2} h_{t_i}^2 B_\theta(s_i, t_i) \right\|$$

$$\leq \| \hat{\boldsymbol{x}}_{\mu_3} - \hat{\boldsymbol{x}}_{t_i} \| + \left\| \frac{\zeta_i - 1}{2} h_{t_i}^2 B_\theta(s_i, t_i) \right\| \tag{58}$$

$$= \mathcal{O}(h_{t_i}^3) + L_1 \mathcal{O}(h_{t_i}^3) = \mathcal{O}(h_{t_i}^3),$$

where $L_1$ is a constant because $\zeta_i$ can be bounded by 1. Denote $\hat{\boldsymbol{x}}_{2,t_{i-1}} = \hat{\boldsymbol{x}}_{t_i} + \hat{\boldsymbol{x}}_{\mu_2}$. Symmetrically, we obtain

$$\|\hat{\boldsymbol{x}}_{2,t_{i-1}} - \hat{\boldsymbol{x}}_{t_i}\| = \mathcal{O}(h_{t_i}^3). \tag{59}$$

Now, combining the results, we obtain

$$\begin{aligned}
\|\hat{\boldsymbol{x}}_{t_{i-1}} - \hat{\boldsymbol{x}}_{t_i}\| &= \left\| \frac{\eta_i}{2} \left( \hat{\boldsymbol{x}}_{1,t_{i-1}} - \hat{\boldsymbol{x}}_{t_i} \right) + \left( 1 - \frac{\eta_i}{2} \right) \left( \hat{\boldsymbol{x}}_{2,t_{i-1}} - \hat{\boldsymbol{x}}_{t_i} \right) \right\| \\
&\leq \frac{\eta_i}{2} \|\hat{\boldsymbol{x}}_{1,t_{i-1}} - \hat{\boldsymbol{x}}_{t_i}\| + \left( 1 - \frac{\eta_i}{2} \right) \|\hat{\boldsymbol{x}}_{2,t_{i-1}} - \hat{\boldsymbol{x}}_{t_i}\| \\
&= \frac{\eta_i}{2} \mathcal{O}(h_{t_i}^3) + \left( 1 - \frac{\eta_i}{2} \right) \mathcal{O}(h_{t_i}^3) = \mathcal{O}(h_{t_i}^3).
\end{aligned} \tag{60}$$

Thus, we have shown that the local error of the RE-based iteration in EVODiff 1 is $\mathcal{O}(h_{t_i}^3)$. Consequently, the RE-based iteration in EVODiff 1 achieves second-order global convergence. The proof is complete. $\qquad\square$

# E  Experiment Details

In our experiments, we utilize several standard pre-trained models. Specifically, we employ the discrete denoising diffusion probabilistic model [2], the continuous score-based model [3], and the uncond EDM model [12], all trained on CIFAR-10 [75]. For larger-scale evaluations on high-dimensional data, we adopt the pre-trained models trained on the ImageNet dataset [76] from the baseline method [4]. Additionally, we use the pre-trained Latent Diffusion Model and Stable Diffusion model [28], where the latter is trained on the LAION-5B dataset [77] using CLIP [78] text embeddings as conditioning signals.

## E.1  Experimental Computational Resources and Data

All experiments were conducted on NVIDIA GPUs. For high-dimensional datasets like ImageNet, we utilized the NVIDIA GeForce RTX 3090 GPU with 24GB VRAM. For other cases like CIFAR-10, experiments were performed on NVIDIA TITAN X (Pascal) with 12GB VRAM. To ensure a fair comparison with prior work, we maintained consistent pre-trained models and experimental settings across both scenarios. We list some of the datasets and codes used in Table 7.

Table 7: Some of the datasets and codes used.

| Name | URL |
|---|---|
| CIFAR10 | `https://www.cs.toronto.edu/ kriz/cifar.html` |
| LSUN-Bedroom | `https://www.yf.io/p/lsun` |
| ImageNet-256×256 | `https://www.image-net.org` |
| ScoreSDE | `https://github.com/yang-song/score_sde_pytorch` |
| EDM | `https://github.com/NVlabs/edm` |
| Guided-Diffusion | `https://github.com/openai/guided-diffusion` |
| Latent-Diffusion | `https://github.com/CompVis/latent-diffusion` |
| Stable-Diffusion | `https://github.com/CompVis/stable-diffusion` |
| DPM-Solver | `https://github.com/LuChengTHU/dpm-solver` |
| DPM-Solver++ | `https://github.com/LuChengTHU/dpm-solver` |
| SciRE-Solver | `https://github.com/ShiguiLi/SciRE-Solver` |
| UniPC | `https://github.com/wl-zhao/UniPC` |
| DPM-Solver-v3 | `https://github.com/thu-ml/DPM-Solver-v3` |

## E.2  Sampling Schedules

Sampling schedules in DMs define how the noise scale evolves during inference and play a crucial role in balancing sample quality and computational efficiency. Several widely used schedules include the Time-uniform schedule [2, 3], the LogSNR schedule [13], and the EDM schedule [12]. Although optimized schedules have been proposed [68, 69], they typically require significant computational resources for optimization. In our experiments, we follow the default schedule of the baseline methods.

### E.3 Parameterization Settings of the Sampling Process

In the sampling process of DMs, various parameterization settings are used to define the target prediction at each iteration step. Below, we list the adopted parameterizations:

*Noise prediction parameterization* [2]: This parameterization directly predicts the noise injected during the forward diffusion process. The connection to the score function is formalized as:

$$\epsilon_\theta\left(\boldsymbol{x}_t, t\right) = -\sigma_t \nabla_{\boldsymbol{x}} \log q\left(\boldsymbol{x}_t\right), \tag{61}$$

where $\nabla_{\boldsymbol{x}} \log q\left(\boldsymbol{x}_t\right)$ denotes the score function [3].

*Data prediction parameterization* [21]: This parameterization estimates the clean data $\boldsymbol{x}_0$ from the noisy input $\boldsymbol{x}_t$ at a given time step $t$. The predicted data satisfies:

$$\boldsymbol{x}_\theta(\boldsymbol{x}_t, t) = \frac{\boldsymbol{x}_t - \sigma_t \epsilon_\theta(\boldsymbol{x}_t, t)}{\alpha_t}. \tag{62}$$

Although these parameterizations have practical predictive value, they may insufficiently minimize discretization errors. Building upon earlier DPM-Solver versions [13, 20], DPM-Solver-v3 [17] extends the parameterization strategy by incorporating empirical model statistics (EMS). This is an approach that requires a reference solution. Essentially, leveraging prior information about the target distribution optimizes both the variance and bias terms in the reconstruction error, as decomposed in Eq. (3.1). Specifically, they formulated the continuous-time ODE as follows:

$$\frac{\mathrm{d}\boldsymbol{x}_\lambda}{\mathrm{d}\lambda} = \left(\frac{\dot{\alpha}_\lambda}{\alpha_\lambda} - \boldsymbol{l}_\lambda\right) \boldsymbol{x}_\lambda - \left(\sigma_\lambda \epsilon_\theta(\boldsymbol{x}_\lambda, \lambda) - \boldsymbol{l}_\lambda \boldsymbol{x}_\lambda\right), \tag{63}$$

where $\lambda$ represents the continuous-time parameter, and $\boldsymbol{l}_\lambda$ is an optimized prior statistics term.

In our ablation study, we employ the default parameterization of the baseline method in all of our experiments. It is important to note that our main baseline is DPM-Solver++ [20], while the other parameterizations are only auxiliary setups intended to validate the effectiveness of the variance-driven optimization.

### E.4 Evaluating Sampling Efficiency and Image Quality in Generative Models

The *Fréchet Inception Distance (FID)* [79] evaluates the quality and diversity of generated images by comparing the statistical distributions of generated and real images in a feature space. It uses a pre-trained Inception-v3 network to extract features [80], computing the mean $\mu$ and covariance $\Sigma$ for both distributions. Specifically, $\mu_g$ and $\Sigma_g$ represent the mean and covariance of features from generated images, while $\mu_r$ and $\Sigma_r$ correspond to real images. Specifically, FID is calculated as:

$$\text{FID} = \|\mu_g - \mu_r\|^2 + \text{Tr}\left(\Sigma_g + \Sigma_r - 2(\Sigma_g \cdot \Sigma_r)^{1/2}\right). \tag{64}$$

Lower FID values indicate higher similarity between generated and real distributions, reflecting better image quality [2–4].

The *Number of Function Evaluations (NFE)* measures computational efficiency by counting neural network function calls during sampling [3, 9, 26, 13, 12]. Lower NFE values indicate faster sampling.

Balancing FID and NFE is crucial for practical applications where both high-quality outputs and computational efficiency are required. Joint evaluation of these metrics provides a comprehensive perspective: FID assesses distribution fidelity, while NFE evaluates algorithmic efficiency.

In this paper, we adopt the evaluation framework used in prior studies [13, 20], combining FID and NFE to jointly assess the quality of generated images and the computational efficiency of sampling algorithms. This comprehensive approach, validated in several studies [4, 3, 13, 12], offers a standardized benchmark for comparing different generative models and sampling methods. Moreover, we adopt CLIP-Score [78], aesthetics score such as PickScore [81] and ImageReward [82] to estimate the quality of generated images using method on Stable-Diffusion [28].

### E.5 Conditional Sampling in DMs

Conditional sampling in DMs enables controlled generation by incorporating conditioning information (e.g., class labels or text) into the sampling process. This is achieved by modifying the noise predictor

$\epsilon_\theta(\boldsymbol{x}_t, t, c)$ to guide generation toward satisfying condition $c$. Two main approaches exist: *classifier-free guidance* [83] and *classifier guidance* [4]. Classifier-free guidance (CFG) combines conditional and unconditional predictions:

$$\epsilon_\theta^{\text{CFG}}(\boldsymbol{x}_t, t, c) := (1 + w)\epsilon_\theta(\boldsymbol{x}_t, t, c) - w\epsilon_\theta(\boldsymbol{x}_t, t, \emptyset), \qquad (65)$$

where $\emptyset$ denotes the unconditional case and $w > 0$ is the guidance scale. This method is simple and efficient as it requires no additional models.

Classifier guidance (CG) uses an auxiliary classifier $p_\phi(c \mid \boldsymbol{x}_t, t)$:

$$\epsilon_\theta^{\text{CG}}(\boldsymbol{x}_t, t, c) := \epsilon_\theta(\boldsymbol{x}_t, t) - s\sigma_t\nabla_{\boldsymbol{x}_t} \log p_\phi(c \mid \boldsymbol{x}_t, t), \qquad (66)$$

where $s$ controls guidance strength and $\sigma_t$ is the noise level at time $t$. While computationally more expensive, this approach can provide finer control over the conditioning process.

In our experiments, we adopt the default guidance approach of the baseline method.

### E.6 Single-step Iteration Details

Our goal is to validate that variance-driven conditional entropy reduction can improve the denoising diffusion process. Compared to iterations based on traditional truncated Taylor expansions, RE-based iterations achieve better sampling performance, as demonstrated in DPM-Solver. This is because DPM-Solver iterations represent a specific instantiation of RE-based iterations, as shown in Proposition 3.3. Nevertheless, through extensive experiments on CIFAR-10 [75], CelebA 64 [84], and ImageNet-256 [76], we validated that RE-based iterations can further improve the denoising diffusion process by minimizing conditional variance. In this validation experiment, we adopt DPM-Solver [13] as our baseline. *Since the single-step iteration mechanism only requires the information from the starting point to the information before the endpoint, RE-based iterations depend on prior variance assumptions to reduce the conditional variance between iterations.* Below, based on the principle of minimizing conditional variance, we demonstrate how to select parameters under the assumption of prior variance.

For clarity, we simplify the RE-based single-step iteration in Eq. (51) as follows:

$$\boldsymbol{f}(\tilde{\boldsymbol{x}}_{t_{i-1}}) = \boldsymbol{f}(\tilde{\boldsymbol{x}}_{t_i}) + h_{t_i}\left(\left(\gamma_i + \frac{r_i}{2}\right)\epsilon_\theta(\tilde{\boldsymbol{x}}_{s_i}, s_i) + \left(1 - \gamma_i - \frac{r_i}{2}\right)\epsilon_\theta(\tilde{\boldsymbol{x}}_{t_i}, t_i)\right), \qquad (67)$$

where $r_i = \frac{h_{t_i}}{\bar{h}_{t_i}}$. To reduce variance of iteration (67) in each step, we configure the parameter $\gamma_i$ in accordance with the effective variance reduction interval prescribed in Remark C.4. Based on Remark C.4, since $\gamma_i \in \left[\frac{\text{SNR}(t_i)}{\text{SNR}(t_i)+\text{SNR}(s_i)}, \frac{\max\{2\cdot\text{SNR}(t_i), \text{SNR}(s_i)\}}{\text{SNR}(t_i)+\text{SNR}(s_i)}\right]$, when considering only $\gamma_i$ in isolation, we recommend three specific selections of prior parameter $\gamma_i$: $\gamma_i = \frac{\text{SNR}(t_i)}{\text{SNR}(t_i)+\text{SNR}(s_i)}$ and $\gamma_i = \frac{1}{2}$. Due to $r_i \in \left[1, \frac{4\,\text{SNR}(s_i)}{\text{SNR}(t_i)+\text{SNR}(s_i)}\right]$ based on Remark C.1. Based on the proof in C.2, when considering only $r_i$ in isolation, a ponential optimal value for $r_i$ is given by $\frac{2\,\text{SNR}(s_i)}{\text{SNR}(t_i)+\text{SNR}(s_i)}$. We recommend three specific selections of prior parameter $r_i$: $r_i = 1$ and $r_i = \sqrt{\frac{2\,\text{SNR}(s_i)}{\text{SNR}(t_i)+\text{SNR}(s_i)}}$. Based on empirical performance, we recommend the combinations $\left(r_i = 1, \gamma_i = \frac{1}{2}\right)$ or $\left(r_i = \sqrt{\frac{2\,\text{SNR}(s_i)}{\text{SNR}(t_i)+\text{SNR}(s_i)}}, \gamma_i = \frac{\text{SNR}(t_i)}{\text{SNR}(t_i)+\text{SNR}(s_i)}\right)$ for balanced inference.

We compare the performance of RE-based iterations against several established solvers, including DDPM [2], Analytic-DDPM [26], DDIM [9], DPM-Solver [13], F-PNDM [11], and ERA-Solver [64]. The comparative results are presented in Figures 2 and Table 8. Remarkably, this consistent improvement in the conditional variance enhances image quality across various scenarios, as demonstrated by the ablation study with $\gamma_i = 1/2$ and $r_i = 1$ in Figures 2. Notably, compared to the 3.17 FID achieved by DDPM with 1000 NFEs [2] on CIFAR-10, our RE-based iteration achieves a *3.15* FID with only *84* NFE, establishing a new *SOTA* FID for this discrete-time pre-trained model while realizing approximately *10×* acceleration. A visual comparison is shown in Figure 6.

### E.7 Multi-step Iteration Details

In this section, we explore the potential of variance-based conditional entropy reduction to further enhance the denoising diffusion process. Unlike single-step mechanisms, multi-step iterations can

Table 8: The performance comparison of sampling methods on CIFAR-10 [75] suggests that RE-based iterations can further improve the denoising diffusion process with enhanced equality.

| Discrete | Continuous | Cond. EDM |
|---|---|---|
| 3.17 | 2.55 | 1.79 |
| DDPM | Hybrid PC | EDM |
| 3.26 | 2.64 | 1.79 |
| F-PNDM | DPM-Solver-v3 | Heun's 2nd |
| **3.15** | **2.41** | **1.76** |
| RE-based | RE-based | RE-based |

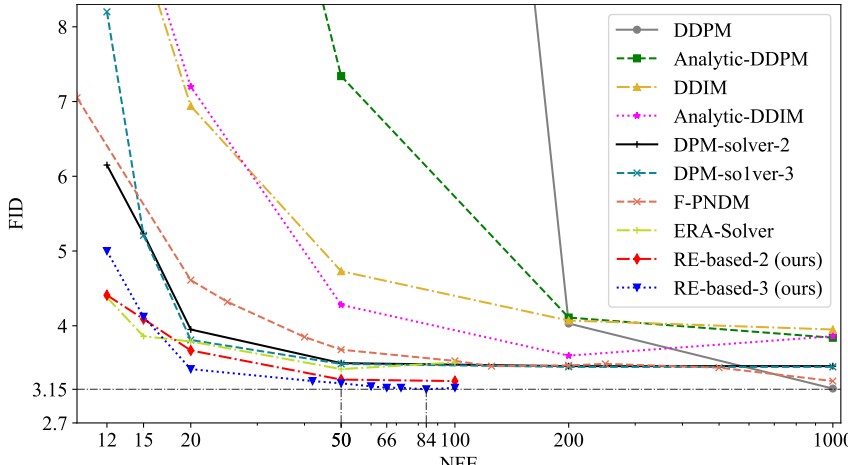

Figure 6: Comparisons of FID ↓ for different iterations on discrete DMs in CIFAR-10.

leverage information from previous steps, providing additional context. Building on this advantage, we propose a training-free and efficient denoising iteration framework aimed at improving the denoising diffusion process through variance-driven conditional entropy reduction. Specifically, the framework minimizes conditional variance by reducing the discrepancies between actual states during the denoising iterations.

**Challenge.** Formulating the optimization objective to achieve this goal presents a significant challenge, requiring a mechanism that can effectively capture subtle state variations across iterations. The key lies in developing an algorithm that can identify meaningful features from state differences and transform these insights into signals that improve the denoising process. This involves not only quantifying state differences but also understanding the underlying deep information patterns in these variations, enabling more precise control over the denoising diffusion process.

Our optimization objective is formulated by considering both the discrepancy between the actual data states and the variation in the gradient states. Building upon these foundations, we outline this efficient conditional entropy reduction iteration mechanism driven by variance minimization in EVODiff 1, which offers an effective means to integrate variance-driven conditional entropy reduction into the denoising diffusion process by minimizing actual state differences.

**Practical Considerations.** Our goal is to develop an iterative denoising sampling algorithm for pre-trained DMs that requires neither additional training nor costly optimization procedures. However, in the iterative scheme aimed at minimizing the variance-driven conditional entropy reduction, we need to optimize the key parameters $\zeta_i$ and $\eta_i$ that control the conditional variance of the denoising iteration. As discussed in the main text, to balance optimality and computational efficiency, we adopt an optimization-guided streamlined approach to obtain optimized variance-reduction control parameters $\zeta_i$ and $\eta_i$. Specifically, the original optimization problem was a standard constrained mathematical programming problem. We observed that the problem possesses a closed-form solution when constraints are removed. Therefore, to directly obtain the optimized parameters in one step, we choose to apply a nonlinear nonnegative mapping to this closed-form solution, using the mapped non-negative substitute as our final parameters. Since this nonnegative substitute solution has

Table 9: We conducted ablation experiments with different shift parameters in EVODiff 1, using the pre-trained model [4] on ImageNet-256×256 [76]. We report the FID ↓ evaluated on 10k samples for various NFEs and and guidance scales.

| Method | Guidance | Shift Parameter | NFE | | | | | | |
|--------|----------|-----------------|-----|-----|-----|-----|-----|-----|-----|
| | | | 5 | 6 | 8 | 10 | 12 | 15 | 20 |
| EVODiff | s=2 | $\mu = 0.25$ | **13.96** | **10.97** | 8.85 | 8.18 | 7.82 | 7.51 | 7.27 |
| | | $\mu = 0.50$ | 13.98 | 10.98 | 8.84 | 8.14 | **7.79** | **7.48** | **7.25** |
| | | $\mu = 0.75$ | 14.01 | 11.00 | **8.83** | **8.10** | 7.80 | 7.54 | 7.32 |
| EVODiff | s=3 | $\mu = 0.25$ | 14.43 | 11.08 | 8.90 | 8.30 | 7.92 | 7.58 | 7.53 |
| | | $\mu = 0.50$ | 14.37 | 11.04 | 8.87 | 8.31 | 7.89 | **7.56** | 7.51 |
| | | $\mu = 0.75$ | **14.32** | **10.99** | **8.85** | **8.23** | **7.87** | 7.56 | **7.50** |
| EVODiff | s=4 | $\mu = 0.25$ | 17.80 | 12.91 | 9.73 | 8.75 | 8.51 | **8.01** | **7.92** |
| | | $\mu = 0.50$ | 17.57 | 12.73 | 9.61 | 8.66 | **8.35** | 8.01 | 7.93 |
| | | $\mu = 0.75$ | **17.39** | **12.57** | **9.55** | **8.61** | 8.41 | 8.01 | 7.94 |

already achieved the objective of quantifying the differences between states, it can serve as an effective alternative for parameter optimization, simultaneously ensuring computational efficiency and preserving the capability to capture critical state variations.

### E.7.1 Ablation Study

**Parameter settings.** In our implementation, we primarily employ the sigmoid activation function, which is one of the most prevalent activation functions in neural networks [85]. Its mathematical expression is $\text{Sigmoid}(x) = \frac{1}{1+e^{-x}}$. In our experiments, the following improved version often yields better results, particularly for high-dimensional datasets: $\zeta_i = \text{Sigmoid}(-\frac{\sigma_{t_i}}{\sigma_{t_{i+1}}}(|\zeta_i^*| - \mu))$, where $\zeta_i^*$ is computed using Eq. (22) and $\mu$ is a shift parameter introduced to fine-tune the solution space.

Conceptually, $\mu$ serves as a dynamic sensitivity regulator, allowing for nuanced control over the transformation of the activation function. By adjusting $\mu$, the inflection point of the sigmoid function can be shifted, effectively modulating the model's responsiveness to input variations across different regions of the input space. For high-dimensional datasets, this provides a principled mechanism for adaptive sensitivity calibration. The shift parameter enables more precise capturing of subtle state variations by expanding or contracting the function's most sensitive transformation region.

Empirical results show that this approach achieves a judicious balance between computational efficiency and the model's ability to discern critical state transitions. Table 9 systematically examines the impact of shift parameters on image generation performance in pre-trained DMs, using comprehensive ablation experiments on the ImageNet-256×256 dataset. Key findings include:

- *Global Performance Characteristics:* A consistent downward trend is observed in FID scores as NFE increases, indicating a progressive refinement of sample quality. Performance differences among the tested shift parameters $\mu \in \{0.25, 0.50, 0.75\}$ remain marginal, reflecting the *robustness* of the sampling process across configurations.

- *Shift Parameter Behavior Across NFE Stages:* Performance variations exhibit nuanced characteristics:
  - At lower NFE stages, performance differences between $\mu$ values are more pronounced.
  - As NFE increases, the performance of different $\mu$ values converges.
  - Different $\mu$ values exhibit unique progression patterns at various guidance scales, despite only marginal differences.

- *Impact of Guidance Scale:* The sensitivity to shift parameters varies with guidance scales:
  - Lower guidance scales (e.g., s=2) slightly more pronounced performance variations with $\mu$, with a change magnitude of 0.05 FID at 20 NFE.
  - As guidance scale increases (to s=3 and s=4), the influence of shift parameters becomes subtler and more stable, with a change magnitude of $0.02 \sim 0.03$ FID at 20 NFE.

Beyond pixel-space DMs, we also conducted an ablation study on $\mu$ within *latent-space DMs* to verify its efficacy and robustness in more computationally efficient frameworks. Specifically, we

Table 10: Ablation Study: Effect of the $\mu$ Shift Parameter on EVODiff Performance for the latent-space diffusion model [28] (LSUN-Bedrooms dataset [86]).

| Method | Model | Dataset | NFE | $\mu$ | FID | Relative to $\mu = 0.5$ |
|--------|-------|---------|-----|-------|-----|--------------------------|
| EVODiff | Latent Diffusion | LSUN-Bedrooms | 5 | 0.25 | **7.6328** | +3.5% |
| | | | | 0.50 | 7.912 | baseline |
| | | | | 0.75 | 8.1845 | -3.4% |
| | | | 10 | 0.25 | 3.3357 | -0.1% |
| | | | | 0.50 | **3.3318** | baseline |
| | | | | 0.75 | 3.3409 | -0.3% |
| | | | 20 | 0.25 | **2.8369** | +0.6% |
| | | | | 0.50 | 2.8534 | baseline |
| | | | | 0.75 | 2.8728 | -0.7% |

used a latent diffusion model trained on the LSUN-Bedrooms dataset. As presented in Table 10, the observations across NFE stages remain largely consistent with the ImageNet findings, but reveal specific trends for latent space:

- Low-NFE Sensitivity: At the lowest 5 NFE, the shift parameter $\mu$ exhibits the largest influence. $\mu = 0.25$ yields the best FID score of 7.6328 (+3.5% relative to the baseline). This supports the notion that $\mu$ is most critical during the early, high-variance sampling phase.

- Robust Convergence: As NFE increases (from 5 to 10 and 20), performance differences across $\mu$ values shrink significantly, confirming the robustness of EVODiff across parameter settings. The $\mu = 0.50$ baseline performs optimally at NFE=10, while $\mu = 0.25$ is marginally best at 20 NFE, with the total variation across all $\mu$ being minimal ($\approx 0.7\%$).

- Conclusion: The LSUN-Bedrooms results validate the role of $\mu$ as an effective fine-tuning mechanism that introduces negligible instability, even when applied to the complex latent space of high-resolution image generation.

In summary, the extensive ablation studies on both pixel-space (ImageNet-256×256) and latent-space (LSUN-Bedrooms) diffusion models validate the function of the $\mu$ shift parameter. The results demonstrate the fundamental robustness of EVODiff across diverse configurations, showing only marginal performance variations across $\mu$ values in high-NFE scenarios. Critically, $\mu$ functions as an effective adaptive fine-tuning mechanism, providing the most significant benefit in the low-NFE, high-entropy sampling phase (e.g., $\mu = 0.25$ leading the performance at 5 NFE in the LSUN-Bedrooms study). This confirms that $\mu$ introduces negligible instability while offering a refined tool for sensitivity calibration in both high-dimensional pixel and latent spaces. Moreover, the properties of the aforementioned shift parameters collectively ensure the convergence and distinctiveness of our variance-driven conditional entropy reduction iterative scheme during the sampling process. Specifically, although these subtle variations are negligible on ImageNet-256×256, their distinctiveness is substantiated through experimental validation on the stable diffusion model, as shown in Figure 11.

**Reducing Conditional Variance with Prior $r_i$.** In multi-step iterations, we require a probing step (an iteration step of Single-step Iteration Framework) to obtain the model value at the next state. Reducing conditional variance is crucial for improving the stability and accuracy of iterative algorithms; thus, we need to balance the conditional variance of the gradient term and the first-order term (see the above Conditional Variance Analysis part). We found that while logSNR typically performs well with larger step sizes, its advantages diminish as the NFE increases, as illustrated in Figure 2 and Table 11. For clarity, we revisit the logSNR as follows:

$$r_{\text{logSNR}}(t) = \frac{\log \frac{\alpha_t}{\sigma_t} - \log \frac{\alpha_{t+1}}{\sigma_{t+1}}}{\log \frac{\alpha_{t-1}}{\sigma_{t-1}} - \log \frac{\alpha_t}{\sigma_t}}. \tag{68}$$

This balance concept of logSNR leads to two potentially useful types of substitutions.

From the perspective of balancing variances, one might consider the following form:

$$r_{\text{normvar}}(t) = \left( \frac{\text{Var}_{t+1} - \text{Var}_t}{\text{Var}_{t+1}} \right) \Big/ \left( \frac{\text{Var}_t - \text{Var}_{t-1}}{\text{Var}_t} \right), \tag{69}$$

where $\mathrm{Var}_t$ can represent any assumed variance, and satisfies $\mathrm{Var}_t > \mathrm{Var}_{t-1}$. If $\mathrm{Var}_t < \mathrm{Var}_{t-1}$, then simply swapping the roles of $\mathrm{Var}_t$ and $\mathrm{Var}_{t-1}$ in Eq. (69) will suffice. Another substitution idea is to change the function space of the step size, for example, to the arctangent space:

$$r_{\mathrm{arctan}}(t) = \frac{\arctan(h_t)}{\arctan(h_{t-1})}, \tag{70}$$

where $h_t$ denotes the step size from $t+1$ to $t$.

In our experiments, we observed that a nonlinear combination of these two substitutions leads to improvements in certain scenarios. We define this nonlinear combination as *refined* $r_i$, and the ablation study of both logSNR and refined $r_i$ in the context of EVODiff 1 can be found in Table 11. Table 11 shows that even when using the same $r_i$ as the baseline, our mathematically principled construction of EVODiff consistently outperforms state-of-the-art ODE solvers. Moreover, Table 11 also demonstrates that employing a more effective $r_i$ within our EVODiff framework further improves performance. Beyond this, we investigate a variance-driven approach that adheres more closely to theoretical principles. Specifically, $r_i = r_{\mathrm{logSNR}}(t) * w_{confidence}$, where $w_{\mathrm{confidence}}$ is a function of the cosine similarity between $B_\theta(t_i, l_i)$ and $\tilde{x}_{t_i}$ at time step $t_i$. This strategy demonstrates strong performance on both CIFAR-10 and ImageNet-256. As shown in Table 2, it yields significant improvements on CIFAR-10.

In summary, the ablation study on the shift parameter $\mu$ demonstrates the robustness of EVODiff. While different $\mu$ values show minor performance variations in specific NFE ranges, the overall results are consistently state-of-the-art, indicating that our method is not highly sensitive to this parameter and can achieve excellent performance with a default setting (e.g., $\mu = 0.5$).

### E.8    Comparison of Reference-Free EVODiff and Learning-Based Methods with Reference Trajectories

A significant advantage of EVODiff is its reference-free nature, enabling it to achieve state-of-the-art performance without the overhead required by methods that rely on pre-computed or learned reference trajectories. This section substantiates this claim by presenting comprehensive comparisons against major classes of reference-based methods, followed by analyses of computational efficiency and the generalizability of our core principles.

**Superiority over Reference-Based Solvers and Learning-Based Methods.**    Our advantage is particularly pronounced when benchmarked against advanced ODE solvers that explicitly incorporate reference information. As noted in our main results in Table 2, DPM-Solver-v3 leverages Empirical Model Statistics (EMS), which is a technique requiring prior knowledge from a high-NFE reference solution to optimize its steps. This essentially provides the solver with a "cheat sheet" on the data distribution. Despite this additional optimization information, EVODiff, with its on-the-fly adaptive strategy, consistently demonstrates superior performance. On CIFAR-10, it achieves a remarkable FID of *3.98* at 8 NFE and *2.78* at 10 NFE, decisively outperforming DPM-Solver-v3's scores of 4.95 and 3.52, respectively. This trend is not limited to low-dimensional data; on ImageNet-256, EVODiff also maintains a competitive edge, further underscoring the robustness of our approach (Table 2).

Furthermore, EVODiff also excels when compared to another class of reference-based techniques: learning-based methods that distill knowledge from prior trajectories. As shown in Table 13, while specialized methods like UniPC [LD3, [59]] are highly effective, EVODiff surpasses them at 10 NFE with a leading FID of *2.74*. It is crucial to note that this result is achieved without the need for an expensive offline distillation or training phase, highlighting a significant practical advantage in terms of flexibility and resource efficiency. Collectively, these results furnish compelling evidence that the reference-free paradigm of EVODiff is not a compromise but a fundamental strength.

**Computational Efficiency.**    A critical consideration is whether these performance gains come at the expense of computational efficiency. The end-to-end generation time comparison in Table 15 and Table 4 confirms that EVODiff introduces negligible or even reduced computational overhead/cost compared to the highly optimized DPM-Solver++ baseline. This is because our algorithm is a second-order method, yet the adaptive optimization of parameters $\zeta_i$ and $\eta_i$ relies on closed-form solutions involving lightweight vector operations (as shown in Lemmas 4.4 and 4.5). Consequently, these steps add minimal latency relative to the computationally intensive forward pass of the neural

Table 11: We conducted ablation experiments under different guidance scales and different random seeds. Quantitative results of the gradient estimation-based denoising iterations using the pre-trained model [4] on ImageNet-256×256 [76]. We report the FID↓ for 10k samples evaluated under various NFEs. **Bold** values indicate the best FID in each iteration step column, while *italicized* values represent the second best.

| Method | Model | NFE | | | | | | |
|---|---|---|---|---|---|---|---|---|
| | | 5 | 6 | 8 | 10 | 12 | 15 | 20 |
| DPM-Solver++-2 | | 16.39 | 12.77 | 9.92 | 8.88 | 8.31 | 8.03 | 7.76 |
| DPM-Solver++-3 | | 15.64 | 11.64 | 9.21 | 8.51 | 8.12 | 7.97 | 7.69 |
| UniPC-2 | | 15.15 | 11.79 | 9.41 | 8.63 | 8.16 | 7.93 | 7.71 |
| UniPC-3 | Guided-Diffusion | 14.93 | 11.22 | 9.21 | 8.55 | 8.19 | 7.98 | 7.70 |
| DPM-Solver-v3-2 | (s=2, seed=1234) | 14.88 | *11.21* | 9.17 | 8.51 | 8.12 | 7.90 | 7.67 |
| DPM-Solver-v3-3 | | 15.62 | 11.73 | 9.57 | 8.89 | 8.37 | 8.01 | 7.65 |
| EVODiff ($r_{\text{logSNR}}$) | | **13.94** | **10.96** | **9.02** | *8.38* | *8.01* | *7.83* | *7.54* |
| EVODiff ($r_{\text{refined}}$) | | *14.21* | *11.21* | 9.05 | **8.34** | **7.97** | **7.80** | **7.48** |
| DPM-Solver++-2 | | 16.62 | 12.86 | 9.73 | 8.68 | 8.17 | 7.80 | 7.51 |
| DPM-Solver++-3 | | 15.69 | 11.65 | 9.06 | 8.29 | 7.94 | 7.70 | 7.48 |
| UniPC-2 | | 15.37 | 11.78 | 9.22 | 8.40 | 8.01 | 7.71 | 7.47 |
| UniPC-3 | Guided-Diffusion | 15.05 | 11.30 | 9.07 | 8.36 | 8.01 | 7.72 | 7.47 |
| DPM-Solver-v3-2 | (s=2, seed=3407) | 14.92 | *11.13* | 8.98 | **8.14** | 7.93 | 7.70 | 7.42 |
| DPM-Solver-v3-3 | | 15.51 | 11.77 | 9.37 | 8.67 | 8.18 | 7.73 | 7.52 |
| EVODiff ($r_{\text{logSNR}}$) | | **13.98** | **10.98** | **8.84** | 8.16 | *7.81* | *7.52* | *7.32* |
| EVODiff ($r_{\text{refined}}$) | | *14.33* | 11.16 | *8.95* | **8.14** | **7.79** | **7.48** | **7.25** |
| DPM-Solver++-2 | | 16.27 | 12.40 | 9.55 | 8.66 | 8.18 | 7.84 | 7.61 |
| DPM-Solver++-3 | | 15.93 | *11.49* | 8.98 | 8.39 | 8.11 | 7.74 | 7.63 |
| UniPC-2 | | *15.44* | 11.64 | 9.11 | 8.46 | 8.17 | 7.75 | 7.62 |
| UniPC-3 | Guided-Diffusion | 16.11 | 11.88 | 9.25 | 8.58 | 8.14 | 7.77 | 7.72 |
| DPM-Solver-v3-2 | (s=3, seed=3407) | 17.97 | 12.04 | 9.17 | 8.40 | 8.11 | 7.76 | 7.67 |
| DPM-Solver-v3-3 | | 20.87 | 14.94 | 10.68 | 9.29 | 8.57 | 7.92 | 7.77 |
| EVODiff ($r_{\text{logSNR}}$) | | **14.37** | **11.04** | **8.87** | *8.37* | 7.89 | **7.56** | 7.51 |
| EVODiff ($r_{\text{refined}}$) | | 15.93 | 11.94 | 9.21 | **8.31** | **7.89** | *7.58* | *7.54* |

network. In many low-NFE scenarios, our method is even marginally faster. This finding is crucial, as it establishes that EVODiff offers a Pareto improvement, achieving superior sample quality at no additional computational cost.

**Generalizability of the Core Principles.** Finally, to demonstrate the fundamental nature of our proposed principles, we tested whether our variance-control concept could enhance other state-of-the-art frameworks. We integrated our entropy-aware approach into the EMS-parameterized structure of DPM-Solver-v3. As evidenced in Table 14 (labeled "RE-based") and Figure 8, this hybrid method surpasses the already formidable performance of the original DPM-Solver-v3 (e.g., achieving *10.61* FID vs. 12.21 at 5 NFE on EDM). This result provides the strongest validation, elevating entropy-aware variance optimization from a mere algorithmic heuristic to a powerful and universal principle for diffusion model inference. Moreover, it suggests that the improvements from our entropy-aware optimization and the EMS-based approach may be orthogonal, opening promising avenues for future work in combining these principles for even greater performance gains.

## E.9 More Experiments for EVODiff

We conducted additional experiments to assess the robustness and versatility of EVODiff. These tests span various pre-trained models, datasets, noise schedules, and complex conditional generation tasks, aiming to demonstrate that EVODiff's superior performance arises from its entropy-aware, reference-free design, rather than being limited to specific conditions.

Table 12: Comparison of FID scores for different sampling methods on CIFAR-10 with the unconditional EDM model.

| Method | Model | Reference-based? | Entropy-aware? | NFE 5 | 6 | 8 | 10 | 12 | 15 |
|---|---|---|---|---|---|---|---|---|---|
| DPM-Solver++ | | ✗ | ✗ | 27.96 | 16.87 | 8.40 | 5.10 | 3.70 | 2.83 |
| UniPC | EDM | ✗ | ✗ | 27.03 | 17.32 | 7.67 | 3.97 | 2.76 | 2.23 |
| DPM-Solver-v3 | | ✓ | ✗ | **11.60** | **8.22** | 4.94 | 3.52 | 2.81 | 2.40 |
| EVODiff | | ✗ | ✓ | **17.84** | **9.17** | **3.98** | **2.78** | **2.30** | **2.12** |

Table 13: Comparison on CIFAR10 with recent learning-based and learning-free methods under 6, 8, 10 NFEs. **Bold** indicates the best FID in each column, *italicized* indicates the second best.

| Dataset | Method Type | Method | NFE 6 | 8 | 10 |
|---|---|---|---|---|---|
| CIFAR10 | Learning-based with prior trajectories | UniPC (3M) | 13.12 | 4.41 | 3.16 |
| | | GITS [70] (UniPC prior) | 11.19 | 5.67 | 3.70 |
| | | LD3 [59] (UniPC prior) | **5.92** | **3.42** | *2.87* |
| | Learning-free and reference-free | **EVODiff (2m)** | *9.07* | *3.88* | **2.74** |

A key indicator of a sampler's utility is its consistent performance across diverse settings. We first demonstrate this quantitative consistency on standard benchmarks. As shown in Table 14, on CIFAR-10, EVODiff excels with both the ScoreSDE and EDM pre-trained models, achieving a state-of-the-art FID of *10.61* at just 5 NFE on EDM. Furthermore, its superiority is maintained under different noise schedules; Tables 16 through 19 show that EVODiff consistently secures the leading FID scores on high-resolution datasets like FFHQ-64 and ImageNet-64, regardless of whether a "logSNR" or "EDM" schedule is employed. This consistent dominance across varied models and schedules strongly indicates that EVODiff's performance gains are intrinsic to its algorithmic design rather than an artifact of a specific setup.

Moving beyond numerical metrics, we evaluated EVODiff's qualitative performance on the highly demanding task of text-to-image synthesis with Stable Diffusion v1.4 and v1.5. This setting tests a sampler's ability to handle complex semantic guidance and generate coherent, high-fidelity images. As visualized in Figures 10 and 12, our method produces images with significantly *fewer artifacts* and *greater structural integrity*. Most compellingly, Figure 13 highlights a crucial advantage in semantic consistency: for the prompt "an astronaut riding a horse," competing methods generated an anatomically incorrect horse with five legs, a common failure mode in DMs. In contrast, EVODiff correctly rendered a four-legged animal, demonstrating its superior ability to preserve semantic and anatomical plausibility. This suggests that our entropy-aware optimization leads to a more stable and accurate information flow from text prompt to pixel space.

A critical consideration for any practical sampler is whether performance gains are achieved at the expense of computational efficiency. We explicitly address this by comparing the end-to-end generation time of EVODiff with the highly optimized DPM-Solver++ baseline. The results, presented in Table 15, confirm that EVODiff introduces negligible computational overhead. In many low-NFE scenarios, it is even marginally faster. This finding is crucial, as it establishes that EVODiff offers a Pareto improvement: superior sample quality at no additional computational cost. This makes it a highly practical reference-free solution for real-world deployment.

Finally, to demonstrate the fundamental and generalizable nature of our proposed principles, we tested whether our variance-control concept could enhance other SOTA frameworks. We integrated our entropy-aware approach into the EMS-parameterized structure of DPM-Solver-v3. As evidenced in Table 14 and Figure 8, this hybrid method (labeled "RE-based") surpasses the already formidable performance of the original DPM-Solver-v3. This result provides the strongest validation that entropy-aware variance optimization is not merely a set of heuristics for a single algorithm, but a powerful, universal principle for improving diffusion model inference.

Table 14: Quantitative results of FID ↓ scores for gradient-based methods on CIFAR-10. The results are evaluated on 50k samples for various NFEs, some results are borrowed from the original papers. The "RE-based (our)" row demonstrates the application of our entropy-aware principles to the DPM-Solver-v3 framework.

| Method | Model | NFE | | | | | | |
|---|---|---|---|---|---|---|---|---|
| | | 5 | 6 | 8 | 10 | 12 | 15 | 20 |
| DEIS [14] | | 15.37 | \ | \ | 4.17 | \ | 3.37 | 2.86 |
| DPM-Solver++ [20] | | 28.53 | 13.48 | 5.34 | 4.01 | 4.04 | 3.32 | 2.90 |
| UniPC [16] | ScoreSDE | 23.71 | 10.41 | 5.16 | 3.93 | 3.88 | 3.05 | 2.73 |
| DPM-Solver-v3 [17] | | **12.76** | **7.40** | **3.94** | 3.40 | 3.24 | 2.91 | 2.71 |
| RE-based (our) | | 13.54 | 8.56 | 4.11 | **3.38** | **3.22** | **2.76** | **2.42** |
| Heun's 2nd [12] | | 320.80 | 103.86 | 39.66 | 16.57 | 7.59 | 4.76 | 2.51 |
| DPM-Solver++ [20] | | 24.54 | 11.85 | 4.36 | 2.91 | 2.45 | 2.17 | 2.05 |
| UniPC [16] | EDM | 23.52 | 11.10 | 3.86 | 2.85 | 2.38 | 2.08 | 2.01 |
| DPM-Solver-v3 [17] | | 12.21 | 8.56 | 3.50 | 2.51 | 2.24 | 2.10 | 2.02 |
| RE-based (our) | | **10.61** | **8.22** | **3.37** | **2.43** | **2.21** | **2.07** | **2.01** |

Table 15: Comparison of Computational Overhead between EVODiff and DPM-Solver++ on ImageNet-256 with 10k samples on a 3090 GPU.

| NFE | DPM-Solver-2m | Total Time | EVODiff | Total Time |
|---|---|---|---|---|
| 5 | 9.56s/it | 1:03:24 (h:m:s) | 9.45s/it | 1:02:15 (h:m:s) |
| 10 | 18.40s/it | 2:02:39 (h:m:s) | 18.39s/it | 2:00:39 (h:m:s) |
| 15 | 27.24s/it | 3:01:34 (h:m:s) | 27.25s/it | 3:01:47 (h:m:s) |
| 20 | 36.07s/it | 4:00:28 (h:m:s) | 36.11s/it | 4:00:48 (h:m:s) |

Table 16: Comparison of FID scores for different sampling methods on FFHQ-64×64 using the logSNR schedule.

| Method | Model | NFE | | | | |
|---|---|---|---|---|---|---|
| | | logSNR schedule | | | | |
| | | 5 | 10 | 15 | 20 | 25 |
| Heun | | 342.28 | 45.46 | 7.60 | 3.25 | 2.71 |
| DPM-Solver++ | | 28.96 | 6.87 | 4.07 | 3.29 | 2.97 |
| UniPC_bh1 | FFHQ-64, EDM | 35.78 | 4.00 | 2.81 | 2.60 | 2.52 |
| UniPC_bh2 | | 27.00 | 5.44 | 3.38 | 2.87 | 2.67 |
| EVODiff | | **20.04** | **3.93** | **2.72** | **2.55** | **2.46** |

Table 17: Comparison of FID scores for different sampling methods on FFHQ-64×64 using the EDM schedule.

| Method | Model | NFE | | | | | |
|---|---|---|---|---|---|---|---|
| | | EDM schedule | | | | | |
| | | 5 | 10 | 15 | 20 | 25 | 35 |
| Heun | | 347.09 | 29.92 | 9.95 | 4.58 | 3.41 | 2.71 |
| DPM-Solver++ | FFHQ-64, EDM | 25.08 | 6.81 | 3.80 | 3.00 | 2.75 | 2.59 |
| UniPC_bh1 | | 28.87 | 6.66 | 3.40 | 2.69 | 2.58 | 2.50 |
| UniPC_bh2 | | 24.09 | 6.17 | 3.35 | 2.73 | 2.58 | 2.50 |
| EVODiff | | **19.65** | **5.31** | **3.02** | **2.64** | **2.56** | **2.48** |

Table 18: Comparison of FID scores for different sampling methods on ImageNet 64×64 using the logSNR schedule.

| Method | Model | NFE | | | | | |
|--------|-------|-----|-----|-----|-----|-----|-----|
| | | logSNR schedule | | | | | |
| | | 5 | 10 | 15 | 20 | 25 | 35 |
| Heun | | 231.754 | 23.269 | 7.413 | 3.597 | 3.020 | 2.513 |
| DPM-Solver++ | ImageNet 64, EDM | 32.529 | 7.052 | 3.922 | 3.077 | 2.738 | 2.465 |
| UniPC_bh1 | | 41.366 | 5.453 | 3.041 | 2.578 | 2.415 | 2.286 |
| UniPC_bh2 | | 31.063 | 5.795 | 3.312 | 2.714 | 2.490 | 2.319 |
| EVODiff | | **23.979** | **4.289** | **2.688** | **2.384** | **2.242** | **2.143** |

Table 19: Comparison of FID scores for different sampling methods on ImageNet 64×64 using the edm schedule.

| Method | Model | NFE | | | | | |
|--------|-------|-----|-----|-----|-----|-----|-----|
| | | EDM schedule | | | | | |
| | | 5 | 10 | 15 | 20 | 25 | 35 |
| Heun | | 248.402 | 15.129 | 5.301 | 3.136 | 2.739 | 2.424 |
| DPM-Solver++ | ImageNet 64, EDM | 27.243 | 5.785 | 3.480 | 2.866 | 2.606 | 2.393 |
| UniPC_bh1 | | 39.158 | 5.649 | 3.456 | 2.701 | 2.424 | 2.268 |
| UniPC_bh2 | | 26.354 | 5.042 | **3.118** | 2.639 | 2.434 | 2.284 |
| EVODiff | | **21.894** | **4.734** | 3.316 | **2.594** | **2.308** | **2.167** |

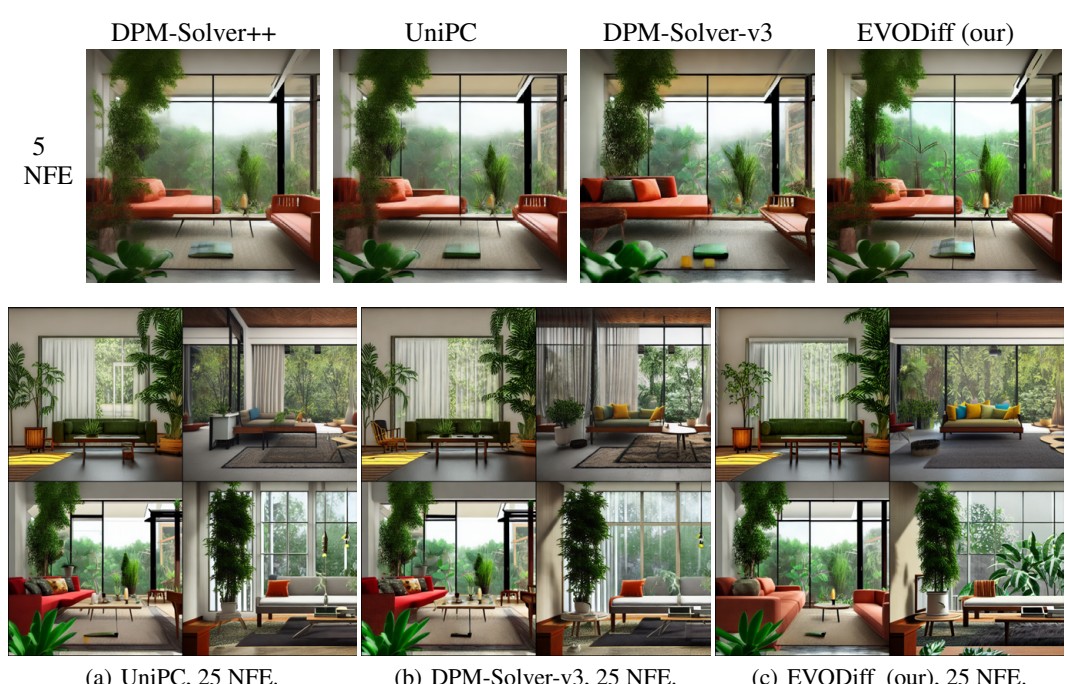

Figure 7: Random samples from Stable-Diffusion-v1.4 [28] with a classifier-free guidance scale 7.5, using the text prompt "*environment living room interior, mid century modern, indoor garden with fountain, retro, m vintage, designer furniture made of wood and plastic, concrete table, wood walls, indoor potted tree, large window, outdoor forest landscape, beautiful sunset, cinematic, concept art, sunstainable architecture, octane render, utopia, ethereal, cinematic light*". Our EVODiff method demonstrates consistent improvements across both low (5 NFE) and higher (25 NFE) inference steps.

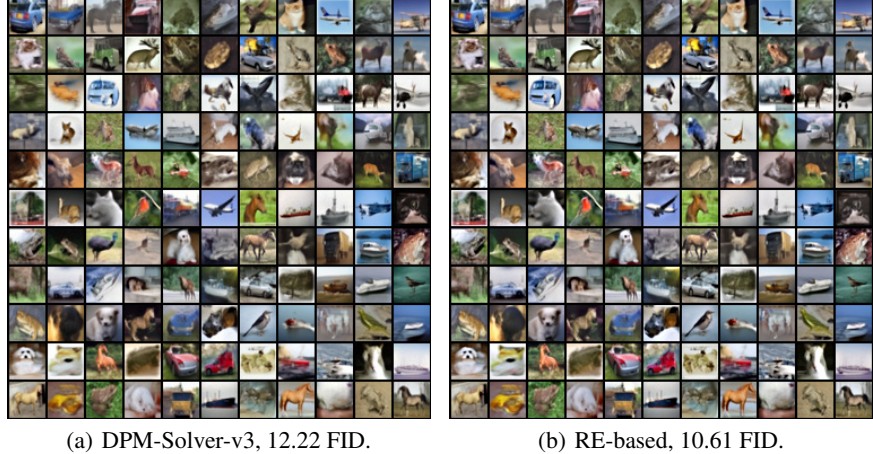

(a) DPM-Solver-v3, 12.22 FID.        (b) RE-based, 10.61 FID.

Figure 8: Random samples of EDM [12] on the CIFAR-10 dataset with only 5 NFEs. Within the EMS-parameterized iterative framework provided by DPM-Solver-v3, the RE-based iterative approach improves FID by explicitly balancing the conditional variance of the gradient term itself.

| DPM-Solver++
7.76 FID | UniPC
7.71 FID | DPM-Solver-v3
7.67 FID | EVODiff 1
7.48 FID |

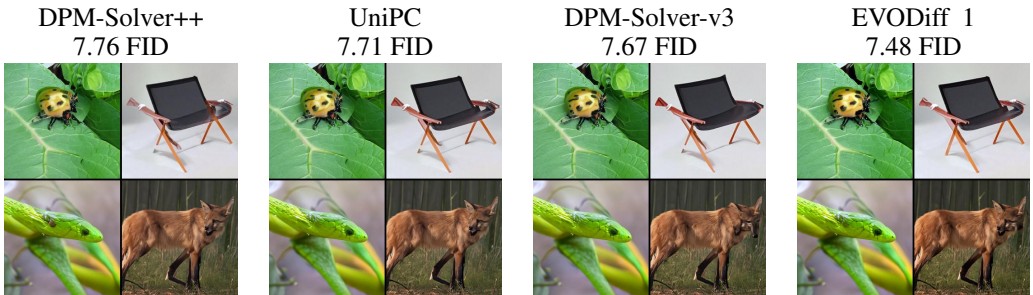

Figure 9: Random samples from the pretrained Guided-Diffusion model [4] with 20 NFE on the ImageNet-256 dataset [76]. Our EVODiff method reduces reconstruction error without relying on a reference solution, while can retain the sample quality benefits of methods like DPM-Solver-v3, which relies on EMS-based statistics optimized using a reference solution.

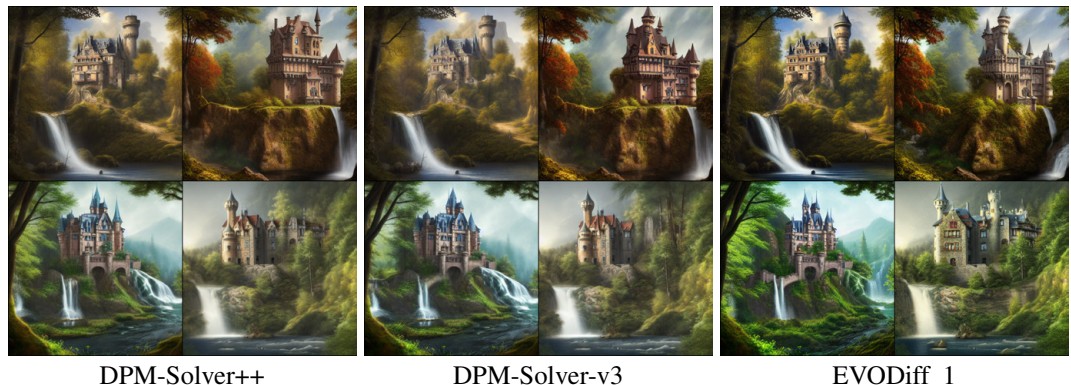

DPM-Solver++        DPM-Solver-v3        EVODiff 1

Figure 10: Random samples from Stable-Diffusion-v1.4 [28] with a classifier-free guidance scale 7.5, using 10 NFE and the prompt "*A beautiful castle beside a waterfall in the woods, by Josef Thoma, matte painting, trending on artstation HQ*". Images generated by our EVODiff method exhibit greater clarity and naturalness, along with more coherent and complete structural content.

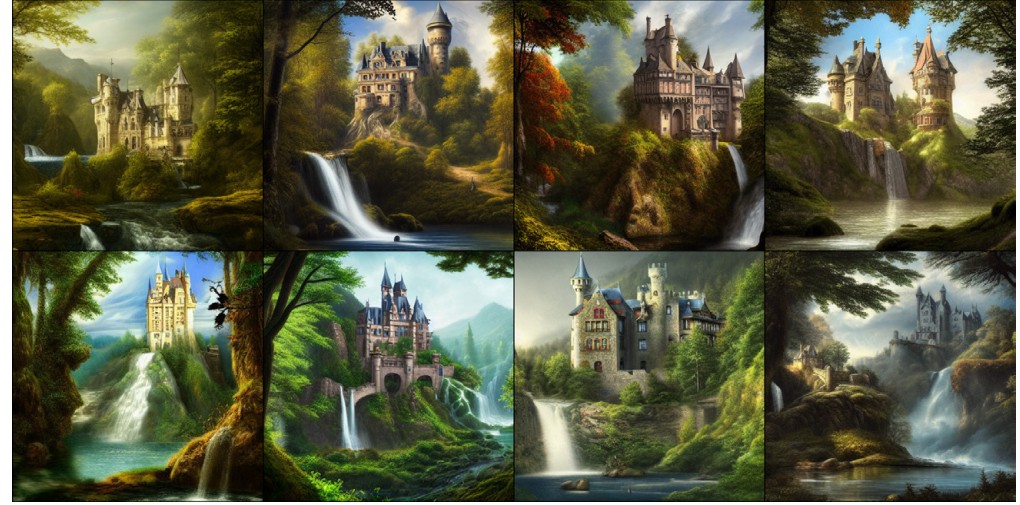

(a) EVODiff 1 ($\mu$=0.75).

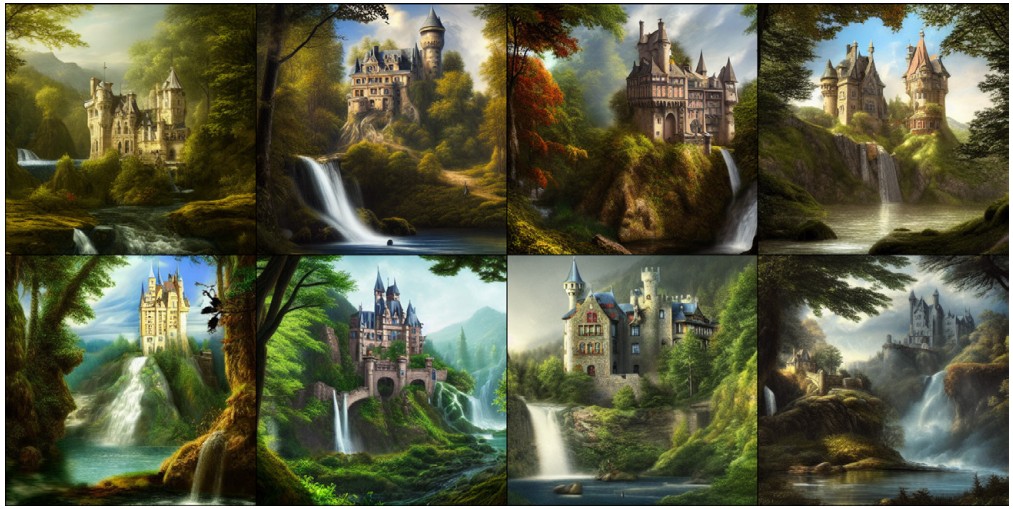

(b) EVODiff 1 ($\mu$=0.5).

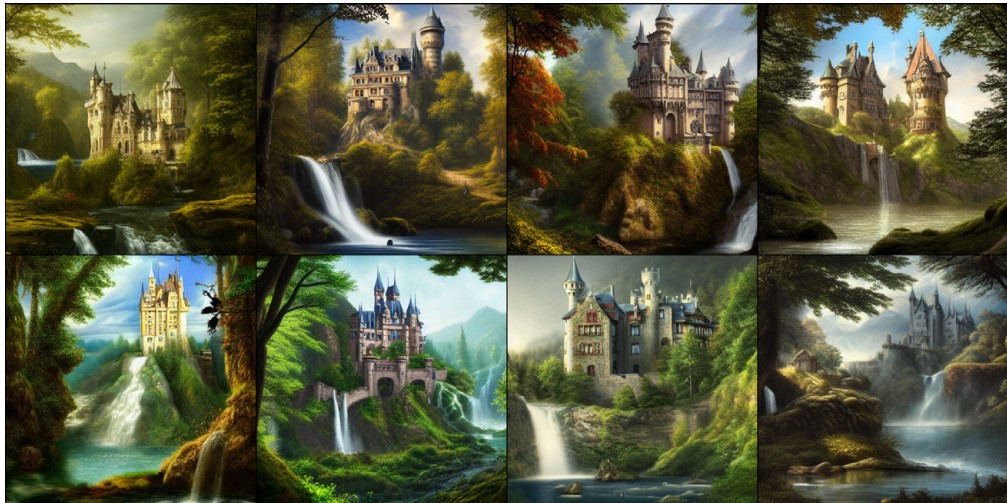

(c) EVODiff 1 ($\mu$=0.1).

Figure 11: Random samples from Stable-Diffusion-v1.4 [28] with a CFG scale 7.5, different shift parameters, using 10 NFE and the text prompt "*A beautiful castle beside a waterfall in the woods, by Josef Thoma, matte painting, trending on artstation HQ*". Our EVODiff inference consistently generates clear and complete content across different shift parameters.

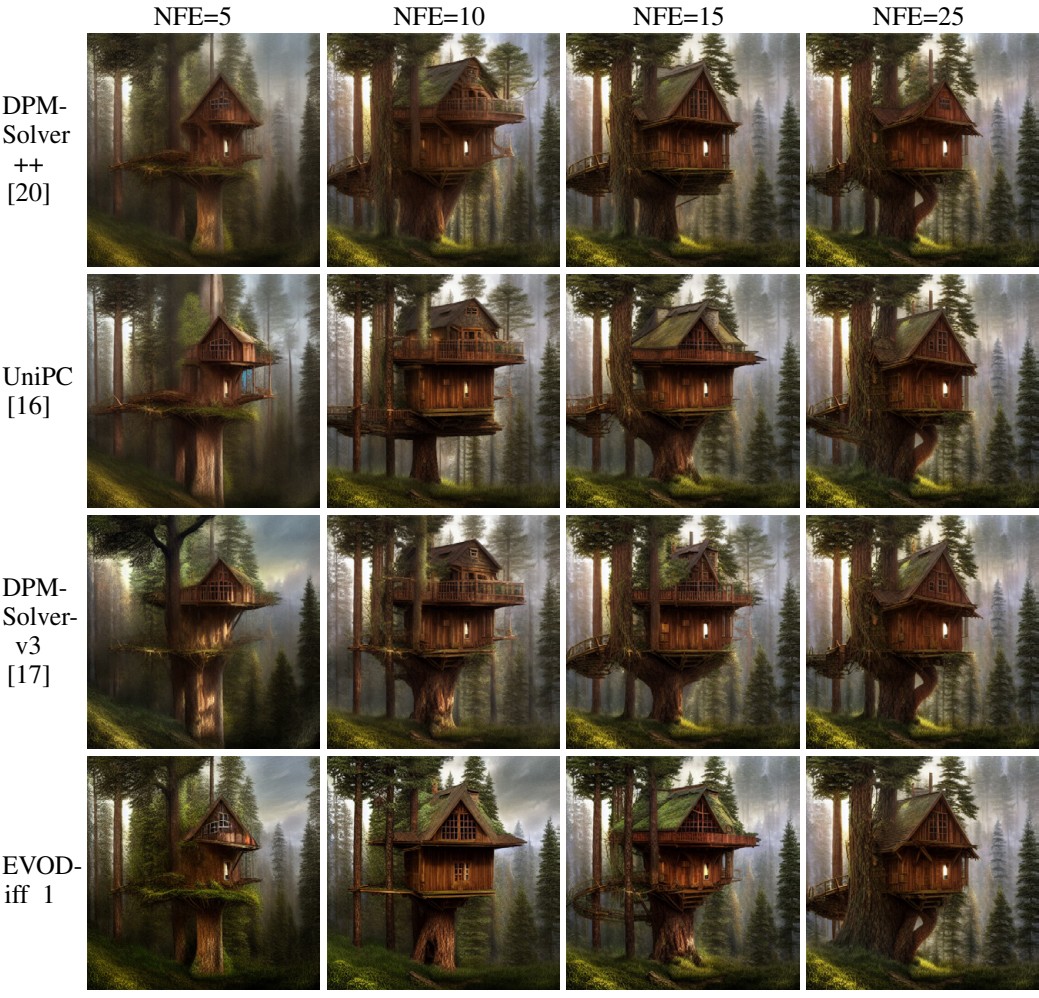

Figure 12: Random samples from Stable-Diffusion [28] with a guidance scale 7.5, using varying NFEs and the prompt "*tree house in the forest, atmospheric, hyper realistic, epic composition, cinematic, landscape vista photography by Carr Clifton & Galen Rowell, 16K resolution, Landscape veduta photo by Dustin Lefevre & tdraw, detailed landscape painting by Ivan Shishkin, DeviantArt, Flickr, rendered in Enscape, Miyazaki, Nausicaa Ghibli, Breath of The Wild, 4k detailed post processing, artstation, unreal engine*". Our EVODiff inference improves content clarity, coherence, and overall completeness. In contrast, other methods exhibit partial content collapse at 25 NFEs, whereas ours preserves structural integrity.

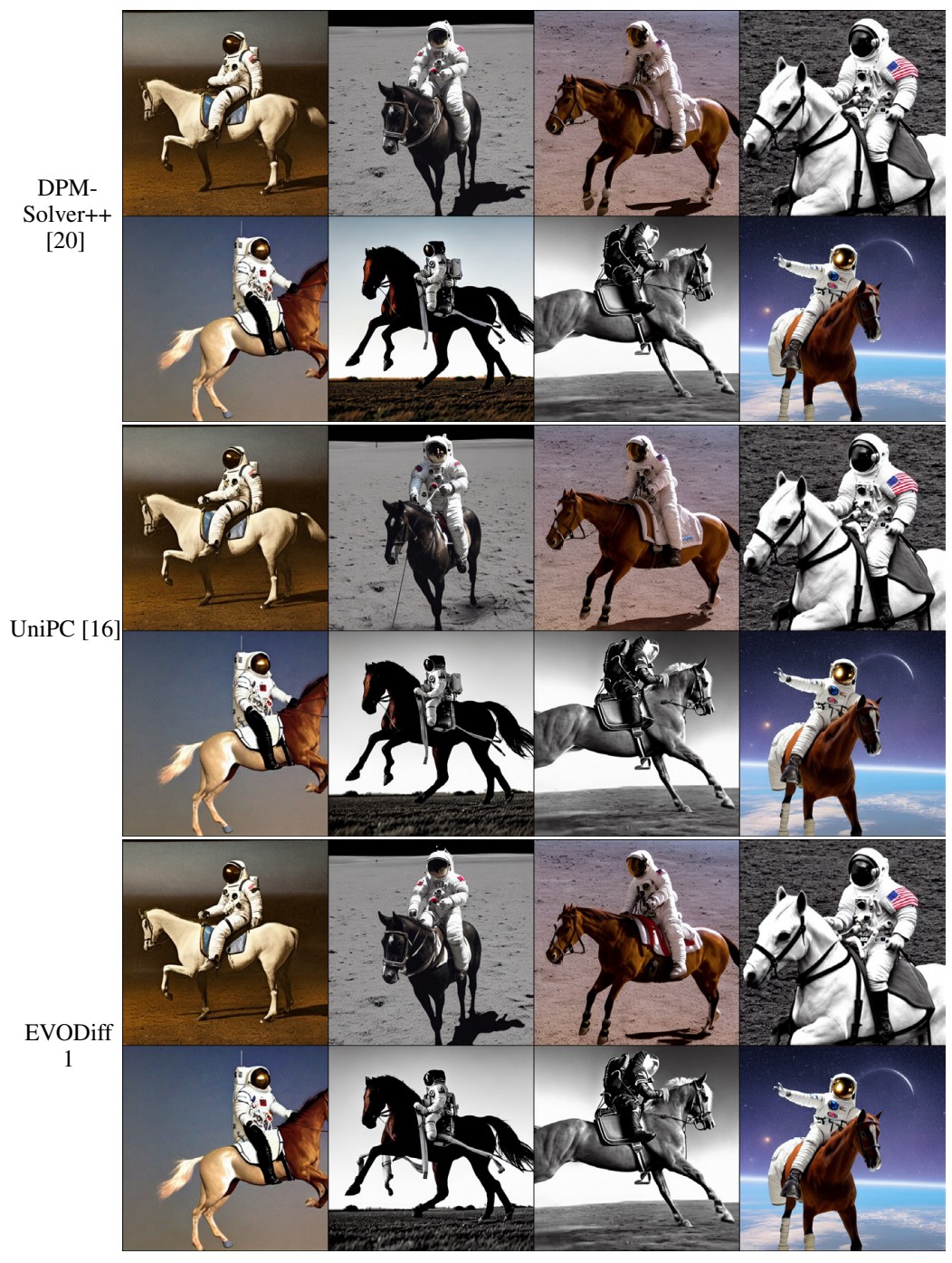

Figure 13: Random samples from Stable-Diffusion-v1.4 [28] with a classifier-free guidance scale 7.5, using 50 NFE and the text prompt "*a photograph of an astronaut riding a horse*". In the images at position (2,2), *other methods produced anatomically incorrect horses with five legs*, whereas our EVODiff inference correctly generated *anatomically accurate horses with four legs*.

NFE=10 NFE=25

DPM-
Solver++
[20]

UniPC
[16]

EVODiff 1

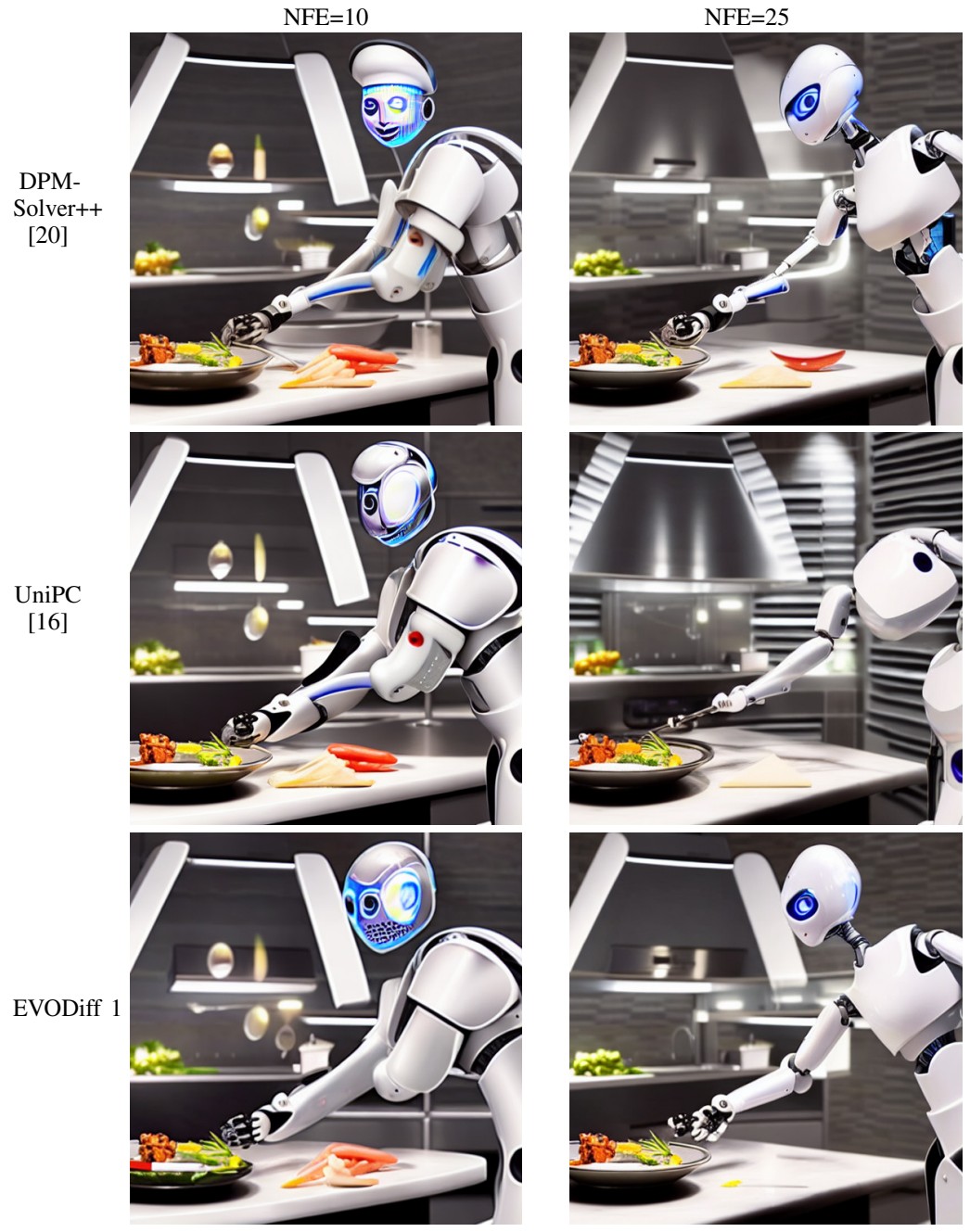

Figure 14: Random samples from Stable-Diffusion-v1.5 with a guidance scale 7.5, using varying NFEs and the prompt "*A robot chef cooking a meal in a futuristic kitchen, with glowing utensils and a holographic recipe book, highly detailed, sci-fi atmosphere*". Our EVODiff inference improves content continuity and consistency, and effectively reduces visual artifacts.

