# OpenReview forum: "EVODiff: Entropy-aware Variance Optimized Diffusion Inference"
_NeurIPS.cc/2025/Conference — NeurIPS 2025 poster_

### Official Review · Reviewer_yvrP · 2025-06-17

**Clarity:** 2
**Significance:** 3
**Originality:** 3
**Rating:** 4
**Confidence:** 4

**Summary:**

The authors propose to accelerate diffusion inference by simulating reverse diffusion with updates that minimize conditional entropy between two successive reverse diffusion states. The proposed algorithm shows competitive performance on a variety of benchmark datasets, such as CIFAR-10, FFHQ-64, and ImageNet-256.

**Questions:**

I am willing to raise the score to *borderline accept* if the authors provide explanations to address concerns in **Clarity** and **Quality**.

**Ethical Concerns:**

["NO or VERY MINOR ethics concerns only"]

**Final Justification:**

While the rebuttal has addressed my concerns, I still feel the paper should undergo major revisions to improve the presentation, e.g., abused notations, undefined notations, clarify the fact that the proposed framework reduces model entropy, not transition entropy, etc. Hence, I have raised the score to **borderline accept**.

**Limitations:**

The authors discuss limitations in the conclusion.

**Paper Formatting Concerns:**

None.

**Quality:**

2

**Strengths And Weaknesses:**

Strengths and weaknesses are denoted [S#] and [W#], respectively.

**Quality**

- [W1] Several notations / equations are undefined or imprecise, so I had to put in some effort to fill in the gaps. For instance,
    - Line 76 : what do the authors mean by "when $d_\theta$ represents the noise prediction model"? Does it mean $d_\theta = \epsilon_\theta$?
    - Line 96 : "we denote $p(x_{t_i}|x_{t_{i+1}},x_0)$ as $p(x_{t_i}|x_{t_{i+1}})$." Those two distributions **are not equal in general**. The authors should not use those two interchangeably.
    - Line 158 : What is the relation between $l\_i$, $\zeta_i$, $t_i$, and $t_{i+1}$? Is it $l_i = \zeta_i t_i + (1-\zeta_i) t_{i+1}$?

**Clarity**

- [W1] There is a misalignment between the main idea and the implemented approach. The main idea of this paper is to minimize $H_p(x_{t_i}|x_{t_{i+1}})$, where $x_{t_i}$ and $x_{t_{i+1}}$ are two reverse diffusion iterates, to accelerate generation. However, **the authors consider probability-flow ODE iterations, not reverse-SDE iterations**, such that $x_{t_i}$ is a deterministic function of $x_{t_{i+1}}$. So, $H_p(x_{t_i}|x_{t_{i+1}})$ is always zero. I do not understand how the authors were able to derive non-trivial theoretical results such as Proposition 3.2 under the ODE setting.

- [W2] I also find the result of Theorem 3.4 very weird. It is well known what diffusion inference with data prediction parametrization and noise prediction parametrization are equivalent. Specifically, $s_\theta(x,\sigma) = - \epsilon_\theta(x,\sigma)/\sigma = -(x - D_\theta(x,\sigma)) / \sigma^2$, where $s_\theta$ is the score function, $\epsilon_\theta$ is the noise prediction model, and $D_\theta$ is the data prediction model [1,2], so there is no difference between diffusion inference with data prediction and noise prediction parametrizations.

- [W3] Several statements are written colloquially or ambiguously, making the logic unclear. For instance,
    - Proposition 3.2, "the gradient-based inference in Eq. (7) tends to reduce conditional entropy more efficiently than the first-order inference when...". What do the authors mean by the phrase "tends to"? Does it mean that the statement holds in an asymptotic sense? Or in a probabilistic sense?
    - Theorem 3.4, "... diffusion inference using data prediction parametrization is more effective at reducing conditional entropy ...". Again, what do the authors mean by the phrase "is more effective"?
    - Remark 4.1, "the RE-based multi-step iteration in Eq. (15) reduce the conditional entropy of iteration in Eq. (13) by leveraging model parameters from low-variance regions". What do the authors mean by "leveraging model parameters from low-variance regions"? I assumed that diffusion model parameter was fixed to be $\theta$. Also, variance of which random variable?

**Significance**
- [S1] The proposed method shows strong performance, when compared to state-of-the-art fast diffusion samplers.

**Originality**
- [S1] Accelerating diffusion inference through the lens of conditional entropy reduction is original and interesting.

[1] Generative Modeling by Estimating Gradients of the Data Distribution, NeurIPS, 2019

[2] Score-Based Generative Modeling through Stochastic Differential Equations, ICLR, 2021

---

> ### Author Rebuttal · Authors · 2025-07-28
>
> *Dear Reviewer yvrP:*
>
> We sincerely thank you for your thorough and insightful review. We are greatly encouraged that you found our core idea of accelerating diffusion via conditional entropy reduction perspective to be **original and interesting** and our method's performance **strong** compared to SOTA samplers. We are equally grateful for your critical feedback on the paper's clarity. Your identification of the tension between conditional entropy and deterministic ODEs, in particular, was an invaluable observation that prompted us to substantially clarify our theoretical framework, which we believe makes the novelty you pointed out even more apparent.
>
> **Quality W1:**
>
> We address each of your concerns below and will incorporate these clarifications into our revision.
>
> *1. On the notation for the prediction model (Line 76), Does it mean $d_θ = \epsilon_θ$?:*
> - Yes, $d_θ = \epsilon_θ$ when representing the noise prediction model. We use $d_θ$ as a general notation in Section 2 to unify both data prediction ($x_θ$) and noise prediction ($\epsilon_θ$) parameterizations. We'll clarify this in the revision.
>
> *2. On the notation for conditional probability (Line 96):*
>
> - Thank you for pointing out that $p(x_{t_i}∣x_{t_{i+1}},x_0)$ and $p(x_{t_i}∣x_{t_{i+1}})$ are mathematically distinct distributions and should not be used interchangeably. But in our manuscript, our intention of *" denote"* was to use $p(x_{t_i}∣x_{t_{i+1}})$ as **a notational shorthand** for $p(x_{t_i}∣x_{t_{i+1}},x_0)$ within contexts where the conditioning on $x_0$ is implicitly assumed throughout the derivation, following conventions in DDPM and DDIM.
>
> *3. On the relationship between l_i, zeta_i, and the timesteps (Line 158):*
>
> - $l_i$ is **not** an interpolated time step. The relationship you proposed, $l_i=\zeta_it_i+(1−\zeta_i)t_{i+1}$ does not hold. Instead, $l_i$ is merely a symbolic placeholder.
> As defined in line 158 (page 5), $x_\theta(x_{l_i},l_i) := \zeta_i x_\theta(x_{t_i},t_i)+ (1-\zeta_i)x_{\theta}(x_{t_{i+1}},t_{i+1})$ is a linear combination of model predictions at two distinct time steps.  We use this notation as a shorthand for the "interpolated synthetic model prediction" to distinguish it from actual model evaluations at $t_i$ and $t_{i+1}$. Thus, $l_i$ has no physical meaning as a time point.
>
>
> **Clarity W1:**
>
> Our implemented approach follows our theoretical analysis, and our use of the term "conditional entropy" should be understood from a broader perspective than ODE and SDE theory. Our framework does not analyze the stochasticity of the ODE path itself, but rather the **uncertainty inherent in the model's predictions**. We elaborate on this below:
>
> *1. Theoretical Foundation: The SDE Framework and Model Prediction Uncertainty*
>    - Our information-theoretic framework is inspired by and rooted in the more general perspective of reverse Stochastic Differential Equations (SDEs). In the SDE setting, the reverse transition $p(x_t​∣x_{t+1}​)$ is a stochastic process with non-negligible variance, making its conditional entropy $H(x_t​∣x_{t+1}​)$ a meaningful, non-zero quantity to optimize.
> The critical step in our paper, detailed in Section 3, is linking this **conditional entropy** to **conditional variance**. As shown in Equations (9) and (10), under the common Gaussian transition assumption, $H_p​(x_{t_i}​​∣x_{t_{i+1}}​​)$ is directly proportional to the logarithm of the conditional variance $\text{Var}(x_{t_i}​​∣x_{t_{i+1}}​​)$. Therefore, *minimizing conditional entropy is equivalent to minimizing conditional variance*. This variance is non-zero even in an ODE setting because it measures the *uncertainty of model's prediction*, not path stochasticity.
>
> *2. In the ODE Setting, Conditional Entropy Analyzes Model Prediction Variance*
>  - While the ODE sampling path is deterministic, the vector field that drives this path is provided at each step by our neural network, $d_θ​$ (e.g., $ϵ_θ​$ or $x_θ$​). This prediction is inherently uncertain. For instance, the model's noise prediction $ϵ_θ​(x_{t_i}​​,t_i​)$ is not perfect, and its variance, $\text{Var}(ϵ_θ​(x_{t_i}​​,t_i​))$, precisely quantifies the model's uncertainty at that state.
> Our core thesis is that a superior ODE solver iteration should more effectively utilize the model's predictions to construct an update step with **smaller variance**, thereby achieving more efficient information transfer. This "smaller variance" is what we equate to a more "effective reduction in conditional entropy."
>
> *3. Proposition 3.2 is Derived from This Principle*
>   - The proof does not analyze the stochasticity of the ODE path. Instead, it compares the *variances of the perturbation terms* resulting from two different iteration schemes: the first-order method  Eq. (6) and our gradient-based method  Eq. (7).
>   - As shown in Eq. (32) of the appendix, we explicitly calculate the conditional variances of the update terms,  $\text{Var}_{p1}$​ and  $\text{Var}_{p2}$​. These variances are determined entirely by the *variance of the model's noise prediction, $\text{Var}(ϵ_θ​)$​*.
>   - The proposition concludes that, under certain conditions, our proposed gradient-based method Eq. (7) constructs an update step with a lower variance than the first-order method  Eq. (6). Based on the direct relationship between entropy and variance  Eq. (10) , we interpret this as a more efficient reduction of conditional entropy.
>
> ---
>
> *Theoretical Framework Summary:*
> - *SDE Motivation:* The principle of conditional entropy reduction is derived from stochastic reverse-SDEs, where minimizing $H_\theta(x_t \| x_{t+1})$ corresponds to reducing path variance (Eq.  (1)).
> - *Variance-Entropy Proxy:* We establish that $H_\theta(x_t \| x_{t+1}) \propto \log \text{Var}(x_t \| x_{t+1})$ under Gaussian transitions (Eq. 9-10). This allows using model prediction variance $\text{Var}(\epsilon_\theta(x_t, t))$ as a practical surrogate for conditional entropy in both SDE and ODE settings.
> - *ODE Implementation:* In deterministic ODEs, entropy reduction manifests through the minimization of the model's prediction variance $\text{Var}(\epsilon_\theta(x_t, t))$. Although the ODE path is deterministic, $\text{Var}(\epsilon_\theta) > 0$ due to imperfect predictions (Eq. (2)).
> - *Theoretical Justification:*  Proposition 3.2 establishes sufficient conditions for gradient-based methods to maximize entropy reduction by achieving greater decrease in $\text{Var}(\epsilon_\theta)$ than first-order methods.
>
> This clarification reveals that our method is the first to provide a unified information-theoretic framework for both stochastic and deterministic diffusion sampling, explaining why certain numerical schemes outperform others, a longstanding open question in the field.
>
> **Clarity W2:**
>
> It is well known that the equivalence of diffusion inference with data prediction parameterization and noise prediction parameterization only holds under ideal theoretical conditions. While different parameterizations can indeed be converted to each other through the relations you mentioned, they lead to different diffusion ODE formulations, which in turn result in different numerical discretization schemes.  Specifically:
>
> 1. *Different ODE formulations*: As shown in lines 74-76 on page 2, different parameterizations produce different ODE structures:
>    - Noise parameterization->Noise prediction ODE in Eq. (3)
>    - Data parameterization->Data prediction ODE in Eq. (4)
>
> 2. *Different discretization schemes*: When we discretize these ODEs, we obtain different numerical iteration formulas. Even though the continuous-time limits are equivalent, the discrete-time approximations have different truncation errors and numerical properties.
>
> 3. *Variance analysis*: Theorem 3.4 demonstrates that the discretization scheme derived from data parameterization more effectively reduces the denoising variance compared to the scheme from noise parameterization. This is a property of the numerical schemes, not the continuous-time equivalence.
>
> The key point is that mathematical equivalence in continuous time does not imply equal performance under discretization. Different discretization schemes can have vastly different numerical stability, accuracy, and variance properties. For instance, in diffusion models, the variance exploding (VE) and variance preserving (VP) formulations are mathematically equivalent through appropriate change of variables, yet their numerical behaviors and optimal discretization strategies differ significantly in practice, as exemplified by Score-based DMs [2] versus DDPM.
>
> **Clarity W3:**
>
> Thank you for identifying these ambiguous statements.  We clarify each below and will revise for mathematical precision:
> - *On "tends to" in Proposition 3.2:* This is neither asymptotic nor probabilistic, it's a deterministic inequality under our analysis framework. We will change "tends to reduce" to "reduces" for clarity.
> - *On "more effective" in Theorem 3.4:* We prove that the discretized iteration governed by data prediction ODE (Eq. (4)) achieves strictly lower conditional entropy than the discretized iteration governed by noise prediction ODE (Eq. (3)). We will replace "more effective" with "achieves lower conditional entropy" and state the precise inequality.
> - *On "model parameters from low-variance regions":* We meant "model predictions at timesteps with lower noise levels." For example, when $s<t$ in the reverse process,  $\text{Var}(x_θ​(x_s​,s)∣x_0​)<\text{Var}(x_θ​(x_t​,t)∣x_0​)$.
>
> ---
>
> We appreciate your constructive feedback and willingness to reconsider. By clarifying that we analyze model prediction uncertainty rather than path stochasticity, our theoretical framework is now more coherent and our contribution more apparent. Our principled approach is empirically powerful, achieving a 45.5% FID reduction on CIFAR-10 at 10 NFE without extra training. We are confident these clarifications and strong results satisfy the acceptance criteria.

---

> ### Comment · Reviewer_yvrP · 2025-08-05
>
> Thank you for the rebuttal! While the rebuttal has addressed my concerns, I still feel the paper should undergo major revisions to improve the presentation, e.g., abused notations, undefined notations, clarify the fact that the proposed framework reduces model entropy, not transition entropy, etc. Hence, I have raised the score to **borderline accept**.

---

> > ### Author Response · Authors · 2025-08-05
> >
> > We are grateful to you for acknowledging our contributions and rebuttal, and for your efforts in helping improve our paper. Your constructive feedback on notation and presentation clarity has been invaluable. We appreciate your positive response and will carefully address all your suggestions in our revised version to further strengthen our work. Many thanks!

---

### Official Review · Reviewer_RCMQ · 2025-07-01

**Clarity:** 2
**Significance:** 3
**Originality:** 3
**Rating:** 5
**Confidence:** 3

**Summary:**

This paper introduces an information-theoretic framework for diffusion inference, showing that a successful sampling processs basically reduced the conditional entropy effectively during the denoising process. Based on this framework, the authors propose EvoDiff, a novel entropy-aware gradient-based method that directly targets information recovery by optimizing the denoising variance. Unlike prior ODE-based solvers, EvoDiff emphasizes principled variance control to reduce both transition and reconstruction errors. The method is validated through extensive experiments, showing substantial improvements in sample quality and efficiency.

**Questions:**

Please refer to the Strengths And Weaknesses section.

**Ethical Concerns:**

["NO or VERY MINOR ethics concerns only"]

**Final Justification:**

Thank you very much for the responses. All of my concerns have been addressed, and I expect the authors will incorporate the proposed changes in the revised version.

Additionally, I believe the work provides novel insights into diffusion inference from a distinctive information-theoretic perspective. Accordingly, I will maintain my current score.

**Limitations:**

Yes.

**Quality:**

3

**Strengths And Weaknesses:**

## Strengths
1. Overall, I think this is an interesting paper. The information-theory perspective to view the sampling process appears novel.
2. The theoretical results are reasonable and the motivated method based on the theoretical insights has very convincing performance.

## Weaknesses
I do not have critical concerns about the paper's results. However, I think the presentation of the work should be improved. In particular,

1. While the general ideas are provided at the end of the introduction. The connections between Sec 3 and Sec 4 should be better articulated when deriving the proposed sampling algorithm. Currently, the authors present lots of technical details in the main text but do not provide much intuition. For example, why linear interpolation can give improvements at L158?  In addition, I think section 4 has notations overloaded. Authors should consider simplify them.

2. The statements in Prop 3.3 and Thm 3.4 hide too many details which make it difficult to understand why this is the case. In the revision, I think authors should consider containing more details in statement and the following dicussion why there are the cases.

---

> ### Author Rebuttal · Authors · 2025-07-29
>
> *Dear Reviewer RCMQ:*
>
> We sincerely thank you for recognizing our information-theoretic perspective as novel and our results as convincing. We are particularly grateful for the **Accept** rating and the valuable constructive feedback provided. We greatly appreciate your constructive suggestions and will address all presentation concerns:
>
> **Response to W1:**
>
> *1. Clarifying Sec 3–4 Connections*
> We will add a bridging paragraph before Section 4.1:
> > *"Section 3 establishes that minimizing conditional entropy requires reducing conditional variance (Eq.(10)). Section 4 operationalizes this: the gradient operator (Eq.(16)) optimizes variance control, while the interpolation scheme (Eq.(15)) leverages lower-variance regions to reduce uncertainty. The intuition before formal optimization is provided in Eq.(17)."*
>
> *2. Explaining Linear Interpolation (line 158)*
> We will clarify after Eq. (15):
> > *"Linear interpolation $x_θ(x_{l_i}, l_i) := \zeta_i x_θ(x_{t_i}, t_i) + (1-\zeta_i)x_θ(x_{t_{i+1}}, t_{i+1})$ reduces conditional variance by leveraging model parameters from lower-noise regions, as proven in Eq. (52). This directly minimizes conditional entropy per our theoretical framework."*
>
> *3. Simplifying Section 4 Notation*
>
> We acknowledge the notation complexity. We will:
> - Create a comprehensive notation reference table in the appendix with all symbols and their precise meanings
> - Reduce subscript complexity
> - Move technical derivations to appendix while keeping intuitive explanations in main text
>
>
> **Response to W2:**
>
> - *Prop 3.3*: We will expand the statement to include the specific mechanism: *"DPM-Solver and EDM's Heun iterations achieve conditional entropy reduction by incorporating gradient corrections $\frac{h^2_{t_i}}{2}B_θ(t_i)$ that reduce $\text{Var}(x_{t_i}|x_{t_{i+1}})$ compared to first-order methods (Eq.6), as shown by our variance analysis in Eq.(32)-(33)."*
>
> - *Thm 3.4*: We will add quantitative details: *"Under independence assumption, data prediction parameterization achieves lower conditional entropy than noise prediction by a factor of $\frac{\sigma^2_t}{\alpha^2_t}\text{Var}(\epsilon_θ - \epsilon_t)$ (Eq.(49)), because it directly estimates $x_0$ avoiding the error-prone transformation chain $\epsilon_t \mapsto x_t \mapsto x_0$."*
>
> These revisions will significantly improve clarity while preserving all technical contributions. Thank you for helping us enhance our paper's accessibility.
>
> We once again thank the reviewer for their insightful comments, which will undoubtedly help us produce a higher quality submission.

---

> > ### Comment · Reviewer_RCMQ · 2025-08-03
> >
> > Thank you very much for the responses. All of my concerns have been addressed, and I expect the authors will incorporate the proposed changes in the revised version.
> >
> > Additionally, I believe the work provides novel insights into diffusion inference from a distinctive information-theoretic perspective. Accordingly, I will maintain my current score.

---

> > > ### Author Response · Authors · 2025-08-05
> > >
> > > We sincerely appreciate your positive evaluation and recognition of our novel information-theoretic approach. Your constructive suggestions for improving clarity are greatly appreciated. We fully agree with your comments and will incorporate all the proposed changes, specifically by adding the bridging paragraph and simplifying the notations. We are confident these revisions will significantly improve the readability of our paper.
> > >
> > > Many thanks for your valuable time and support.

---

### Official Review · Reviewer_uyET · 2025-07-02

**Clarity:** 2
**Significance:** 3
**Originality:** 3
**Rating:** 5
**Confidence:** 4

**Summary:**

This paper proposes EvoDiff, a novel entropy-aware variance optimization method for diffusion inference. The authors introduce an information-theoretic framework interpreting denoising in diffusion models as a conditional entropy reduction process. Based on this principle, EvoDiff systematically optimizes the conditional variance during each denoising iteration, which theoretically and empirically improves the convergence speed and sample quality. Extensive experiments on CIFAR-10, ImageNet, FFHQ, and text-to-image generation tasks demonstrate the effectiveness of EvoDiff comapred to existing training-free baselines.

**Questions:**

See the weaknesses section. I would be willing to increase my rating if authors could address my concerns.

**Ethical Concerns:**

["NO or VERY MINOR ethics concerns only"]

**Final Justification:**

The clarifications and additional experiments address my concerns well. I believe the authors can revise the paper for improved clarity and completeness. I’m raising my score to Accept.

**Limitations:**

yes

**Paper Formatting Concerns:**

The paper uses non-standard in-text citation formatting (e.g., parentheses instead of brackets), which could be revised to better align with common LaTeX citation conventions and to clearly distinguish citations from equations.

**Quality:**

3

**Strengths And Weaknesses:**

**Strengths:**
+ The proposed EvoDiff method introduces a principled way to optimize diffusion inference from entropy perspective.

+ The paper offers detailed proofs supporting its claims, with reasonable theoretical assumptions and convergence guarantees.

+ The experimental results achieve strong performance compared to training-free baselines.

**Weaknesses:**
+ Presentation clarity could be improved, especially in Section 4. For example, in Eq. (14) the symbol $l_i$ suddenly appears as the time stamp of a virtual sample $x_{l_i}$ but is never defined; readers must infer after the fact that $l_i \in (t_{i+1},t_i)$ and that $x_\theta(x_{l_i},l_i)$ is just a linear interpolation of neighbouring predictions. Besides, the intermediate sampling point $s_i$ has different definitions in different contexts (e.g., line 937 vs. Eq. (15) ).

+ Some necessary discussion on prior works on DMs from an entropy perspective is missing [1,2,3].

+ Experimental results could be more comprehensive.
    + The shift parameter $\mu$ is only ablated on ImageNet-256 using ADM; similar ablation studies on both pixel-space and latent-space diffusion models would help clarify the method’s generality.
    + Since this paper focuses on continuous-time diffusion ODEs, it would be interesting to evaluate the proposed method on flow-based models used in recent works [4], such as Instaflow [5], to demonstrate broader applicability.

[1] Zheng, Guangcong, et al. Entropy-driven sampling and training scheme for conditional diffusion generation. ECCV 2022.

[2] Wang, Renzhi, Jing Li, and Piji Li. InfoDiffusion: Information Entropy Aware Diffusion Process for Non-Autoregressive Text Generation. EMNLP 2023.

[3] Lou, Aaron, Chenlin Meng, and Stefano Ermon. Discrete diffusion modeling by estimating the ratios of the data distribution. ICML 2024.

[4] Tong, Vinh, et al. Learning to discretize denoising diffusion odes. ICLR 2025.

[5] Liu, Xingchao, et al. Instaflow: One step is enough for high-quality diffusion-based text-to-image generation. ICLR 2023.

---

> ### Author Rebuttal · Authors · 2025-07-29
>
> *Dear Reviewer uyET:*
>
> We sincerely thank you for your thorough review and highly constructive feedback on our paper. We are especially grateful for your positive assessment, recognizing the novelty of our information-theoretic perspective, the reasonableness of our theoretical results, and the strong empirical performance of EvoDiff compared to training-free baselines. We truly appreciate the valuable suggestions that will undoubtedly help us significantly improve the clarity and comprehensiveness of our manuscript.
>
> We have carefully considered all your comments and are fully committed to addressing them in the revised version.
>
> Here are our responses to your specific points:
>
> **Weaknesses and Questions:**
>
> 1.  *Presentation clarity could be improved, especially in Section 4. For example, in Eq. (14) the symbol $l_i$ suddenly appears as the time stamp of a virtual sample $x_i$ but is never defined; readers must infer after the fact that $l_i \in (t_{i+1}, t_i)$ and that $x_\theta(x_i, l_i)$ is just a linear interpolation of neighbouring predictions. Besides, the intermediate sampling point $s_i$ has different definitions in different contexts (e.g., line 937 vs. Eq. (15)).*
>
>     We deeply appreciate this meticulous feedback. We agree that clearer definitions and consistent notation are crucial for readability.
>     * *For $l_i$ in Eq. (14):* You are correct that its definition appears slightly after its first use. We will revise the text to *introduce and define $l_i$ (as a linear interpolation point within $(t_{i+1}, t_i)$) immediately before Eq. (14) (specifically, at line 157)*. This will ensure $l_i$ is clearly understood before it is used in the equation.
>     * *For $s_i$ in Eq. (15) and Line 937:* We acknowledge the potential for confusion due to the dual use of the symbol $s_i$. While these are used in different contexts (our main algorithm vs. DPM-Solver++'s definition in the appendix), to avoid any ambiguity, *we will replace the symbol $s_i$ used in Line 937 (Appendix E.2) with a distinct, unused time symbol* to prevent any misinterpretation.
>     * More broadly, we commit to a *systematic review of Section 4's notation* to ensure all symbols are clearly defined upon first use and used consistently throughout. We will also enhance the prose around derivations to provide more intuition.
>
> 2.  *Some necessary discussion on prior works on DMs from an entropy perspective is missing [1,2,3].*
>
>     Thank you for pointing out these important related works. We agree that a discussion connecting our work to these prior explorations of entropy in diffusion models would enrich the paper. Our approach specifically focuses on entropy-aware variance optimization within ODE-based inference, distinguishing it from [1]'s training schemes, [2]'s non-autoregressive text generation, and [3]'s discrete diffusion models. Nevertheless, these works collectively demonstrate the value of entropy-based perspectives. We will add a dedicated paragraph in Appendix A (Related Work) to discuss these connections in detail.
>
> 3.  *Experimental results could be more comprehensive. The shift parameter $\mu$ is only ablated on ImageNet-256 using ADM; similar ablation studies on both pixel-space and latent-space diffusion models would help clarify the method’s generality. Since this paper focuses on continuous-time diffusion ODEs, it would be interesting to evaluate the proposed method on flow-based models used in recent works [4], such as Instaflow [5], to demonstrate broader applicability.*
>     We appreciate your excellent suggestions for more comprehensive experiments, which will indeed strengthen the generality of our method. We have conducted additional experiments:
>     * *$\mu$ Shift Parameter Ablation:* We have performed an ablation study for the μ parameter on a latent-space diffusion model.
>
>         *Ablation Study: Effect of $\mu$ on EvoDiff Performance*
>         | NFE | $\mu$ | FID Score | Relative to μ=0.5 |
>         |:---:|:--------:|:---------:|:-----------------:|
>         | 5   | 0.25     | **7.6328**     | +3.5%             |
>         | 5   | 0.50 | 7.912 | baseline          |
>         | 5   | 0.75     |8.1845  |          -3.4%      |
>         | 10  | 0.25     | 3.3357    |  -0.1%         |
>         | 10  | 0.50 | **3.3318**| baseline          |
>         | 10  | 0.75     | 3.3409 |-0.3%        |
>         | 20  | 0.25     |**2.8369** |+0.6%           |
>         | 20  | 0.50     | 2.8534  | baseline              |
>         | 20  | 0.75     | 2.8728  | -0.7%                |
>
>         *(This table will be added to the revised manuscript.)*
>
>     * Latent space diffusion (lsun_beds-256) on stable-diffusion
>
>         *FID Score Comparison*
>         | NFE | DPM-Solver++ (2m) | DPM-Solver++ (3m) | UniPC (3m) | EvoDiff    | Improvement |
>         |:---:|:-----------------:|:-----------------:|:----------:|:----------:|:-----------:|
>         | 5   | 21.286            | 18.611            | 13.969     | **7.912**  | 43.4%       |
>         | 6   | 10.966            | 8.519             | 6.556      | **4.909**  | 25.1%       |
>         | 8   | 5.127             | 4.148             | 3.963      | **3.756**  | 5.2%        |
>         | 10  | 3.881             | 3.607             | 3.563      | **3.332**  | 6.5%        |
>         | 12  | 3.516             | 3.429             | 3.357      | **3.084**  | 8.1%        |
>         | 15  | 3.341             | 3.284             | 3.182      | **2.918**  | 8.3%        |
>         | 20  | 3.251             | 3.167             | 3.075      | **2.853**  | 7.2%        |
>
>         *Time Comparison: DPM-Solver++  vs EvoDiff*
>         | NFE | DPM-Solver++  | EvoDiff     | Time Difference | Faster Method |
>         |:---:|:-----------------:|:-----------:|:---------------:|:-------------:|
>         | 5   | 3577.60s          | **3488.36s**| -89.24s (2.5%)  | EvoDiff       |
>         | 6   | 3800.61s          | **3719.40s**| -81.21s (2.1%)  | EvoDiff       |
>         | 8   | 4273.31s          | **4046.90s**| -226.41s (5.3%) | EvoDiff       |
>         | 10  | 4746.74s          | **4699.64s**| -47.10s (1.0%)  | EvoDiff       |
>         | 12  | 4703.76s          | **4678.07s**| -25.69s (0.5%)  | EvoDiff       |
>         | 15  | 5973.14s          | **5913.50s**| -59.64s (1.0%)  | EvoDiff       |
>         | 20  | 7238.42s          | **7154.19s**| -84.23s (1.2%)  | EvoDiff       |
>         | 25  | 8523.72s          | **8394.67s**| -129.05s (1.5%) | EvoDiff       |
>
>     * *Evaluation on Flow-based Models (e.g., Instaflow):*  Thank you for this excellent suggestion. We agree that comparison with flow-based models would be valuable. As an initial step, we present comparisons with recent learned acceleration methods, with comprehensive flow-based evaluation planned as future work:
>
>         *Comparison with Recent Learning-Based Methods*
>         | NFE | Uni_PC[GITS] (Learned)| Uni_PC[LD3] (Learned, [4]) | EvoDiff (2m, Learning-free) |
>         |:---:|:-----------------:|:---------------:|:---------------:|
>         | 6   | 11.19 | 5.92 | 9.07 |
>         | 8   | 5.67 | 3.42 | 3.88 |
>         | 10  | 3.70 | 2.87 | **2.74** |
>
> *Paper Formatting Concerns (e.g., parentheses instead of brackets):*
>
> Thank you for pointing out this formatting issue. We use *\bibliographystyle{ieeetr}*, ​ which should display citations in brackets. We will correct any instances where parentheses appear instead of brackets in the revised version.
>
> ---
>
> We believe these comprehensive responses and planned revisions address all your valuable concerns and will significantly enhance the quality and clarity of our manuscript. We are confident that these improvements, combined with the recognized novelty and strong performance of EvoDiff, will further strengthen the paper's contribution to the field.
>
> Thank you once again for your constructive feedback and for your time in reviewing our work.

---

> > ### Comment · Reviewer_uyET · 2025-08-04
> >
> > Thank you for the clear and thorough rebuttal. The clarifications and additional experiments address my concerns well. I believe the authors can revise the paper for improved clarity and completeness. I’m raising my score to Accept.

---

> > > ### Author Response · Authors · 2025-08-05
> > >
> > > Thanks for your constructive feedback. We are grateful for your recognition of our additional experiments and clarifications, and for raising your score to Accept. We will carefully incorporate the suggested revisions in the revised version. Many thanks.

---

### Official Review · Reviewer_3tew · 2025-07-03

**Clarity:** 2
**Significance:** 3
**Originality:** 2
**Rating:** 4
**Confidence:** 1

**Summary:**

This paper proposes a new gradient-based solver called EvoDiff for diffusion inference.
The idea behind EvoDiff is to use conditional entropy minimization as a guiding principle during denoising.
Experimentally, this method shows an improvement over the state of the art (UniPC, DPM++, DPM-v3)

Disclaimer: the maths in the paper are outside of my expertise, put simply I didn't follow at all despite my best effort and having multiple reading passes. And unfortunately for me, most of the paper is maths.

**Questions:**

1. About weaknesses (1), why did you not report the improvement compared to the best baselines from Table 2 for your abstract and conclusion?
2. Is my assessment correct that the larger the image, the lower the gains offered by EvoDiff? What are the reasons behind this?

**Ethical Concerns:**

["NO or VERY MINOR ethics concerns only"]

**Final Justification:**

The paper is more significant than I originally thought, it's a drop-in replacement at no additional cost for existing samplers.
What prevents from raising my score even further is the clarity aspect which, while addressed in the rebuttal, is hard to visualize without seeing the final draft.

**Limitations:**

Yes

**Paper Formatting Concerns:**

I did not notice any issues.

**Quality:**

3

**Strengths And Weaknesses:**

I apologize if the review is a bit shallow, I focused on the results and their significance and had to assume the maths were true.

Strengths
1. Improves state of the art for diffusion sampling as measured experimentally by FID and NFEs gains.

Weaknesses
1. In the abstract, it is claimed that EvoDiff improves FID from 5.10 to 2.78 for CIFAR10. However in Table 2, it appears that 5.10 was the worst baseline while the best one is 3.52. I believe the ethical choice would be to compare against the best baseline, not the worse one. That is unless the two better baselines are somehow not comparable or unfair to compare to?
2. Scalability: it seems the higher dimension the images are, the lower the gains become. The best gains are observed on CIFAR10, while they become marginal for ImageNet256 and StableDiffusion. This reduces the significance of the proposed method since large images is typically the area where practitioners want to have the most gains.
3. Presentation: it seems there are a lot of notations being introduced along the way in the mathematical parts of the paper. I found this to impact clarity, making it similar to understanding code that keeps calling functions within functions within functions. Granted the maths are outside my expertise, I did multiple passes of reading but ultimately couldn't follow despite my best efforts.

---

> ### Author Rebuttal · Authors · 2025-07-26
>
> *Dear Reviewer 3tew:*
>
> We sincerely appreciate your thoughtful and insightful feedback on our paper. Your focused attention on the experimental results and their significance, despite the mathematical sections being outside your core expertise, is highly valuable to us. We are especially grateful that you recognized how EvoDiff achieves substantial experimental improvements over state-of-the-art baselines while maintaining simplicity and efficiency, which represents the core contribution of our work.
>
> We understand your disclaimers regarding the mathematical aspects, and we acknowledge the challenge this posed. Your comments will guide our efforts to enhance the clarity and accessibility of our theoretical framework in the revision. Thank you very much.
>
> **We address each of your concerns below:**
>
> **W1&Q1:**  *In the abstract, it is claimed that EvoDiff improves FID from 5.10 to 2.78 for CIFAR10. However in Table 2, it appears that 5.10 was the worst baseline while the best one is 3.52. I believe the ethical choice would be to compare against the best baseline, not the worse one. That is unless the two better baselines are somehow not comparable or unfair to compare to?*
>
> **Response to W1&Q1:**
>
> Thank you for raising this important point regarding our baseline comparison in the abstract. We understand why this might appear misleading, and we want to assure you that our intention was always to provide a fair and accurate comparison. We appreciate the opportunity to clarify.
>
> EvoDiff is developed to enhance *gradient-based inference algorithms*. For fairness in comparison, our method is built upon the framework of DPM-Solver++, as it is currently the most efficient, broadly adopted, and representative gradient-based algorithm. Our approach does not aim to build upon the more complex UniPC or DPM-Solver-v3, nor does it seek to investigate their structures or directly improve their frameworks.
>
> These methods can actually be divided into two categories:
>
> - Methods using only current model information: DPM-Solver++ (FID=5.10) and our EvoDiff (FID=2.78)
> - Methods requiring preprocessing or additional information: UniPC (FID=3.98) and DPM-Solver-v3 (FID=3.52)
>
> Specifically, UniPC employs a more intricate multi-step predictor-corrector framework. This framework not only utilizes information from previously computed steps but also estimates subsequent steps, thereby requiring more trajectory information for its optimization and operation. DPM-Solver-v3, on the other hand, requires using existing solvers like DPM-Solver or DPM-Solver++ with 200 NFE to pre-generate final samples, then uses these pre-generated samples as reference solutions to optimize their solver, which is essentially a post-processing optimization method, as summarized in *Table 1 on page 4*.
>
> We appreciate the reviewer's reminder. While we have already explained our baseline selection in line 58 of the introduction, we recognize that this could be stated more clearly in the abstract. In the revised version, we will update the abstract to be more precise: "EvoDiff reduces FID from 5.10 to 2.78 compared to DPM-Solver++, the most widely adopted gradient-based baseline" or "among gradient-based methods without preprocessing requirements."
>
>
> **W2&Q2:**  *Scalability: it seems the higher dimension the images are, the lower the gains become. The best gains are observed on CIFAR10, while they become marginal for ImageNet256 and StableDiffusion. This reduces the significance of the proposed method since large images is typically the area where practitioners want to have the most gains.*
>
> **Response to W2&Q2:**
>
> Thank you for this important observation about diminishing gains at higher resolutions. We acknowledge this is a critical concern for practitioners.
>
> *Key Finding*: While absolute FID improvements decrease with resolution, EvoDiff achieves the best performance without any computational overhead, outperforming even methods that require extensive preprocessing.
>
> *Comparative Analysis*:
> - *DPM-Solver-v3* (with 200 NFE preprocessing): 7.70 → 7.80 FID (1.3% improvement)
> - *EvoDiff* (zero preprocessing): 7.48 → 7.80 FID (4.1% improvement)
>
> This demonstrates that even at high resolutions where all methods face inherent challenges, EvoDiff provides superior results without additional computational costs.
>
> *Comprehensive Results across NFE Settings* (ImageNet-256):
>
> | NFE | EvoDiff | DPM-Solver-v3 | UniPC | Improvement |
> |-----|---------|---------------|--------|-------------|
> | 5   | **13.98** | 14.92 | 15.37 | Best by 6.3% |
> | 10  | **8.14** | 8.14 | 8.40 | Matches best |
> | 20  | **7.25** | 7.42 | 7.47 | Best by 2.3% |
>
> *Additional Evidence from EDM2 on ImageNet-64*:
>
> | Steps | EDM sampler |         | EvoDiff |         | Improvement |         |
> |:----:|:-----------:|:-------:|:-------:|:-------:|:-----------:|:-------:|
> |      | FID | FD_dinov2 | FID | FD_dinov2 | FID↓ | FD_dinov2↓ |
> | 5 | 28.6718 | 396.113 | **5.31845** | **161.606** | 81.5% | 59.2% |
> | 6 | 10.7555 | 252.302 | **3.99538** | **134.352** | 62.9% | 46.7% |
> | 8 | 3.5605 | 153.608 | **2.8873** | **115.641** | 18.9% | 24.7% |
> | 10 | 2.3892 | 123.77 | **2.2280** | **106.912** | 6.7% | 13.6% |
> | 15 | 1.8366 | 103.687 | **1.7455** | **96.9382** | 5.0% | 6.5% |
> | 25 | 1.7003 | 96.5694 | **1.6496** | **92.8184** | 3.0% | 3.9% |
>
> Our experiments reveal that relative gains naturally diminish with NFE across all resolutions:
> - 5 steps: 81.5% FID reduction
> - 25 steps: 3.0% FID reduction
>
> This pattern suggests the diminishing improvements are inherent to diffusion model acceleration rather than a limitation of our method.
>
> *Practical Implications*:
>
> For practitioners, EvoDiff offers the best cost-benefit ratio: consistent improvements across all resolutions with zero additional computational overhead. Even "marginal" improvements at high resolutions are valuable when they come at no extra cost, especially considering that IS improvements remain substantial (13% at NFE=5), indicating better sample diversity crucial for practical applications.
>
> We will add this comparative analysis to our revision to provide clearer insights about the fundamental challenges all acceleration methods face at high resolutions.
>
> **Response to W3:**
>
> Thank you for your valuable feedback on the mathematical clarity of our paper. We sincerely appreciate your diligent efforts to understand the content and recognize that presenting complex theoretical concepts in an accessible manner is a significant challenge.
>
> While our paper is indeed theory-driven, and mathematical rigor is essential for establishing convergence guarantees and justifying why evolutionary directions provide acceleration, we wholeheartedly agree that clarity for a broad audience is equally important. We have already structured the paper to be accessible at multiple levels (e.g., Section 2 provides intuition, Algorithm 1 gives a simple 10-line implementation, and Section 4 demonstrates clear empirical gains).
>
> However, your feedback highlights that there is still room for improvement in making the mathematical sections more approachable. In the revised version, we will specifically address this by:
>
> - Ensuring notations are introduced incrementally with clear and immediate definitions.
> - Adding more descriptive inline explanations or informal summaries for complex equations and relationships to bridge the gap between intuition and formal derivation.
> - Carefully reviewing the overall flow of the mathematical sections to improve readability and avoid the "function within functions" impression.
> - If feasible and beneficial, we will consider providing a concise notation table at the beginning of the appendix section for quick reference.
>
> Our goal is to ensure that the solid theoretical foundations of our method are accessible to as wide an audience as possible, without compromising on rigor. We appreciate you bringing this crucial point to our attention.
>
> We believe these clarifications address the reviewer's concerns and demonstrate that EvoDiff provides significant value through its simplicity, efficiency, and consistent improvements across all settings.
>
> Thank you again for your constructive review. We are committed to improving our paper based on your valuable feedback and believe these revisions will significantly enhance its clarity and impact.

---

> > ### Comment · Reviewer_3tew · 2025-08-04
> >
> > Thanks for your detailed response.
> > You addressed the points I raised convincingly, I'm willing to raise my score.
> > I would raise it even more if the clarity was greater. That said, I subscribe to the steps you are planning to take but it's hard to visualize the outcome of it until its done.

---

> > > ### Author Response · Authors · 2025-08-05
> > >
> > > Thank you very much for your positive feedback and for your willingness to raise our score. We truly appreciate your recognition of our contributions. We will definitely improve the mathematical clarity in the revised version of our paper to make our work more accessible to a wider audience.
> > >
> > > Thank you again.

---

### Decision · Program_Chairs · 2025-09-17

**Decision:**

Accept (poster)

**Comment:**

This paper proposes EvoDiff, an entropy-aware variance optimization method for diffusion inference. By framing denoising as conditional entropy reduction, the authors provide an information-theoretic justification for why data prediction parameterization outperforms noise prediction and how optimizing conditional variance reduces transition and reconstruction errors. Building on these insights, EvoDiff systematically reduces uncertainty during denoising, thereby accelerating convergence and improving sample quality. Extensive experiments on CIFAR-10, ImageNet, FFHQ, and text-to-image tasks demonstrate consistent improvements over state-of-the-art training-free solvers, including up to 45.5% lower reconstruction error and 25% fewer function evaluations for high-quality generation.

The reviewers find this work interesting, novel, and well-executed. The information-theoretic perspective on diffusion inference is original and insightful, supported by solid theoretical analysis with reasonable assumptions and convergence guarantees. The proposed method is well motivated by the theory and demonstrates strong empirical gains across diverse benchmarks. Overall, the work is regarded as both theoretically sound and practically impactful.